# Multiomic QTL mapping reveals phenotypic complexity of GWAS loci and prioritizes putative causal variants

## Graphical abstract

## Authors

Timothy D. Arthur, Jennifer P. Nguyen, Benjamin A. Henson, ..., Matteo D'Antonio, Graham McVicker, Kelly A. Frazer

## Correspondence

kafrazer@health.ucsd.edu

## In brief

Arthur and Nguyen et al. leverage iPSCORE, a large collection of three early developmental-like tissues with corresponding molecular datasets (RNA-seq, ATAC-seq, H3K27ac ChIP-seq), to map over 70,000 multiomic QTLs. By integrating these multiomic QTLs with GWAS loci, they advance our understanding of complex traits and provide a resource of putative causal variants for experimental validation.

## Highlights

- Mapped 70,446 multiomic QTLs affecting gene expression and chromatin phenotypes

- Chromatin QTLs capture GWAS loci that are not associated with expression QTLs

- iPSCORE QTLs exhibit temporal activity consistent with early developmental stages

- Multiomic QTLs greatly improve the prioritization of putative causal GWAS variants

 Arthur et al., 2025, Cell Genomics 5, 100775
March 12, 2025 © 2025 The Authors. Published by Elsevier Inc.

# Cell Genomics

CellPress

## Article

# Multiomic QTL mapping reveals phenotypic complexity of GWAS loci and prioritizes putative causal variants

Timothy D. Arthur,[1,2,11] Jennifer P. Nguyen,[2,3,11] Benjamin A. Henson,[4] Agnieszka D'Antonio-Chronowska,[5] Jeffrey Jaureguy,[3,6] Nayara Silva,[7] iPSCORE Consortium, Athanasia D. Panopoulos,[8,9] Juan Carlos Izpisua Belmonte,[10] Matteo D'Antonio,[2] Graham McVicker,[6] and Kelly A. Frazer[4,7,12,*]

[1]Biomedical Sciences Program, University of California, San Diego, La Jolla, CA 92093, USA
[2]Department of Biomedical Informatics, University of California, San Diego, La Jolla, CA 92093, USA
[3]Bioinformatics and Systems Biology Graduate Program, University of California, San Diego, La Jolla, CA 92093, USA
[4]Institute of Genomic Medicine, University of California San Diego, La Jolla, CA 92093, USA
[5]Center for Epigenomics, University of California San Diego, School of Medicine, La Jolla, CA 92093, USA
[6]Integrative Biology Laboratory, Salk Institute for Biological Studies, La Jolla, CA 92037, USA
[7]Department of Pediatrics, University of California, San Diego, La Jolla, CA 92093, USA
[8]Board of Governors Regenerative Medicine Institute, Cedars-Sinai Medical Center, Los Angeles, CA 90048, USA
[9]Department of Biomedical Sciences, Cedars-Sinai Medical Center, Los Angeles, CA 90048, USA
[10]Altos Labs, Inc., San Diego, CA 94022, USA
[11]These authors contributed equally
[12]Lead contact
*Correspondence: kafrazer@health.ucsd.edu

## SUMMARY

Most GWAS loci are presumed to affect gene regulation; however, only ~43% colocalize with expression quantitative trait loci (eQTLs). To address this colocalization gap, we map eQTLs, chromatin accessibility QTLs (caQTLs), and histone acetylation QTLs (haQTLs) using molecular samples from three early developmental-like tissues. Through colocalization, we annotate 10.4% (n = 540) of GWAS loci in 15 traits by QTL phenotype, temporal specificity, and complexity. We show that integration of chromatin QTLs results in a 2.3-fold higher annotation rate of GWAS loci because they capture distal GWAS loci missed by eQTLs, and that 5.4% (n = 13) of GWAS colocalizing eQTLs are early developmental specific. Finally, we utilize the iPSCORE multiomic QTLs to prioritize putative causal variants overlapping transcription factor motifs to elucidate the potential genetic underpinnings of 296 GWAS-QTL colocalizations.

## INTRODUCTION

Over 90% of genome-wide association study (GWAS) loci are in non-coding regions of the genome, and the causal variants at these loci are presumed to modulate the expression of genes. Expression quantitative trait loci (eQTL) analyses have been employed to interpret the regulatory function of GWAS signals, however, only ~43% of GWAS loci colocalize with eQTLs identified in postmortem adult tissues.[1] Various hypotheses have been proposed to explain this colocalization gap including that regulatory variants may not be captured by eQTLs identified in adult bulk tissues because they are only active in context-specific conditions (e.g., early fetal development,[2–7] rare cell types[3,8–11]). In addition, it has been proposed that current eQTL study sample sizes are underpowered and biased toward discovering common variants that overlap promoters and have large effects on gene expression.[12] In contrast, GWAS loci often overlap distal regulatory elements with small effects.[12] These proposed hypotheses for the existing colocalization gap could be examined by focusing on

cells representing early developmental time points, and by mapping QTLs for other types of molecular assays that specifically capture the function of distal regulatory elements such as enhancers.

The iPSC Omics Resource (iPSCORE)[2,3,13–24] was developed to study the association between regulatory variation and molecular phenotypes in tissues representing early developmental stages. iPSCORE is composed of early embryonic-like induced pluripotent stem cells (iPSCs) from hundreds of individuals with whole-genome sequencing (WGS) data,[13,14,19] as well as fetal-like iPSC-derived cardiovascular progenitor cells (CVPCs)[3,20–22] and iPSC-derived pancreatic progenitor cells (PPCs).[2] We have previously shown that the iPSCs, CVPCs, and PPCs are suitable surrogate models to identify eQTLs active during embryonic and fetal-like stages because they exhibit early development-like (EDev-like) molecular properties.[2,3,14] Moreover, we have shown the utility of combining the iPSCORE EDev-like and the GTEx adult expression datasets[1] to functionally annotate GWAS regulatory variants in a

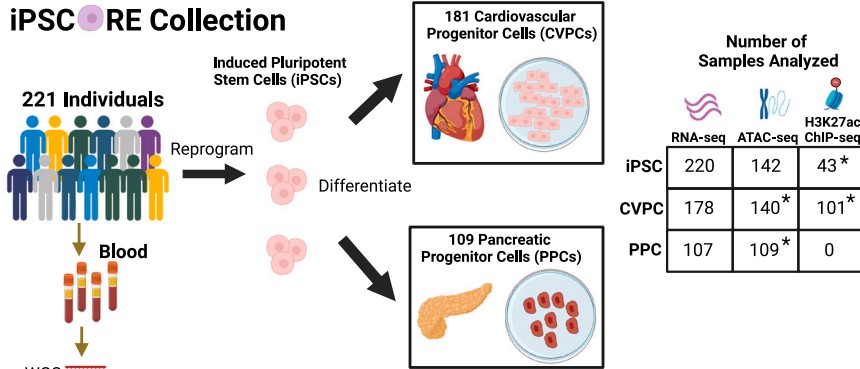

**Figure 1. Overview of iPSCORE multiomic samples**

Overview of the iPSCORE molecular samples generated from blood, reprogrammed iPSCs, and derived tissues. Of the 1,261 molecular samples, 861 were previously published and 400 were newly released in this study. In addition to the 393 samples in the four new molecular datasets (indicated by asterisks), 7 of the 220 iPSC RNA-seq samples were not previously published. WGS analyses identified 16,360,123 single-nucleotide polymorphisms (SNPs). The RNA-seq, ATAC-seq, and H3K27ac ChIP-seq libraries were sequenced to median depths of 71.7, 90.9, and 52.1 million reads, respectively (see methods). Created in Biorender. https://BioRender.com/d18y377.

temporal-specific manner[2,3] using colocalization[25] and evaluate the fetal origins of disease hypothesis.[26–28]

Molecular assays for transposase-accessible chromatin (ATAC-seq) and chromatin immunoprecipitation for H3K27 acetylation (H3K27ac ChIP-seq) capture both proximal and distal regulatory elements that modulate gene expression.[29,30] H3K27ac peaks primarily mark active promoters and enhancers,[31,32] whereas ATAC-seq identifies all open chromatin regions, which may contain both activating and repressive regulatory elements as well as insulators.[29] Integration of chromatin accessibility QTL (caQTL) and histone acetylation QTL (haQTL) mapping with eQTL analyses in the iPSCORE collection could be useful for discovering variants that affect different types of regulatory elements and enhance the annotation of GWAS variants,[33–39] particularly those that are not associated with eQTLs.

To better understand the utility of using multiomic QTLs from early developmental-like (EDev-like) tissues to functionally annotate GWAS loci through colocalization, we analyzed RNA-seq, ATAC-seq, and H3K27ac ChIP-seq samples generated from iPSCs, CVPCs, and PPCs derived from 221 individuals in the iPSCORE collection. We mapped 70,446 QTLs including, 25,659 eQTLs, 33,618 caQTLs, and 11,169 haQTLs. We annotated the QTLs based on chromatin states, stage specificity, and the number of qElements (genes and/or peaks) that they affect (complexity). We performed colocalization with 5,192 GWAS loci from 15 developmental and adult traits and diseases and identified 540 loci that colocalized with a QTL. We next integrated the QTL annotations to functionally characterize GWAS loci. Of all 5,192 GWAS loci, 5.8% (n = 301) only colocalized with caQTLs and/or haQTLs (chromatin QTLs) while 4.6% (n = 239) colocalized with an eQTL. We show that this 2.3-fold increase in colocalization rate is due to chromatin QTLs capturing GWAS loci that are distal to promoters and not captured with the traditional eQTL approach. Of the 239 GWAS loci that colocalized with an eQTL, 5.4% (n = 13) were associated with eQTLs that were specific to early developmental stage (EDev-specific), supporting that regulatory variation active during fetal development contributes to adult disease.[2,28] We also show that complex QTLs affecting multiple qElements (genes and/or peaks) have higher GWAS colocalization rates compared with singleton QTLs that only affect one element. Finally, we demonstrate the utility of multiomic QTLs for prioritizing putative causal GWAS-associated regulatory variants based on their overlap

with transcription factor (TF) motifs, and highlight two examples of causal variants that we identified using this approach with previous experimental validation and/or inferred TF disrupting activity.[40,41]

In summary, our study shows that integrative multiomic QTL analyses could explain a large proportion of GWAS loci that do not colocalize with eQTLs alone. We colocalized a large set of temporally annotated iPSCORE EDev-like eQTLs with GWAS loci and show that 5.4% of the colocalized eQTLs were EDev-specific, while most were also active in adult tissues. Finally, we show the utility of the iPSCORE multiomic QTLs for prioritizing putative causal variants underlying GWAS loci.

## RESULTS

### Overview of molecular datasets

We analyzed 1,261 molecular samples including WGS and three molecular data types generated from three different iPSCORE early developmental like (EDev-like) tissues (Figure 1) from 221 ethnically diverse iPSCORE subjects (170 Europeans, 4 Africans, 34 East Asians, 6 South Asians, and 7 Admixed Americans, Figure S1; Table S1).[13] Specifically, we examined RNA-seq, ATAC-seq, and H3K27ac ChIP-seq (Table S2), from 220 iPSC lines, 181 iPSC-derived CVPCs, and 109 iPSC-derived PPCs. The resource contains paired molecular data types (RNA-seq, ATAC-seq, and H3K27ac ChIP-seq) from over 100 samples of each tissue (iPSCs, CVPCs, and PPCs), as well as paired molecular data types across tissues, as the majority of CVPCs and PPCs are derived from an iPSC line that was also molecularly profiled (Figure S2). Of the 1,261 molecular samples, 400 are newly released in this study (Figure 1), and 861 have been previously published.[2,13,14,19,21]

### iPSCORE early developmental-like tissues display lineage-specific regulatory landscapes

To examine the epigenomic properties of the three EDev-like tissues, we performed ATAC-seq on 142 iPSCs, 140 CVPCs, and 109 PPCs, as well as H3K27ac ChIP-seq on 43 iPSCs and 101 CVPCs (Figure 1). For each tissue and data type, we called consensus peaks using a subset of the samples from unrelated individuals. After filtering (see methods), we identified 172,075 iPSCs, 202,941 CVPCs, and 193,428 PPC ATAC-seq peaks. Across the EDev-like tissues, we observed similar proportions

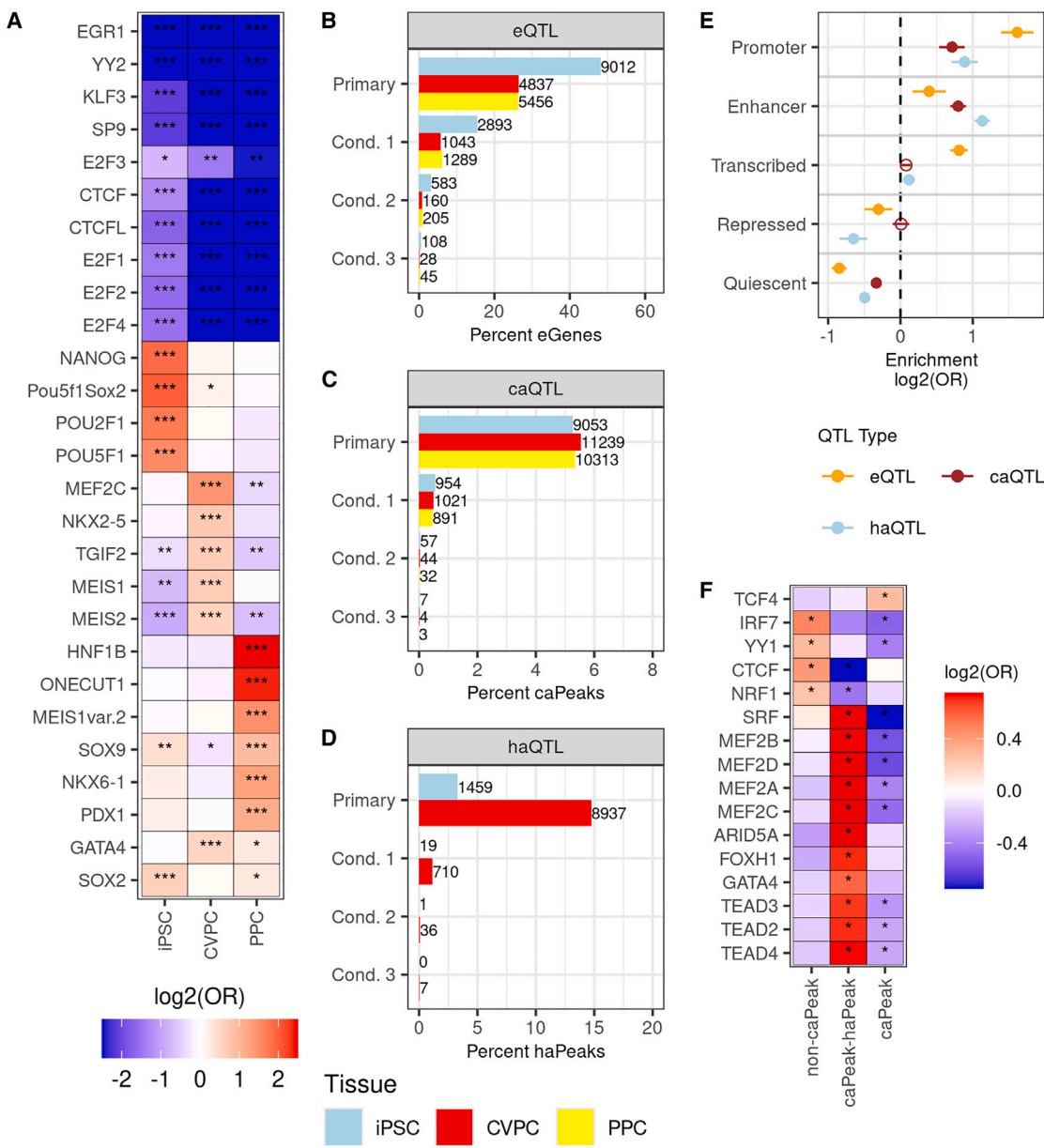

**Figure 2. Characterization of multiomic regulatory variation in early developmental tissues**

(A) Heatmap showing TFBS enrichments in iPSC-, CVPC-, and PPC-specific ATAC-seq peaks. Two-sided Fisher's exact tests were performed to test the enrichment (odds ratio) of the predicted TFBSs in each of the three ATAC-seq peak sets. TFBSs depleted in all three of the tissue-specific ATAC-seq peaks are considered shared. Each cell is filled with the $log_2$(odds ratio) of the association between predicted TFBSs (y axis) and tissue-specific ATAC-seq peaks (x axis). Asterisks indicate significant enrichments (Benjamini-Hochberg adjusted ***$p < 5 \times 10^{-10}$; Benjamini-Hochberg adjusted **$p < 5 \times 10^{-3}$; Benjamini-Hochberg adjusted *$p < 0.05$). Limits for $Log_2$(odds ratio) were set to $-2.5$ and $2.5$ for plot legibility.

(B–D) Bar plots showing the percent of qElements (eGenes, caPeaks, and haPeaks) with at least one eQTL (B), caQTL (C), and haQTL (D). For eQTLs, variants (MAF > 0.05) within 1 Mb of each gene were tested for an association with gene expression. For chromatin QTLs, variants (MAF > 0.05) within 100 kb of each peak were tested for an association with chromatin accessibility (C) or histone acetylation (D). If a QTL was discovered for a gene or peak, up to three additional conditional QTLs were tested by using the lead variant as a covariate. The reported numbers reflect the conditional QTLs remaining after the filtering step. The x axis is the percent of qElements with a QTL for each tissue and the y axis is QTL type (i.e., primary or conditional). Each bar is colored by tissue (iPSC, light blue; CVPC, red; PPC, yellow).

(E) Plot showing the enrichment of primary CVPC eQTLs, caQTLs, and haQTLs in chromatin states. The x axis is the enrichment $log_2$(odds ratio) and the y axis contains the five collapsed chromatin states. The points are colored by the QTL type (eQTL, orange; caQTL, brown; and haQTL, light blue). The whiskers represent the $log_2$ upper and lower 95% confidence intervals. Significant enrichments are represented by filled circles and non-significant enrichments are represented by circles without a fill. Enrichment of primary iPSC eQTLs, caQTLs, and haQTLs in chromatin states is shown in Figure S9.

*(legend continued on next page)*

of ATAC-seq peaks in promoters (mean = 16.3%), intergenic regions (mean = 28.0%), and intronic regions (mean = 46.3%; Figure S3). Similarly, after filtering, we identified 44,206 consensus H3K27ac ChIP-seq peaks in iPSCs and 60,556 in CVPCs. The largest proportion of ChIP-seq peaks were in intronic regions (39.4%) and promoters (32.6%), while a smaller proportion were in intergenic regions (17.0%; Figure S3). A UMAP analysis of the ATAC-seq and ChIP-seq peaks showed that samples cluster by tissue, indicating that the iPSCs and the derived EDev-like CVPCs and PPCs each have distinct regulatory landscapes (Figures S4A and S4B).

To examine the TF binding patterns of the EDev-like tissue specific and shared regulatory elements (Figure S4C), we performed footprinting analysis to predict TF binding sites (TFBSs) for 1,147 motifs in the consensus ATAC-seq peaks for each tissue.[42–44] TFs with known roles in pluripotency, cardiac development, and pancreatic development were strongly enriched in iPSC-, CVPC-, and PPC-specific ATAC-seq peaks, respectively (Figure 2A; Table S3). For example, NANOG and POU5F1 TFBSs[19,45] were exclusively enriched in iPSC-specific ATAC-seq peaks, MEF2 and NKX2-5[20,21] TFBSs were strongly enriched in CVPC-specific ATAC-seq peaks, and HNF1B,[46] ONECUT1,[47] MEIS1,[48] NKX6-1,[49] and PDX1[50] TFBSs were strongly enriched in PPC-specific ATAC-seq peaks. Alternatively, TFs associated with essential cellular processes including chromatin organization[51] (CTCF, CTCFL) and cell growth during the G1 phase[52] (E2F family) were strongly enriched in shared ATAC-seq peaks (Figures 2A and S4C). As expected, TFs associated with tissue-specific ATAC-seq peaks had higher expression in the corresponding tissue, and TFs associated with binding sites enriched in shared ATAC-seq peaks had similar expression levels in all three tissues (Figure S5).

In summary, tissue-specific regulatory elements in the EDev-like iPSC, cardiac, and pancreatic tissues are bound by appropriate lineage-specific developmental TFs, and shared regulatory elements are bound by TFs governing essential cellular processes such as chromatin organization and cell growth.

### Identification of multiomic regulatory variation in iPSCORE tissues

To identify and characterize regulatory variation associated with the three molecular phenotypes (gene expression, open chromatin, H3K27 acetylation) in the iPSCORE EDev-like tissues, we established a two-step quantitative trait loci (QTLs) pipeline (Figure S6). In the first step, we utilized a linear mixed model to discover QTLs by calculating the association between single-nucleotide polymorphism (SNP) genotypes (5.5 M with MAF > 5%) and molecular phenotypes, while controlling for relatedness of iPSCORE donors. Multiple independent QTLs often exist for a given qElement (gene or peak associated with a QTL),[53] therefore we calculated up to three conditional QTLs for each data type. In the second step, we filtered conditional

QTLs that were not independent based on their high LD (linkage disequilibrium; $r^2 \geq 0.8$ and/or $D' \geq 0.8$) with the primary lead variant (Figure S6).

We identified 25,659 eQTLs (19,305 primary and 6,354 conditional) for 19,305 eGenes across the three iPSCORE EDev-like tissues (Figure 2B; Table S4). iPSCs had approximately 1.7-fold more eGenes ($n = 9,012$) than the CVPCs ($n = 4,837$) and PPCs ($n = 5,456$) most likely because there were more iPSC samples, and they have lower cellular heterogeneity. To examine the relative power of identifying eQTLs, we compared the three tissues with the 49 tissues in the GTEx Consortium[1] and showed that they had similar eGene discovery rates (Figure S7).

We identified 33,618 caQTLs (30,605 primary and 3,013 conditional) for 30,605 caPeaks (ATAC-seq peaks with at least one caQTL, Figure 2C; Table S4). Across all three tissues, between 5% and 6% of accessible ATAC-seq peaks had a caQTL. We examined the caQTL discovery rate by ATAC-seq peak width and observed that caQTL discovery rates were highest for ATAC-seq peaks with widths between 751 and 1,000 bp (Figure S8). To evaluate whether eGenes are more likely to have proximal caPeaks than genes without an eQTL, we first defined a 100 kb window upstream of expressed genes in each tissue. We then performed Fisher's exact tests to evaluate the enrichment of caPeaks in windows upstream of eGenes, using genes without an eQTL as background. In all tissues, caPeaks were enriched upstream of eGenes (iPSC odds ratio [OR] = 1.3, $p = 1.1 \times 10^{-16}$; CVPC OR = 1.3, $p = 2.6 \times 10^{-17}$; PPC OR = 1.4, $p = 4.4 \times 10^{-26}$), which is consistent with caPeaks being regulatory elements of neighboring genes whose transcription is affected by variants.

In the iPSC and CVPC tissues, we identified 11,169 haQTLs (10,396 primary and 773 conditional) for 10,396 haPeaks (H3K27ac peaks with at least one haQTL) (Figure 2D; Table S4). Of the 11,169 haQTLs, 9,690 were detected in the CVPCs and 1,479 were detected in iPSCs, reflecting the greater number of CVPC H3K27ac samples ($n = 101$ CVPCs versus $n = 43$ iPSCs). Of note, ~15% of the CVPC H3K27ac peaks had at least one haQTL, which was approximately 3-fold greater than the percent of CVPC ATAC-seq peaks with caQTLs.

In summary, we identified 70,446 QTLs across the three iPSCORE EDev-like tissues, making this one of the largest reports of multiomic QTLs from paired samples. We show that caPeaks are enriched for being near eGenes, and that haPeaks are discovered at a ~3-fold high rate than caPeaks.

### Integrative QTL analyses capture variation impacting different types of regulatory elements

To examine whether regulatory variation affecting the three molecular phenotypes is located in different types of regulatory elements, we annotated the iPSC and CVPC primary QTL lead variants with five chromatin states (promoters, enhancers, transcribed, repressed, and quiescent regions). For each tissue, we

---

(F) Heatmap showing the enrichment of TFBSs in CVPC ATAC-seq peaks without caQTLs (non-caPeaks), with caQTLs overlapping haQTLs (caPeak-haPeak), and with caQTLs not overlapping haQTLs (caPeaks). For each category, a two-sided Fisher's exact test was performed to test the enrichment of TFBSs, using the other two categories as background. The y axis represents the TFBSs, the x axis corresponds to the ATAC-seq peak annotation, and each cell is filled with the corresponding $\log_2$(odd ratio) from the Fisher's exact test. Asterisks (*) indicate significant enrichments (Benjamini-Hochberg adjusted $p < 0.05$). Limits for $\log_2$(odds ratio) were set to $-0.75$ and $0.75$ for plot legibility.

tested the QTL chromatin state enrichment, using lead variants for non-significant elements (i.e., genes and peaks) as background (Figure 2E; Figure S9). Consistent with previous findings,[1,12] CVPC primary eQTLs are most significantly enriched in promoters (OR = 3.1) and transcribed regions (OR = 1.8), exhibit weaker enrichments in enhancers (OR = 1.2), and are strongly depleted in quiescent chromatin (OR = 0.56; Figure 2E). CVPC primary caQTLs are strongly enriched in both promoters (OR = 1.6) and enhancers (OR = 1.7), and, similarly, CVPC primary haQTLs are enriched in promoters (OR = 1.9) and enhancers (OR = 2.2) and exhibit a weak enrichment in transcribed regions (OR = 1.1; Figure 2E). iPSC primary QTLs showed similar chromatin state enrichments (Figure S9). These findings support that eQTLs are biased toward identifying regulatory variation in promoters,[1,12] and suggest that chromatin QTLs (caQTLs and haQTLs) capture both promoter- and enhancer-acting regulatory variation.

Recent studies have shown that some TFs are more likely than other TFs to have their binding sites impacted by variants.[35,54] To investigate if different TFs bind CVPC caPeaks and haPeaks versus non-caPeaks (i.e., peaks not associated with a caQTL), we examined the 55,331 CVPC ATAC-seq peaks with at least one predicted TFBS. We next binned the peaks into three categories: ATAC-seq peaks without a caQTL (non-caPeak), caPeaks that overlap haPeaks (caPeaks-haPeak), and caPeaks that do not overlap an haPeak (caPeaks). For each category, we performed Fisher's exact tests to evaluate the enrichment of TFBSs, using the other two categories as background (Figure 2F). Predicted TFBSs for four TFs were enriched in the 51,097 non-caPeaks (Figure 2F) including CTCF TFBSs (OR = 1.3, $p = 7 \times 10^{-14}$). The binding sites of several cardiac TF markers (e.g., MEF2 TFs) were enriched in the 1,641 caPeaks that overlapped haPeaks, and the 2,593 caPeaks not overlapping haPeaks (Figure 2F). Our observations show that caPeaks harbor different predicted TFBSs than non-caPeaks, which is consistent with previous findings that regulatory variation impacts the binding of some TFs more than others.[35,54]

In summary, we show that regulatory variation affecting the three molecular phenotypes (i.e., gene expression, chromatin accessibility, or histone acetylation) is located in different types of regulatory elements and that caPeaks harbor different predicted TFBSs than non-caPeaks.

## Identification and functional characterization of early developmental-specific QTLs

To identify and characterize temporal QTLs only active during early development (EDev-specific), we used *mashr*,[55] which accounts for the correlation structure between eQTLs across multiple tissues and estimates condition specificity (i.e., temporal stage, cell-type, response to stimuli) by calculating a local false sign rate (LFSR) for each SNP-eGene pair. We focused on the 19,305 iPSCORE primary eQTLs due to the lack of suitable adult chromatin QTL datasets. We applied *mashr* on 250,564 eQTL lead variants for 33,793 unique eGenes (281,938 SNP-eGene pairs) from iPSCORE and GTEx[1] tissues. We removed all SNP-gene pairs that either were not tested in at least one adult and at least one EDev-like tissue or were not significant in any tissue (minimum LFSR > 0.05), resulting in 102,375 SNP-eGene pairs for 10,984 eGenes. We identified 2,299

EDev-specific SNP-eGene pairs that were significant in at least one iPSCORE tissue (LFSR < 0.05) and not significant in any adult GTEx tissue, 27,881 adult-specific SNP-eGene pairs, and 72,195 shared SNP-eGene pairs that were significant at least one iPSCORE tissue and one adult GTEx tissue.

To further examine the temporal specificity of the EDev-specific and adult-specific eQTLs, we determined the correlation of their effect sizes independently across 50 (3 EDev-like and 47 GTEx adult) tissues. We observed that the EDev-specific (mean $r^2 = 1.2 \times 10^{-3}$) and adult-specific eQTLs (mean $r^2 = 6.9 \times 10^{-3}$) had distinct effect sizes in EDev-like and adult tissues (Figure 3A). We then examined the shared eQTL correlations between the 50 tissues and observed that they had similar effects (mean $r^2 = 0.28$) in the EDev-like and adult tissues. These observations support our EDev-specific, adult-specific, and shared SNP-eGene pair classifications based on the *mashr* analysis.

To examine the distribution of the EDev-specific SNP-eGene pairs across the three iPSCORE tissues, we annotated the 19,305 primary iPSCORE eQTLs based on the *mashr* assignments (see methods). Of the three EDev-like tissues, CVPCs exhibited the largest fraction of EDev-specific eQTLs (17.7%, n = 855) suggesting that CVPCs are a good model to evaluate temporal regulatory variation in cardiac tissue (Figure 3B). A large fraction of iPSC eQTLs are also EDev specific (10.6%, n = 951) likely because stem cells have a distinct transcriptomic profile and lack an analogous adult tissue. Finally, we compared the effect sizes between iPSCORE EDev-specific and adult-shared eQTLs and found that EDev-specific eQTLs had smaller effects across the three EDev-like tissues (Figure 3C).

Taken together, these results suggest that the majority of regulatory variation is not temporal but rather active both in early development and adulthood. We show that temporal EDev-specific eQTLs are distinct from adult specific and shared eQTLs and they have lower effect sizes than shared eQTLs.

## A large fraction of QTLs is complex and regulate multiple molecular elements

The same QTL signal can be associated with multiple qElements,[2,39,56] therefore we sought to determine the fraction of qElements within each EDev-like tissue that shared primary QTL signals. To identify QTLs with the same genetic signal, we calculated LD between the lead variants of 60,306 primary eQTLs, caQTLs, and haQTLs separately for each tissue. We found 13,604 QTLs (22.6%) that shared a signal ($r^2 \geq 0.8$ and lead variant distance $\leq 100$ kb, see methods) with at least one other QTL (a total of 11,634 QTL pairs) and hence affected multiple qElements; and 46,702 (77.4%) QTLs that were not shared with any other QTL (singletons) and hence affected a single qElement (Table S4).

To generate discrete annotations for complex QTLs associated with multiple qElements, we created three networks by loading the 11,634 QTL pairs as edges for each of the tissues independently and identified 5,672 modules, representing complex QTLs. Across the three tissues, 4,327 (76.3%) of the complex QTLs were associated with only two qElements (Figure 4A; Table S4), while the remaining 1,345 (23.7%) were associated with three or more qElements. The largest complex QTLs

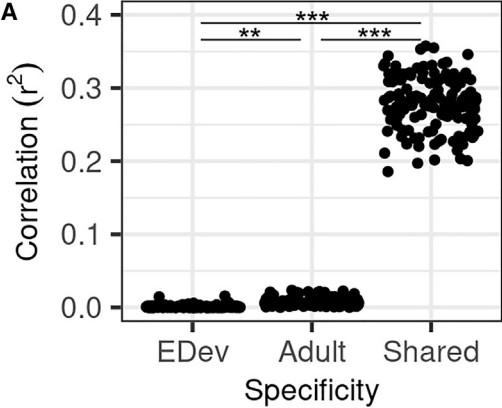

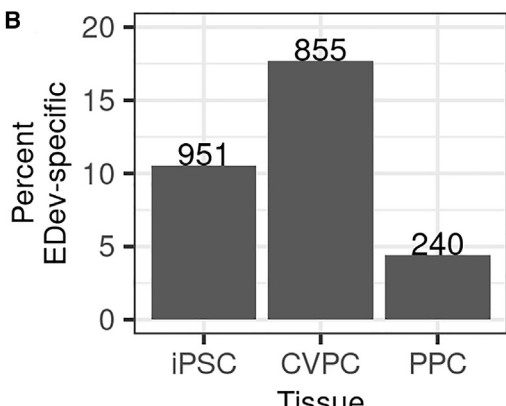

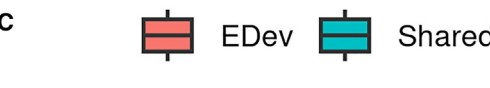

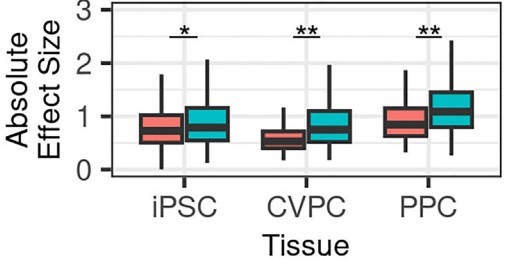

**Figure 3. Characterization of EDev-specific QTLs**

(A) Boxplot showing the correlation of EDev-specific, adult-specific, and shared eQTL effect sizes between the EDev-like and adult GTEx tissues. The x axis is the eQTL specificity, the y axis is the Pearson correlation coefficient ($r^2$) and each point represents the effect size correlation between one of the 3 EDev-like tissues and one of the 47 adult GTEx tissues. Student's t tests were performed to test effect size correlation differences between each group and the p values are reported for each comparison.

(B) Bar plot showing the fraction of iPSCORE EDev-specific eQTLs found in the three tissues. The x axis are the tissues, and the y axis is the fraction of EDev-specific eQTLs. The bars are labeled with the number of EDev-specific eQTLs found in the indicated tissue.

(C) Boxplot showing the differences in effect size between iPSCORE EDev-specific and shared eQTLs by tissue. The x axis is the tissue, the y axis is the

affected between 10 and 15 different qElements and 3.3% complex QTLs ($n$ = 189) affected 5 or more qElements. A total of 2,212 (39.0%) complex QTLs affected at least one eGene and at least one chromatin qElement (caPeak and haPeak, Figures 4B–4D), supporting that caQTLs and haQTLs capture variation affecting regulatory elements that modulate gene expression. The CVPCs had the greatest number of complex QTLs, of which 51.3% (1,560) affected only caPeaks and ha-Peaks (Figure 4C), suggesting that these analyses capture enhancer-acting regulatory variation (Figure 2E) that is missed by eQTL analyses conducted using similarly sized sample sets.

Finally, to assess whether complex QTLs compared with singleton QTLs were enriched for affecting promoters or distal regulatory elements, we calculated the minimum distance between the lead variants and TSS of the nearest expressed gene. Focusing on CVPCs, we showed that the complex QTL lead variants were closer to TSSs compared with singleton QTLs across all three phenotypes (two-sided Mann-Whitney U test, eQTL $p$ = 3.5 × 10$^{-27}$; caQTL $p$ = 2.4 × 10$^{-44}$; haQTL $p$ = 9.2 × 10$^{-13}$; Figure 4E); however, both complex and singleton caQTLs and haQTLs are more distal to promoters compared with complex and singleton eQTLs (two-sided Mann-Whitney U test $p$ = 3.3 × 10$^{-133}$).

In summary, we identified 5,672 complex QTLs that affected 23.7% of qElements. Notably, we found that nearly half of the complex QTLs exclusively affect caPeaks and haPeaks, underscoring the importance of these QTLs in capturing regulatory variations missed by eQTL analyses. In addition, our findings show that, compared with singleton QTLs, complex QTLs are closer to promoters; and also suggest that complex and singleton eQTLs are closer to promoters than either complex or singleton caQTLs and haQTLs.

### Colocalization of iPSCORE multiomic QTLs with GWAS trait and disease loci

To examine the impact of including caQTLs and haQTLs on the annotation rate of GWAS loci, we performed Bayesian colocalization[25] between iPSCORE QTLs (Figure S10) and 5,192 independent GWAS loci from 15 traits associated with early development, longevity, cardio-metabolism, or diabetes that were enriched in the peaks from the three EDev-like tissues (Figures 5A and S11). Given that complex QTLs influence multiple qElements, we randomly chose one qElement per complex and assigned its corresponding QTL to represent the complex QTL for GWAS colocalization (Table S5; see methods).

In total, 10.4% of the GWAS loci ($n$ = 540) across the 15 traits colocalized (PP.H4 ≥ 80%, GWAS $p$ ≤ 5 × 10$^{-8}$, QTL $p$ ≤ 5 × 10$^{-5}$, causal SNP PP ≥ 1%) with 863 EDev-like QTLs, including 699 singleton QTLs and 164 representative complex QTLs (Table S5; see methods). Of the 540 colocalized GWAS loci, 373 (69.0%) colocalized with QTL(s) from only one molecular

absolute effect size of the eQTLs and the boxes are filled by category (EDev, red; shared, turquoise). A two-sided Mann-Whitney U test was performed to test the difference between the groups and the asterisks (*$p$ < 5 × 10$^{-5}$, **$p$ < 5 × 10$^{-20}$) indicate that the tests are significant. The whiskers represent 1.5-times the interquartile range (IQR) and the line in the box represents the median. Outliers are not shown for plot legibility.

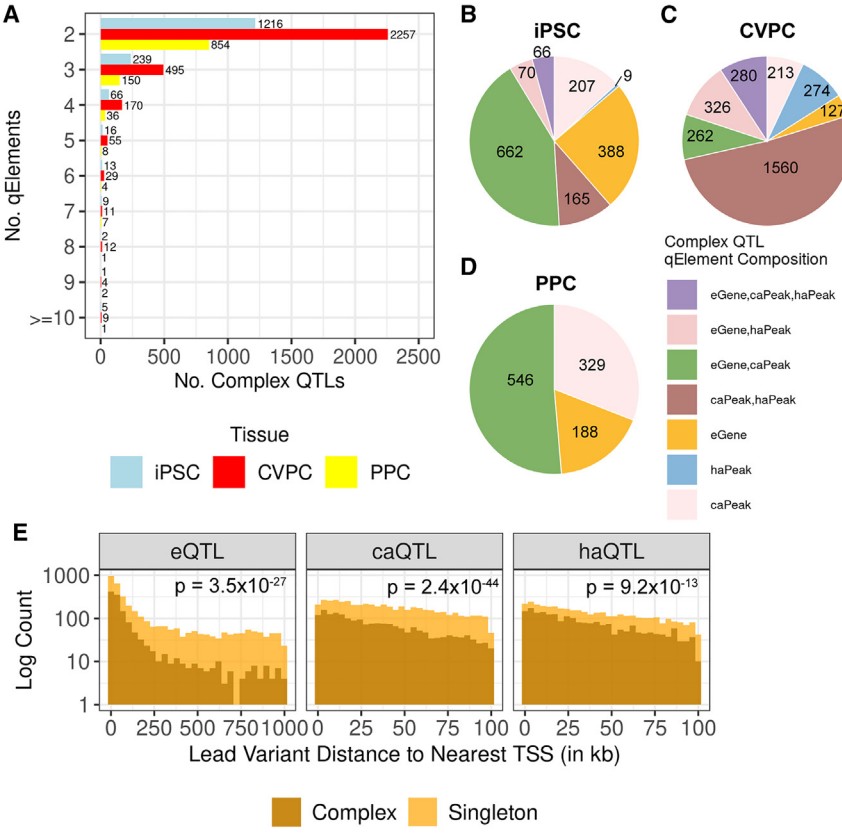

**Figure 4. Characterization of the 5,672 complex QTLs affecting multiple qElements**

(A) Bar plot showing the number of qElements associated with each of the 5,672 complex QTLs. The x axis is the number of complex QTLs, the y axis represents the number of qElements, and the bars are colored by tissue (iPSC, light blue; CVPC, red; and PPC, yellow).

(B–D) Pie charts showing the number of complex QTLs characterized based on their associated molecular qElements (i.e., eQTLs, caQTLs, and haQTLs).

(E) Overlaid histogram showing the different distributions of the distance between the lead variant and the TSS of the nearest expressed gene between complex and singleton QTLs for the three molecular phenotypes in CVPCs. The x axis is the minimum distance between the lead variant and the nearest TSS in kilobases, the y axis is the $\log_{10}$ of the number of QTLs, and the bars show the number of QTLs in each category (complex QTLs, dark orange; and singleton QTLs, light orange). The maximum distance for eQTLs was set to 1 Mb and the maximum distance for chromatin QTLs was set to 100 kb.

were located near the promoter, and, in general, had an intermediate distribution. Interestingly, the most distal colocalized GWAS loci are associated with only chromatin QTLs (caQTL-haQTL, caQTL, haQTL).

phenotype, and 167 (31.0%) colocalized with QTLs from two or more molecular phenotypes across the three EDev-like tissues (Figure 5B). Ventricular rate had the largest proportion of GWAS loci that colocalized with an iPSCORE QTL (*n* = 3; 27%), all of which only colocalized with a chromatin QTL (caQTL or haQTL; Figure 5A), followed by aging and birth weight with 20% of their loci colocalizing with an iPSCORE QTL. Of the 5,192 GWAS loci, 301 (5.8%) colocalized with only caQTLs and/or haQTLs compared with 239 (4.6%) that colocalized with QTLs containing an eQTL (caQTL-eQTL-haQTL, caQTL-eQTL, eQTL-haQTL, eQTL). Therefore, including chromatin QTLs increased the number of GWAS loci annotated with a molecular phenotype by 2.3-fold (Figures 5A and 5B; Table S5).

### Chromatin QTLs colocalize with distal GWAS loci

To evaluate whether the 2.3-fold increase in GWAS colocalization by including chromatin QTLs was driven by their ability to capture more distal regulatory elements, we first calculated the distance between the 5,192 GWAS loci indices and the TSS of the nearest protein-coding gene. We examined the 540 colocalized to determine if GWAS loci associated with QTLs affecting different molecular phenotypes exhibited distinct distributions relative to the nearest TSS (Figure 5C). We observed that QTLs affecting all three molecular phenotypes were the closest to promoters compared with all other groups (two-sided Mann-Whitney U test *p* = 5.5 × $10^{-5}$; Figure 5C). While many loci associated with only eQTLs or eQTLs and one chromatin QTL type (caQTL-eQTL, eQTL-haQTL)

We next sought to determine if GWAS loci that colocalize with QTLs are closer to promoters compared with GWAS loci that do not colocalize with QTLs. Consistent with previous observations,[12] colocalized GWAS loci were closer to the nearest TSS compared with the loci that did not colocalized (Figure 5D; two-sided Mann Whitney U test *p* = 1.2 × $10^{-11}$).

Taken together, these findings show that GWAS loci that do not colocalize with QTLs are further from protein-coding genes and suggest that the integration of chromatin QTLs explain more GWAS loci because they capture distal regulatory elements not captured by eQTLs.

### Complex QTLs have highest GWAS colocalization rates

We examined if GWAS loci were significantly more likely to colocalize with either complex QTLs or singleton QTLs. We binned the 5,672 complex QTLs and 46,702 singletons into 10 categories based on their associated molecular phenotypes (Figures 5E and S12; see methods). We found that complex QTLs affecting all three phenotypes (caQTLs, haQTLs, and eQTLs) were the most enriched, which is consistent with our findings that they are closest to promoters compared with all other QTL categories (Figure 5C). We found that other categories located near promoters (complex QTLs affecting eQTLs and caQTLs, as well as singleton eQTLs; Figure 5E) were also enriched. Complex QTLs associated with haQTLs and caQTLs were the second most enriched category (Figure 5E) and the furthest to promoters (Figure 5C).

Altogether, our results show that complex QTLs affecting all three phenotypes (caQTLs, eQTLs, and haQTLs) have the highest

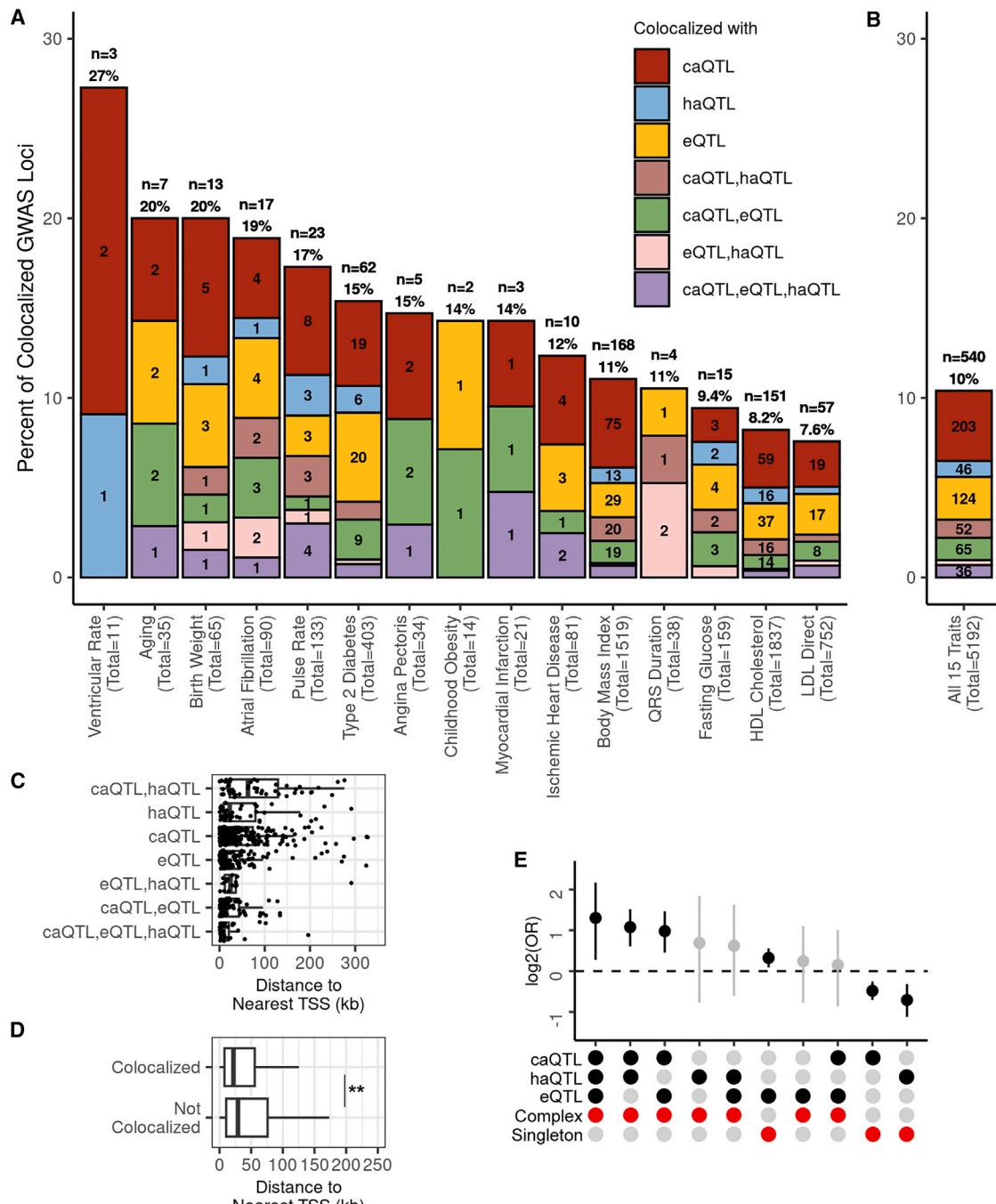

**Figure 5. Chromatin QTLs capture distal GWAS loci missed by eQTLs**

(A and B) Bar plots showing the percent of GWAS loci that are explained by QTLs in the iPSCORE EDev-like tissues for (A) all traits combined and (B) independently. The x axis contains the GWAS trait name, along with the total number of GWAS loci for each trait, and the y axis shows the proportion of GWAS loci that colocalize with iPSCORE QTLs. The bars were colored according to the colocalized QTL types (i.e., caQTL-haQTL-eQTL, eQTL-haQTL, eQTL-caQTL, caQTL-haQTL, eQTL, caQTL, haQTL), and the numbers correspond to the number of GWAS loci that colocalized with the QTL types. At the top of each bar, we indicate the total number and percent of GWAS loci that colocalized with the QTLs.

(C) Boxplot showing the distance to the nearest TSS for colocalized GWAS loci (*n* = 540) by QTL types. The x axis is the distance between the colocalized GWAS loci index and the TSS of the nearest protein-coding gene in kilobases, and the y axis is the combination of QTLs that colocalize with a GWAS locus. The whiskers represent the 1.5× IQR and the line in the box represents the median. For plot legibility, the maximum distance was set to 350 kb.

(D) Boxplot showing the distance to the nearest TSS for GWAS loci by colocalization status. The x axis is the distance between the GWAS loci index and the TSS of the nearest protein-coding gene in kilobases, and the y axis is the GWAS loci colocalization status. The asterisks (**) indicate that there is significantly different

*(legend continued on next page)*

colocalization rate because they capture regulatory variation affecting promoters. In contrast, complex QTLs affecting only chromatin phenotypes (caQTLs and haQTLs) are in distal regions and capture regulatory variation not captured by eQTLs.

### Early developmental-specific eQTLs explain a small fraction of GWAS loci

To test the fetal origins of adult disease hypothesis, we determined the fraction of EDev-specific eQTLs that colocalized with GWAS loci compared with shared QTLs. Of the 239 colocalized GWAS loci associated with at least one eQTL, 5.4% ($n$ = 13) were associated with EDev-specific eQTLs, 75.7% ($n$ = 181) were associated with Shared eQTLs, and 18.8% ($n$ = 45) were associated with eQTLs with low confidence mashr assignments (see methods; Table S4).

Our findings suggest that EDev-specific regulatory variation explains a small fraction of GWAS loci that are not associated with regulatory variation active during adulthood. Future studies aimed at exploring these loci may provide important insights into the role the genetic variation active in this understudied developmental stage plays in complex traits.

### Utility of multiomic QTLs for prioritizing putative causal GWAS variants

Identifying the causal variant(s) in GWAS loci is challenging because the variant with the highest posterior probability of association is often not the causal variant and is in LD with numerous other non-causal variants. To further demonstrate the utility of multiomic QTLs for the identification of putative causal variants, we characterized variants within the 99% credible sets of GWAS-QTL colocalizations based on their overlap with TF motifs. We first aggregated the 99% credible sets of 992 QTLs (699 singleton QTLs, 164 representative complex QTLs, as well as 129 non-representative complex QTLs; see methods) that colocalized with one or more of the 540 GWAS loci; and observed that the distributions of 99% credible set sizes are similar across eQTLs, caQTLs, and haQTLs (Figure 6A), suggesting that each molecular phenotype is similarly capable of identifying causal variants. The posterior probabilities in large credible sets are less reliable, therefore we focused on the 611 high-confidence GWAS-QTL colocalizations with small, high-confidence credible sets (≤25 SNPs) for downstream analyses. We intersected 6,164 SNPs from the 611 high-confidence credible sets with the JASPAR[43] and HOCOMOCO[44] motifs identified from the ATAC-seq peak footprinting analyses (Figure 2A). Across the 15 traits and 3 tissues, 296 GWAS-QTL colocalizations had high-confidence credible sets with 548 motif-overlapping putative causal variants (MOPCVs) (Figure 6B; Table S6). Interestingly, only 14.4% ($n$ = 84) MOPCVs were the predicted top variant from the colocalization (maximum

posterior probability), suggesting that this approach can be leveraged to identify causal variants that are masked by more significant variants in high LD. Of the 548 MOPCVs, 13.1% ($n$ = 72) were present in multiple high-confidence credible sets across traits and tissues, resulting in 365 unique putative causal variants that overlapped at least one TF motif (Table S6).

We sought to prioritize the 548 MOPCVs for experimental validation in future studies, therefore we utilized their epigenomic properties to bin them into three groups based on the evidence for casual association. "High" priority was assigned to MOPCVs if they were present in a caQTL signal that overlapped its associated caPeak, and in the case of complex QTLs (with two or more caPeaks), was assigned to MOPCVs if they were present in a caQTL signal in the complex and overlapped any caPeak in the complex. "Moderate" priority was assigned to MOPCVs that were present in a caQTL signal that overlapped a caPeak associated with a different caQTL signal (e.g., the MOPCV belonged to one caQTL signal and overlapped motifs in a caPeak associated with a different caQTL signal). "Low" priority was assigned to MOPCVs that were present in caQTL signals that overlapped non-caPeaks (e.g., the caQTL signal overlapped an ATAC-seq peak not associated with any caQTL). In total, we annotated 167 high-, 37 moderate-, and 344 low-priority MOPCVs (Table S6). Focusing on these 167 high-priority MOPCVs for biological interpretation would result in a 36.9-fold reduction compared with the 6,164 high-confidence credible set variants.

To substantiate the efficacy of multiomic QTLs in pinpointing putative causal variants underlying human traits, we present in-depth analysis for two high priority MOPCVs with experimental validation[40] or inferred motif disrupting activity.[41]

### *Multiomic PPC complex QTL colocalizes with type 2 diabetes JAZF1 locus*

We identified a high-priority MOPCV (rs1635852; chr7: 28149792:T>C) that was associated with a well-known, experimentally validated[40] type 2 diabetes locus within a *JAZF1* intron.[57] This GWAS locus colocalized with the PPC complex QTL 122 (PP.H4 = 97.2%), which is composed of two caPeaks (ppc_atac_peak_244298 and ppc_atac_peak_244305), as well as the *JAZF1* eGene (Figure 6C). rs849133 was identified as the top variant (PP = 80.7%; Figure 6C); however, MOPCV rs1635852 (PP = 1.4%) has experimentally been shown to be the true causal variant.[40] As previously noted,[40] the T allele of rs1635852 creates a PDX1 binding site, which we identified in ppc_atac_peak_244298 but was predicted by TOBIAS to be unbound. rs1635852 also overlapped 20 other TF motifs, including NKX6-1 (Figure 6D) and HNF1B. Our findings align with previous characterization showing that the rs1635852 T allele in the PDX1 core motif sequence creates a binding site for TFs that repress *JAZF1* expression.[40]

---

distribution (two-sided Mann-Whitney U test $p$ = 1.2 × 10$^{-11}$) between GWAS loci with and without colocalization. The whiskers represent the 1.5× IQR and the line in the box represents the median. For plot legibility, the maximum distance was set to 250 kb.

(E) Plot showing the relative enrichment of GWAS loci colocalization with complex and singleton QTLs. We categorized each complex and singleton QTL based on their associated molecular phenotype(s). Two-sided Fisher's exact tests were performed to test the relative enrichment (odds ratio) of each QTL category for GWAS colocalization compared with all other categories. In the first three rows of the x axis, the black circles indicate the QTL composition categories (i.e., QTLs affecting different combinations of qElements). In the last two rows of the x axis, red circles indicate the QTL category (i.e., complex or singletons). The y axis is the log$_2$(odds ratio) enrichment. Tests that had $p$ < 0.05 were considered significant (colored in black). The whiskers represent the log$_2$ upper and lower 95% confidence intervals.

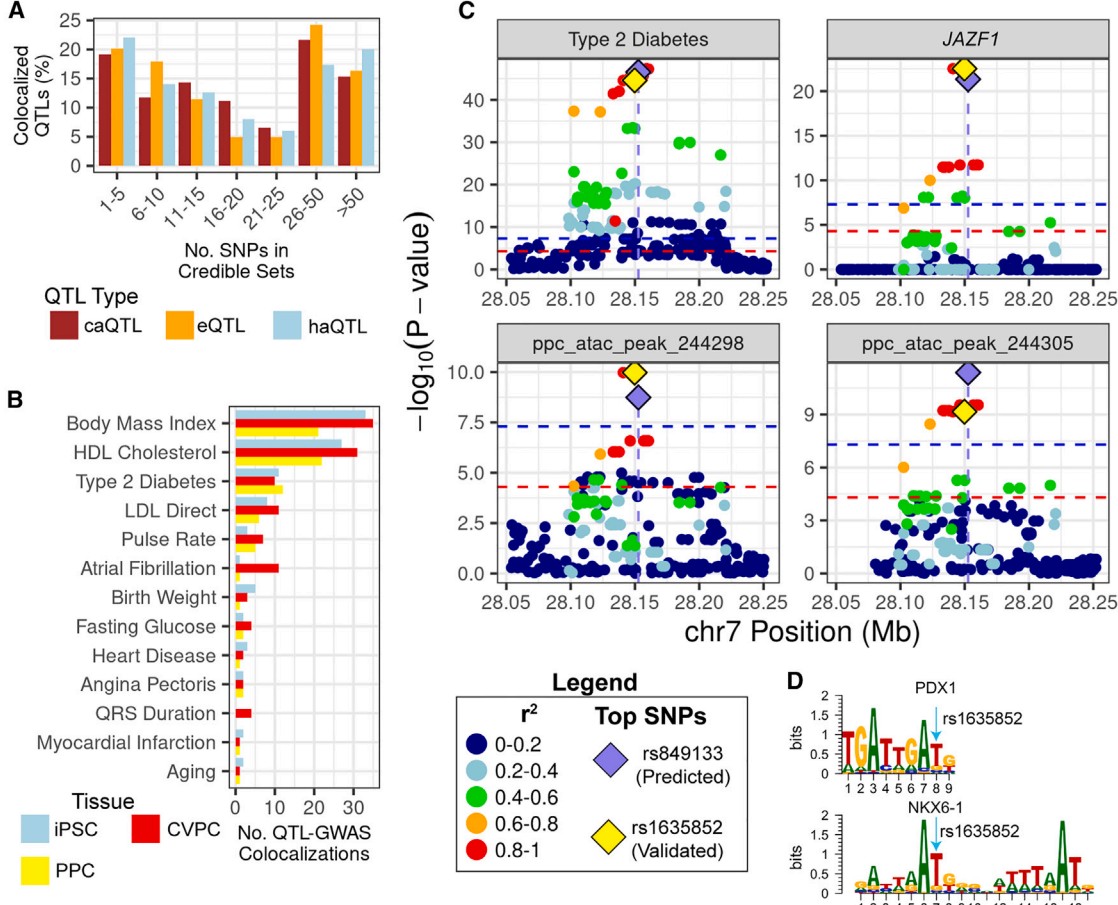

**Figure 6. Multiomic QTLs improve the characterization of causal GWAS variants**

(A) Histogram showing the size of 99% credible sets for the 992 colocalized QTLs (both complex and singleton) across molecular phenotypes. The x axis describes the numbers of variants in the credible set, the y axis is the percent of GWAS-QTL colocalizations, and the bars are colored by QTL molecular phenotype (caQTL, brown; eQTL, orange; and haQTL, light blue).

(B) Bar plot showing the number of the GWAS signals that are associated with a high-confidence credible set SNP that overlaps a TF motif in each of the three tissues. The x axis is the number of GWAS-QTL colocalizations, the y axis divides the 15 GWAS traits into bars colored by tissue (iPSC, light blue; CVPC, red; and PPC, yellow).

(C) A type 2 diabetes signal colocalized with PPC complex QTL 122 containing one eGene, and two caPeaks. The genomic coordinates are on the x axes, and the $-\log_{10}(p$ values) for the associations between the genotype of the tested variants and gene expression, chromatin accessibility or type 2 diabetes are plotted on the y axes. Horizontal lines indicate genome-wide significance thresholds for QTL ($p = 5 \times 10^{-5}$, red) and GWAS ($p = 5 \times 10^{-8}$, blue) for plotting purposes. Each variant was colored according to their LD with the lead fine-mapped variant (purple diamond; rs849133, chr13:28152661:C>T, causal PP = 61.4%) using the 1000 Genomes Phase 3 Panel (Europeans only) as reference. rs1635852 (yellow diamond; chr7:28149792:T>C, causal PP = 1.4%) disrupts TF motifs and is the validated causal variant.[40]

(D) Binding site motifs for PDX1 and NKX6-1 are affected by a high-priority MOPCV (rs1635852; chr7:28149792:T>C) for type 2 diabetes. The light blue arrow indicates which position in the motifs is affected by rs1635852.

## CVPC complex QTL containing four chromatin qElements colocalizes with QRS duration locus

We also identified a high-priority MOPCV (rs9573330; chr13:73944073:G>A) for the QRS duration GWAS locus[41,58,59] in a *KLF12* intron. The *KLF12* QRS duration GWAS locus colocalized with the CVPC complex QTL 274 (PP.H4 = 0.99), which is composed of one caPeak (cvpc_atac_peak_73241) and three haPeaks (cvpc_chip_peak_17303, cvpc_chip_peak_17034, and cvpc_chip_peak_17305; Figure S13A). Through colocalization, the top causal SNP was rs17061696 (chr13:73937854:G>C; PP = 37.4%; Figure S13A); however, we identified a high-priority MOPCV

(rs9573330; PP = 22.0%; Figure S13A) that overlapped 10 unique motifs, including motifs for cardiac markers MEF2A and MEF2C (Figure S13B). These findings recapitulate a previous observation that MOPCV (rs9573330) creates a MEF2A binding site in the *KLF12* intron[41] and demonstrate that the most likely causal variant often does not have the highest posterior probability.

Together, these results illustrate that annotating GWAS loci with complex QTLs and disrupted TF binding sites can help elucidate the underlying molecular mechanisms relevant to development and disease and demonstrates the utility of the iP-SCORE collection of MOPCVs.

## DISCUSSION

After the discovery that 90% of the GWAS variants were in intergenic, non-coding regions, eQTL analyses were pursued to understand the mechanisms by which these variants affect gene expression and disease pathologies.[60] Surprisingly, GTEx eQTL analyses only explain approximately 43% of GWAS loci, indicating that they alone are not sufficient to understand genetic associations with disease. There are several leading hypotheses for the poor overlap between GWAS and GTEx eQTLs, including (1) regulatory variation acts in a context-specific manner (e.g., during fetal development[2–7] and in rarer cell types[3,8–11]), and (2) eQTL and GWAS loci have fundamentally different characteristics.[12] Specifically, eQTLs have strong effect sizes and are biased toward promoter-proximal regulatory regions, while GWAS variants have weak effect sizes and are biased toward distal enhancers. We set out to determine if the GWAS colocalization gaps could be overcome by focusing on tissues representing earlier developmental time points, and by mapping chromatin QTLs that specifically capture variation affecting active regulatory elements, such as promoters and enhancers.

In our study, we evaluated the fetal origins of adult disease hypothesis, which suggests that missing heritability from integrative GWAS-eQTL studies is explained by temporal regulatory variation that is exclusively active during fetal development, and therefore is not captured in adult tissues.[2–4] The EDev-like phenotypic properties of iPSCs and derived tissues provide a powerful model to address this understudied interval of human development. Of the 239 colocalized GWAS loci associated with at least one eQTL, 5.4% ($n$ = 13) were associated with EDev-specific eQTLs, suggesting that a small fraction of GWAS loci are not associated with regulatory variation active during adulthood.

We leveraged the multiomic data from the iPSCORE collection to show that, compared with the traditional eQTL approach, inclusion of chromatin QTLs capture regulatory variation more distal to protein-coding genes and explains 2.3-fold more GWAS loci. Our findings suggest that the fraction of adult disease GWAS loci annotated as harboring identifiable regulatory variation would dramatically increase if studies comparable in sample size with GTEx conducted integrative eQTL, caQTL, and haQTL analyses.

We defined complex QTLs that affected multiple molecular elements and showed that these phenotypically complex loci colocalize with GWAS loci at greater rates than singleton QTLs. Of note, complex QTLs affecting all three phenotypes (caQTLs, haQTLs, and eQTLs) were the closest to promoters compared with all other QTL categories and the most enriched for GWAS loci colocalizaton. On the other hand, complex QTLs associated with haQTLs and caQTLs were the second most enriched category and the furthest to promoters. While power biases may influence the differential GWAS colocalization rates between complex and singleton QTLs, the integration of multiple phenotypes provides additional information that aids in the interpretation of molecular mechanisms underlying disease-associated regulatory variation.

Our work underscores the value of integrating multiomic QTL data with TF motif analysis to enhance the accuracy of putative causal variant identification within GWAS loci. We demonstrate that the iPSCORE collection of MOPCVs refines the set of potential causal variants and provides a framework for understanding the genetic underpinnings of 296 GWAS-QTL colocalizations in 15 complex traits.

### Limitations of the study

Although our study demonstrates the utility of mapping and integrating QTLs affecting multiple molecular phenotypes for the functional characterization and prioritization of putative causal GWAS variants, several limitations warrant consideration. First, the effective sample size in our study is reduced due to the inclusion of related individuals within the iPSCORE cohort. While we accounted for relatedness in our QTL analysis pipeline by incorporating a kinship matrix as a random effects term in the linear mixed model, the nominal number of individuals in the study does not accurately reflect the analytical power. This reduction may influence the robustness of downstream analyses, such as colocalization. Second, variable sequencing depth across the eight molecular datasets posed a challenge. To address this, we calculated the optimal number of PEER factors for each dataset independently and included them as covariates. Our analysis revealed that the PEER factors were highly correlated with sequencing depth-related sample attributes (i.e., number of reads passing filters) and biological variables, such as cellular heterogeneity (i.e., %cTnT in CVPCs and %PDX1-NKX6.1 in PPCs; Figure S14). During optimization (Figure S15), we observed that the inclusion of PEER factors increased the number of QTLs mapped, suggesting that they correct for both technical and biological variability and improve statistical power. Finally, we conducted integrative analyses using datasets with uneven sample sizes, which limited our ability to correct for discovery biases across modalities. To mitigate this constraint, we identified complex QTLs that affect multiple qElements using LD and proximity between their leading variants. This approach depends less on sample size compared with other techniques such as colocalization.

### Conclusion

In conclusion, we identified 70,446 QTLs that affect gene expression, chromatin accessibility, or H3K27 acetylation using 1,261 molecular assays (including WGS) from iPSCs, CVPCs, and PPCs in the iPSCORE collection, making this one of the largest QTL studies conducted using paired multiomic data. By characterizing the properties of the QTLs, we showed that integrating chromatin QTLs can explain a large fraction of GWAS loci that are not explained by eQTLs. Our study provides biological insights into the characteristics of regulatory variation underlying GWAS loci and QTLs and provides a valuable resource for guiding experimental investigation of disease-associated regulatory variation.

### RESOURCE AVAILABILITY

#### Lead contact

Further information and requests for resources should be directed to and will be fulfilled by the lead contact, Kelly A. Frazer (kafrazer@health.ucsd.edu).

### Materials availability

The individual iPSC lines in the iPSCORE resource are available to non-profit organizations through WiCell Research Institute (WiCell: www.wicell.org). Non-profit organizations interested in obtaining the entire iP-SCORE collection and for-profit organizations can contact the corresponding author directly to discuss the availability of iPSC lines as well as differentiated cell types.

### Data and code availability

- Scripts for processing FASTQ files and performing downstream analyses are in GitHub: https://github.com/frazer-lab/iPSCORE_Multi-QTL_Resource. The code and source data are also published in Zenodo: https://doi.org/10.5281/ZENODO.14585175.[61]
- The raw FASTQ sequencing data are accessible via dbGaP phs000924 dbGaP: https://www.ncbi.nlm.nih.gov/projects/gapprev/gap/cgi-bin/study.cgi?study_id=phs000924.v5.p2. WGS data for iPSCORE subjects were downloaded as a VCF file (hg19) from dbGaP phs001325.v5.p1 dbGaP: https://www.ncbi.nlm.nih.gov/projects/gap/cgi-bin/study.cgi?study_id=phs001325.v5.p1. QTL summary statistics from this study are included in dbGaP phs001325.v6.p1 dbGaP: https://www.ncbi.nlm.nih.gov/projects/gap/cgi-bin/study.cgi?study_id=phs001325.v6.p1. For additional information about iPSCORE and data availability, please visit https://frazerlab.ucsd.edu. GWAS summary statistics were obtained from the Pan UK BioBank resource (https://pan.ukbb.broadinstitute.org/),[62,63] the MAGIC (Meta-Analyses of Glucose and Insulin-related traits) Consortium[64] (https://magicinvestigators.org/downloads/), the Early Growth Genetics (EGG) Consortium (http://egg-consortium.org/),[65–67] the DIAMANTE Consortium[68] (https://diagram-consortium.org/downloads.html), and a multivariate longevity/aging study.[69] The processed data generated for this study, including QTL summary statistics, peak coordinates, count matrices, TF predictions, mashr results, and GWAS-QTL colocalization results can be found on Figshare: https://plus.figshare.com/collections/Multiomic_iPSCORE_QTLs/7553361.

### CONSORTIA

Members of the iPSCORE Consortium are Angelo D. Arias, Timothy D. Arthur, Paola Benaglio, W. Travis Berggren, Juan Carlos Izpisua Belmonte, Victor Borja, Megan Cook, Matteo D'Antonio, Agnieszka D'Antonio-Chronowska, Christopher DeBoever, Kenneth E. Diffenderfer, Margaret K.R. Donovan, KathyJean Farnam, Kelly A. Frazer, Kyohei Fujita, Melvin Garcia, Olivier Harismendy, Benjamin A. Henson, David Jakubosky, Kristen Jepsen, Isaac Joshua, He Li, Hiroko Matsui, Angelina McCarron, Naoki Nariai, Jennifer P. Nguyen, Daniel T. O'Connor, Jonathan Okubo, Athanasia D. Panopoulous, Fengwen Rao, Joaquin Reyna, Lana Ribeiro Aguiar, Bianca M. Salgado, Nayara Silva, Erin N. Smith, Josh Sohmer, Shawn Yost, and William W. Young Greenwald.

### ACKNOWLEDGMENTS

We thank Drs. Chris Wallace, Claudia Giambartolomei, Melissa Gymrek, Kyle Gaulton, and Tiffany Amariuta for helpful discussions about Bayesian colocalization. This work was supported by the National Library Training grant T15LM011271; the National Institute of Diabetes and Digestive and Kidney Disease (NIDDK) F31DK131867, U01DK105541, DP3DK112155, and P30DK063491; the National Heart, Lung, and Blood Institute F31HL158198 and U01HL107442; and the National Human Genome Research Institute RM1HG011558 and R41HG008118. Additional support was also received from a California Institute for Regenerative Medicine grant GC1R-06673-B, NSF-CMMI division award 1728497. This publication includes data generated at the UC San Diego IGM Genomics Center utilizing an Illumina NovaSeq 6000 that was purchased with funding from a National Institutes of Health SIG grant S10OD026929.

### AUTHOR CONTRIBUTIONS

T.D.A., J.P.N., and K.A.F. conceived the study. J.P.N., T.D.A., B.A.H., M.D., and J.J. performed computational analyses. K.A.F., G.M., J.C.I.B., and iP-SCORE Consortium members oversaw the study. A.D.-C., A.D.P., N.S., and iPSCORE Consortium members performed the differentiations and generated molecular data. T.D.A., J.P.N., and K.A.F. prepared the manuscript.

### DECLARATION OF INTERESTS

J.C.I.B. is the Founding Scientist and Director of the San Diego Institute of Science at Altos Labs.

### STAR★METHODS

Detailed methods are provided in the online version of this paper and include the following:

- KEY RESOURCES TABLE
- EXPERIMENTAL MODEL AND STUDY PARTICIPANTS
  - Subject information
  - Molecular data sources
- METHOD DETAILS
  - iPSC generation and differentiation
  - Molecular data generation and sequencing
- QUANTIFICATION AND STATISTICAL ANALYSIS
  - Data Processing
  - Characterization of epigenomic properties
  - Quantitative Trait loci (QTLs) mapping
  - QTL characterization
  - Temporal eQTL annotations
  - Identification of Complex QTL
  - GWAS associations with QTLs

### SUPPLEMENTAL INFORMATION

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

**CellPress**

# STAR★METHODS

## KEY RESOURCES TABLE

| REAGENT or RESOURCE | SOURCE | IDENTIFIER |
|---|---|---|
| **Antibodies** | | |
| Tra-1-81, Alexa Fluor 488 anti-human | Biolegend | Cat# 330710; RRID:AB_2561742 |
| SSEA-4, PE anti-human | Biolegend | Cat# 330406; RRID:AB_1089206 |
| Troponin T, Cardiac Isoform Ab-1, Mouse Monoclonal Antibody | Thermo Scientific | Cat# MS-295-P0; RRID:AB_61807 |
| A11001, Alexa Fluor 488 secondary antibody | Life Technologies | Cat#: A11001; RRID:AB_2534069 |
| PE Mouse anti-PDX1 Clone-658A5 | BD Biosciences | Cat# 562161; RRID:AB_10893589 |
| Alexa Fluor® 647 Mouse anti-NKX6.1 Clone R11-560 | BD Bioscience | Cat# 563338; RRID:AB_2738144 |
| PE Mouse anti-IgG1 κ R-PE Clone MOPC-21 | BD Biosciences | Cat# 559320; RRID:AB_397218 |
| Alexa Fluor® 647 Mouse anti IgG1 κ Isotype Clone MOPC-21 | BD Biosciences | Cat# 557732; RRID:AB_396840 |
| H3K27ac antibody | Abcam | Cat# ab4729; RRID:AB_2118291 |
| **Bacterial and virus strains** | | |
| Cytotune Sendai virus | Life Technologies | Cat#: A1378001 |
| **Chemicals, peptides, and recombinant proteins** | | |
| FBS | Invitrogen | Cat# FB-02 |
| DMEM | Invitrogen | Cat# 11330-057 |
| TrypLE | Life Technologies | Cat# 12604013 |
| Matrigel | BD Corning | Cat# 354230 |
| mTeSR1 medium | Stem Cell Technologies | Cat# 85850 |
| Versene | Lonza | Cat# 17-711E |
| Accutase | Innovative Cell Technologies Inc. | Cat# AT 104 |
| RPMI 1640 | Gibco-life Technologies | Cat# 11875119 |
| Penicillin – Streptomycin | Gibco/Life Technologies | Cat# 15140122 |
| B-27 Minus Insulin | Gibco/Life Technologies | Cat# A1895601 |
| IWP-2 | Tocris | Cat# 3533 |
| B-27 Supplement 50X | Gibco/Life Technologies | Cat# 17504044 |
| PBS without Ca2+ and Mg2+ | Gibco/Life Technologies | Cat# 14190250 |
| RPMI 1640 no glucose | Gibco/Life Technologies | Cat# 11879020 |
| Non-Essential Amino Acids | Gibco/Life Technologies | Cat# 11140050 |
| Sodium L-Lactate | Sigma | Cat# 71718-10G |
| 10 μM Y-27632 ROCK Inhibitor | Selleckchem | Cat# S1049 |
| Dispase II | ThermoFisher Scientific | Cat# 17105041 |
| AMPure XP DNA beads | Beckman Coulter | Prod# A63882 |
| SDS Lysis Buffer | Sigma | Prod# 20163 |
| Protein G Dynabeads | Thermo Scientific | Cat# 10003D |
| Protein A Dynabeads | Thermo Scientific | Cat# 10001D |
| RNAse | Sigma | Prod # 70856 |
| Proteinase K Solution 20 mg/mL | ThermoFisher Scientific | Cat# 25530-049 |
| L-Glutamine | Gibco/Life Technologies | Cat# 25030081 |
| HEPES | Gibco/Life Technologies | Cat# 15630080 |
| **Critical commercial assays** | | |
| STEMdiffTM Pancreatic Progenitor Kit | StemCell Technologies | Cat#05120 |

*(Continued on next page)*

*Continued*

| REAGENT or RESOURCE | SOURCE | IDENTIFIER |
|---|---|---|
| Fixation/Permeabilized Solution Kit with BD GolgiStop™ | BD Biosciences | Cat#554715 |
| DNEasy Blood & Tissue Kit | QIAGEN | Cat#69504 |
| TruSeq Nano DNA HT kit | Illumina | Cat#20015965 |
| Quant-iT | Life Technologies | Cat# Q33232 |
| AllPrep DNA/RNA Mini Kit | QIAGEN | Cat# 80204 |
| TruSeq stranded mRNA kit | Illumina | Cat#20020595 |
| Quick-RNA MiniPrep Kit | Zymo Research | Cat#R1054 |
| Nextera DNA Library Preparation Kit | Illumina | Cat# FC-131-1096 |
| NEBNext® High-Fidelity 2X PCR Master Mix | NEB | Cat# M0541L |
| Qubit | Thermo Scientific | Cat# Q33230 |
| KAPA Hyper Prep Kit | KAPA Biosystems | Cat# 07962363001 |
| KAPA Real Time Library Amplification Kit | KAPA Biosystems | Cat: 07958951001 |
| **Deposited data** | | |
| iPSCORE WGS Raw data and analysis | dbGaP | dbGaP: phs001325.v6.p1 |
| iPSCORE RNA-Seq, ATAC-Seq, and ChIP-Seq Raw data | dbGaP | dbGaP: phs000924.v5.p2 |
| Code associated with "Multiomic QTL mapping reveals phenotypic complexity of GWAS loci and prioritizes putative causal variants." | This paper[61] | Zenodo: https://doi.org/10.5281/ZENODO.14585175 |
| UCSC hg38 reference genome | UCSC[70] | https://hgdownload.soe.ucsc.edu/goldenPath/hg38/bigZips/; RRID:SCR_005780 |
| Gencode version 44 hg38 reference genome | Frankish et al.[71] | RRID:SCR_014966 |
| 1000 Genomes Project EUR population | The 1000 Genomes Project Consortium[72] | broad-alkesgroup-public-requester-pays/LDSCORE/GRCh38/plink_files.tgz; RRID:SCR_008801 |
| EpiMap Repository | EpiMap Repository[73] | https://compbio.mit.edu/epimap |
| GTEx Consortium | GTEx Consortium[1] | https://console.cloud.google.com/storage/browser/gtex-resources; RRID:SCR_013042; RRID:SCR_001618 |
| Pan UKBB (UK Biobank) | Pan UKBB[62,63] | https://pan.ukbb.broadinstitute.org/downloads/index.html; RRID:SCR_012815 |
| Early Growth Genetic Consortium | EGG: Early Growth Genetic Consortium[65–67] | http://egg-consortium.org/ |
| Meta-Analyses of Glucose and Insulin-related Traits Consortium | Chen et al.[64] | http://magicinvestigators.org/downloads/ |
| DIAGRAM Consortium | Mahajan et al.[68] | https://diagram-consortium.org; RRID:SCR_015675 |
| Aging/Multivariate Longevity Summary Statistics | Edinburgh Data Share[69] | https://datashare.ed.ac.uk/handle/10283/3599 |
| JASPAR (2020 version) | Fornes et al.[43] | https://jaspar.elixir.no/; RRID:SCR_003030 |
| HOCOMOCO v11 | Kulakovskiy et al.[44] | https://hocomoco11.autosome.org/; RRID:SCR_005409 |
| **Experimental models: Cell lines** | | |
| iPSCORE human iPSC lines | WiCell | www.wicell.org |
| **Software and algorithms** | | |
| FlowJo version 10.2 & version 10.4 | BD Biosciences | https://www.bdbiosciences.com/en-us/products/software/flowjo-v10-software; RRID:SCR_008520 |
| CrossMap | Zhao et al.[74] | https://anaconda.org/bioconda/crossmap; RRID:SCR_001173 |
| STAR 2.7.10b | Dobin et al.[75] | https://github.com/alexdobin/STAR; RRID:SCR_004463 |

*(Continued on next page)*

**Continued**

| REAGENT or RESOURCE | SOURCE | IDENTIFIER |
|---|---|---|
| Picard v3.1.0 | | https://github.com/broadinstitute/picard; RRID:SCR_006525 |
| RSEM v1.3.3 | Li et al.[76] | https://github.com/deweylab/RSEM; RRID:SCR_000262 |
| Samtools v1.17 | Danecek et al.[77] | https://github.com/samtools/samtools; RRID:SCR_002105 |
| edgeR v3.38.4 | Robinson et al.[78] | https://bioconductor.org/packages/release/bioc/html/edgeR.html; RRID:SCR_012802 |
| ENCODE ATAC-seq pipeline | ENCODE Project Consortium[30] | https://github.com/ENCODE-DCC/atac-seq-pipeline; RRID:SCR_023100 |
| ENCODE ChIP-seq pipeline | ENCODE Project Consortium[30] | https://github.com/ENCODE-DCC/chip-seq-pipeline2; RRID:SCR_021323 |
| BWA MEM | Li et al.[79] | https://bio-bwa.sourceforge.net/bwa.shtml; RRID:SCR_010910 |
| Bedtools | Quinlan et al.[80] | https://bedtools.readthedocs.io/en/latest/; RRID:SCR_006646 |
| featureCounts v2.0.6 | Liao et al.[81] | RRID:SCR_012919 |
| bcftools | Li and Dewey[76] | RRID:SCR_005227 |
| ChIPseeker version 1.26.2 | Yu et al.[82] | RRID:SCR_021322 |
| TOBIAS | Bentsen et al.[42] | https://github.com/loosolab/TOBIAS |
| plink 1.90b6.21 | Purcell et al.[83] | https://www.cog-genomics.org/plink/; RRID:SCR_001757 |
| R v4.2.1 | The R Project for Statistical Computing | RRID:SCR_001905 |
| limix v3.0.4 | Limix software | https://github.com/limix/limix |
| eigenMT | Davis et al.[84] | https://github.com/joed3/eigenMT |
| igraph | Csardi et al.[85,86] | RRID:SCR_019225 |
| mashr | Urbut et al.[55] | https://github.com/stephenslab/mashr |
| tabix | Danecek et al.[77] | https://github.com/samtools/htslib/blob/develop/tabix.c |
| LD Score Regression v1.0.1 | Bulik-Sullivan et al.[87] | https://github.com/bulik/ldsc |
| MACS2 | Zhang et al.[88] | https://hbctraining.github.io/Intro-to-ChIPseq/lessons/05_peak_calling_macs.html |

## EXPERIMENTAL MODEL AND STUDY PARTICIPANTS

### Subject information

The iPSCORE collection consists of whole genome sequences (WGS) for 273 iPSCORE subjects, 238 iPSCs derived from 221 of these individuals (Figure S1; Table S1) as well as iPSC-derived cell types (cardiovascular progenitor cells [CVPC] and pancreatic progenitor cells [PPC]) with RNA-seq, ATAC-seq and H3K27 acetylation ChIP-seq. Of the 221 individuals with iPSCs, 141 belong to 40 families composed of two or more subjects (range: 2–14 subjects) and 80 are genetically unrelated (some individuals were in the same family but only related by marriage).[13] Each subject was assigned a Universal Unique Identifier (UUID) and an iPSCORE_ID (i.e., iPSCORE_4_1) which designates family (4) and individual number (1). Sex and age were recorded at the time of enrollment, with molecular data analyzed from 123 females and 98 males, aged 9 to 88 years. We previously estimated the ancestry of each subject by comparing their genomes to those of individuals in the 1000 Genomes Project (KGP).[13] Of the 221 iPSC donors, 170 are of European descent, 34 are of East Asian descent, 7 are admixed Americans, 6 are of South Asian descent, and 6 are of African descent. Other subject information can be found in Table S1 and Figure S1. Recruitment of individuals was approved by the Institutional Review Boards of the University of California, San Diego, and The Salk Institute (project no. 110776ZF).

### Molecular data sources

We used the following datasets from the iPSCORE resource.

(1) 50X WGS (Illumina; 150 bp paired-end) generated from the blood or skin fibroblasts of the 273 iPSCORE subjects.[14].
(2) RNA-seq data (Table S2) from:
- 220 iPSC lines from 220[∓] individuals[14]
- 178 CVPCs derived from 147 iPSC lines from 137 individuals[3,21,22]
- 107 PPCs derived from 106 iPSC lines from 106 individuals[2]
(3) ATAC-seq data (Table S2) from:
- 142 iPSC lines from 129 individuals[19]
- 140 CVPCs derived from 132 iPSC lines from 124 individuals *
- 109 PPCs derived from 108 iPSC lines from 108 individuals *
(4) H3K27ac ChIP-seq data (Table S2) from:
- 43 iPSC lines from 38 individuals *
- 101 CVPCs derived from 97 iPSC lines from 96 individuals *

*The datasets previously published are referenced while the four datasets released here are indicated by an asterisk. In addition to the samples in these four new datasets, 7 of the 220 iPSC RNA-seq samples were not previously published.

[∓]There are 221 subjects in the study and 220 iPSC RNA-seq samples from 220 subjects. One subject had CVPC and PPC RNA--seq samples but not an iPSC RNA-seq sample.

## METHOD DETAILS

### iPSC generation and differentiation
#### iPSC generation

Generation of the 238 iPSC lines has previously been described in detail.[13] Briefly, cultures of primary dermal fibroblast cells were generated from a punch biopsy tissue, expanded for approximately 3 passages, and cryopreserved. In batch, the fibroblasts were thawed and plated at a density of 2.5x10^5 cells/well of 6-well plate and infected with the Cytotune Sendai virus (Life Technologies) per manufacturer's protocol to initiate reprogramming. The Sendai-infected cells were maintained with 10% FBS/DMEM (Invitrogen) for Days 4–7 until the cells recovered and repopulated the well. These cells were then enzymatically dissociated using TrypLE (Life Technologies) and seeded onto a 10-cm dish pre-coated with mitotically inactive mouse embryonic fibroblasts (MEFs) at a density of 5x10^5 cells/dish and maintained with hESC medium, as previously described. Emerging iPSC colonies were manually picked after Day 21 and maintained on Matrigel (BD Corning) with mTeSR1 medium (Stem Cell Technologies). From each individual, multiple independently established iPSC clones (i.e., referred to as lines) were derived, cultured typically to passage 12 (P12), and then cryopreserved. Sendai virus clearance typically occurred at or before P9 and was not detected in the iPSC lines at the P12 stage of cryopreservation. A subset of the iPSC lines was evaluated by flow cytometry for expression of two pluripotent markers: Tra-1-81 (Alexa Fluor 488 anti-human, Biolegend) and SSEA-4 (PE anti-human, Biolegend). Pluripotency was also examined using PluriTest-RNAseq.[89]

Harvesting of material for molecular assays.

(1) At P12, for 220 iPSC lines from 220 individuals, pellets were collected and frozen in RTL plus buffer (Qiagen) for the RNA-seq assay.
(2) iPSC nuclear pellets for the ATAC-seq and H3K27ac ChIP-seq assays were collected at D0 of the CVPC differentiation protocol (see below: **CVPC differentiation**).

#### CVPC differentiation

As previously described in detail,[90] to generate CVPCs, we used a small molecule cardiac differentiation protocol.[91] The 25-day differentiation protocol consisted of four phases.

(1) *Expansion of iPSC*: One vial of each iPSC line was thawed into mTeSR1 medium containing 10 μM ROCK Inhibitor (Sigma) and plated on one well of a 6-well plate coated overnight with matrigel. During the expansion phase, cells were cultured in mTeSR. The iPSCs were passaged using Versene (Lonza) from one well into three wells of a 6-well plate. Next, the iPSCs were passaged using Versene onto three 10-cm dishes at 2.5x10^4 per cm^2 density. The iPSCs monolayer was plated onto three T150 flasks at the density of 3.7x10^4 per cm^2 using Accutase (Innovative Cell Technologies Inc.). iPSCs were at passage 22.7 ± 4.8 (range 17–44) at the monolayer stage (i.e., initiation of differentiation). When iPSC lines were visually estimated to be at 80% confluency, they were passaged in mTeSR1 medium containing 5 μM ROCK inhibitor.
(2) *Differentiation:* After reaching 80% confluency, differentiation (D0) was initiated with the addition of the RPMI 1640 medium (Gibco-life technologies) with Penicillin-Streptomycin (Gibco/Life Technologies) and B-27 Minus Insulin (Gibco/Life Technologies) (hereafter referred to as RPMI Minus) supplemented with 12μM CHIR-99021. After 24h of exposure to CHIR-99021 (D1), medium was changed to RPMI Minus. On D3, medium was changed to 1:1 mix of spent and fresh RPMI Minus, supplemented with 7.5μM IWP-2 (Tocris). On D5, after 48h of exposure to IWP-2, the medium was changed to RPMI Minus. On D7, medium was changed to RPMI 1640 with Penicillin-Streptomycin (Gibco/Life 22 Technologies) and B-27 Supplement 50X (hereafter referred to as RPMI Plus) (Gibco/Life Technologies). Between D7 and D13, RPMI Plus medium was changed every 48h.

For CVPCs with a sufficient proportion of cardiomyocytes, the first beating cells were usually observed between D7 and D9, with some as early as D7 (immediately after the media change). Robust beating was usually observed between D8 and D11.

(3) *Purification:* Since fetal cardiomyocytes have a higher capacity to use lactate as a primary energy source than other cell types,[92,93] we incorporated lactate metabolic selection for five days to increase CVPC purity.[94] On D15, the cells were collected from the flask using Accutase and plated onto fresh T150 flasks at confluency 1-1.3x10^6 per cm². On D16, cells were washed with PBS without $Ca^{2+}$ and $Mg^{2+}$ (Gibco/Life Technologies), and medium was changed to RPMI 1640 with no glucose (Gibco/Life Technologies) supplemented with Non-Essential Amino Acids (Gibco/Life Technologies), L-Glutamine (Gibco/Life Technologies), Penicillin-Streptomycin 10,000U (Gibco/Life Technologies) and 4mM Sodium L-Lactate (Sigma) in 1M HEPES (Gibco/Life Technologies). Medium supplemented with lactate was changed on D17 and D19. During the lactate selection, CVPCs were beating robustly less than 16 h after reseeding.

(4) *Recovery:* After metabolic selection, CVPCs were maintained in cell culture for five days. On D21, cells were washed with PBS, and the medium was changed to RPMI Plus. We changed the media on D23 using RPMI Plus. To evaluate the efficiency of CVPC differentiation, we performed flow cytometry on D25 with the cardiac marker Troponin T (cTnT, TNNT2). Specifically, 5x10^5 CVPCs were permeabilized and blocked in 0.5% BSA, 0.2% TX-100 and 5% goat serum in PBS for 30 min at room temperature. Cells were stained with Troponin T, Cardiac Isoform Ab-1, Mouse Monoclonal Antibody (Thermo Scientific, MS-295-P0) at 4°C for 45 min, followed by Alexa Fluor 488 secondary antibody (Life Technologies, A11001). Stained cells were acquired using BD FACSCanto II system (BD Biosciences) and the fraction of cTnT-positive cells were calculated using FlowJo software version 10.2.[21]

Harvesting of material for molecular assays.

(1) At D0 of the CVPC differentiation, 142 iPSC lines from 129 individuals were collected and frozen as nuclear pellets for the ATAC-seq assay**.

(2) At D0 of the CVPC differentiation, 43 iPSC lines from 41 individuals were collected and frozen as nuclear pellets for the H3K27ac ChIP-seq assay**.

(3) At D25, 178 CVPCs derived from 147 iPSC clones from 137 individuals, pellets were collected and frozen in RTL plus buffer (Qiagen) for the RNA-seq assay.

(4) At D25, 140 CVPCs derived from 132 iPSC clones from 124 individuals were collected and frozen as nuclear pellets for the ATAC-seq assay.

(5) At D25, 101 CVPCs derived from 97 iPSC clones from 96 individuals were collected and cross-linked for the H3K27ac ChIP-seq assay.

Asterisks (**) indicate that the iPSC ATAC-seq and H3K27ac ChIP-seq data were generated for iPSC lines treated with ROCK inhibitor before initiating CVPC differentiation, whereas the RNA-seq data were generated on ROCK inhibitor-naïve iPSC lines.

### PPC differentiation

As previously described,[2] the iPSC lines were differentiated into PPCs using the STEMdiff Pancreatic Progenitor Kit (StemCell Technologies, Catalog #05120) protocol with minor modifications. Briefly, iPSC lines were thawed into mTeSR1 medium containing 10 μM Y-27632 ROCK Inhibitor (Selleckchem) and plated onto one well of a 6-well plate coated with Matrigel. iPSCs were grown until they reached 80% confluency and then passaged using 2 mg/ml solution of Dispase II (ThermoFisher Scientific) onto three wells of a 6-well plate (ratio 1:3). To expand the iPSC cells for differentiation, iPSCs were passaged a second time onto six wells of a 6-well plate (ratio 1:2). When the iPSCs reached 80% confluency, cells were dissociated into single cells using Accutase (Innovative Cell Technologies Inc.) and resuspended at a concentration of 1.85x10^6 cells/ml in mTeSR medium containing 10 μM Y-27632 ROCK inhibitor. Cells were then plated onto six wells of a 6-well plate and grown for approximately 16–20 h to achieve a uniform monolayer of 90–95% confluence (3.7x10^6 cells/well; about 3.9x10^5 cells/cm²). Differentiation of the iPSC monolayers was initiated by replacing the mTeSR medium with STEMdiff Stage Endoderm Basal medium supplemented with Supplement MR and Supplement CJ (2 mL/well) (Day 1, D1). The following media changes were performed every 24 h after initiation of differentiation (2 mL/well). On D2 and D3, the medium was changed to fresh STEMdiff Stage Endoderm Basal medium supplemented with Supplement CJ. On D4, the medium was changed to STEMdiff Pancreatic Stage 2–4 Basal medium supplemented with Supplement 2A and Supplement 2B. On D5 and D6, the medium was changed to STEMdiff Pancreatic Stage 2–4 Basal medium supplemented with Supplement 2B. From D7 to D9, the medium was changed to STEMdiff Pancreatic Stage 2–4 Basal medium supplemented with Supplement 3. From D10 to D14, the medium was changed to STEMdiff Pancreatic Stage 2–4 Basal medium supplemented with Supplement 4. On D15, to evaluate the efficiency of PPC differentiation, we performed flow cytometry on two pancreatic precursor markers, PDX1 and NKX6-1. Specifically, at least 2x10^6 cells were fixed and permeabilized using the Fixation/Permeabilized Solution Kit with BD GolgiStop TM (BD Biosciences) following the manufacturer's recommendations. Cells were resuspended in 1x BD Perm/Wash TM Buffer at a concentration of 1x10^7 cells/ml. For each flow cytometry staining, 2.5x10^5 cells were stained for 75 min at room temperature with PE Mouse anti-PDX1 Clone-658A5 (BD Biosciences; 1:10) and Alexa Fluor 647 Mouse anti-NKX6.1 Clone R11-560 (BD Bioscience; 1:10), or with the appropriate class control antibodies: PE Mouse anti-IgG1 κ R-PE Clone MOPC-21 (BD Biosciences) and Alexa Fluor 647 Mouse anti IgG1 κ Isotype Clone MOPC-21 (BD Biosciences). Stained cells were washed three times, resuspended

in PBS containing 1% BSA and 1% formaldehyde, and immediately analyzed using FACS Canto II flow cytometer (BD Biosciences). The fraction of PDX1-and NKX6-1-positive cells were calculated using FlowJo software version 10.4[2].

Harvesting of material for molecular assays.

(1) At D15, 107 PPCs derived from 106 iPSC clones from 106 individuals, pellets were collected and frozen in RTL plus buffer (Qiagen) for the RNA-seq assay.
(2) At D15, 109 PPCs derived from 108 iPSC clones from 108 individuals were collected and frozen as nuclear pellets for the ATAC-seq assay.
(3) H3K27ac ChIP-seq assay was not performed for the PPCs, however, we have frozen samples, and this dataset could be added in the future.

### Molecular data generation and sequencing
#### *Whole Genome Sequencing*
As previously described,[13,14] we generated whole genome sequences (WGS) for 273 iPSCORE subjects, though only 221 had their fibroblasts reprogrammed into 238 iPSC lines. Genomic DNA was isolated from whole blood (or in 19 cases directly from the fibroblasts) using DNEasy Blood & Tissue Kit (Qiagen), quantified, normalized, and sheared with a Covaris LE220 instrument. The samples were normalized to 1 µg and submitted to Human Longevity (HLI) for whole genome sequencing. DNA libraries were prepared (TruSeq Nano DNA HT kit, Illumina), characterized with regards to size (LabChip DX Touch, PerkinElmer) and concentration (Quant-iT, Life Technologies), normalized to 2-3.5nM, combined into 6-sample pools, clustered and sequenced to ~50X depth on the Illumina HiSeqX (150 bp paired-end).

#### *RNA-seq*
As previously described in detail,[14] for the iPSCs, total RNA was isolated from cell lysates using AllPrep DNA/RNA Mini Kit (QIAGEN). RNA quality was assessed based on RNA integrity number (RIN) using an Agilent Bioanalyzer, and libraries were prepared using the Illumina TruSeq stranded mRNA kit and sequenced with 100 bp paired-end reads on an Illumina HiSeq2500 (an average of 22 million read pairs/per sample).

As previously described in detail,[3,21,22] for the CVPCs, total RNA was isolated from cell lysates using the Quick-RNA MiniPrep Kit (Zymo Research). RNA quality was assessed based on RIN, and libraries were prepared using the Illumina TruSeq stranded mRNA kit and sequenced with either 100 bp paired-end or 150 bp paired-end reads on an Illumina HiSeq4000 (an average of 28 million read pairs/per sample).

As previously described in detail,[2] for the PPCs, total RNA was isolated from cell lysates using the Quick-RNA MiniPrep Kit (Zymo Research), RNA quality was assessed based on RIN, and libraries were prepared using the Illumina TruSeq stranded mRNA kit and sequenced with 100 bp paired-end reads on an Illumina NovaSeq 6000 (an average of 71 million read pairs/sample).

#### *ATAC-seq*
All ATAC-seq samples were processed in the same manner using a modified version of the Buenrostro et al. protocol[29] as previously described.[19] Briefly, frozen nuclear pellets of $2.5 \times 10^4$ iPSC or $1 \times 10^5$ CVPC or PPC cells were thawed on ice and tagmented in total volume of 25µL in permeabilization buffer containing digitonin (10mM Tris-HCl pH 7.5, 10mM NaCl, 3mM MgCl,[2] 0.01% digitonin) and 2.5µL of Tn5 from Nextera DNA Library Preparation Kit (Illumina) for 45-75 min at 37°C in a thermomixer (500 RPM shaking). We included a double size selection step during purification using AMPure XP DNA beads (Beckman Coulter). To eliminate confounding effects due to index hopping, all libraries within a pool were indexed with unique pairs of i7 and i5 barcodes. Libraries were amplified for 12 cycles using NEBNext High-Fidelity 2X PCR Master Mix (NEB) in total volume of 25µL in the presence of 800nM of barcoded primers (400nM each) custom synthesized by Integrated DNA Technologies (IDT) and sequenced with either 100 bp paired-end and 150 bp paired-end reads on an Illumina HiSeq4000 for iPSCs and CVPCs and 150 bp paired-end reads on an Illumina NovaSeq 6000 for PPCs.

#### *H3K27 acetylation ChIP-seq*
All H3K27ac ChIP-seq samples were processed in the same manner. For H3K27ac, $5-15 \times 10^6$ formaldehyde crosslinked cells were lysed and sonicated in 110µL of SDS Lysis Buffer (0.5% SDS, 50mM Tris-HCl pH 8.0, 20mM EDTA, 1x cOmplete Protease Inhibitor Cocktail (Sigma)) using Covaris E220 Focused-ultrasonicators (Covaris) for 14 cycles, 1 min per cycle, duty cycle 5. For each sample, H3K27ac antibody (Abcam ab4729, lot GR00324078) was coupled for 4 h to 40µL of 1:1 mix Protein G and Protein A Dynabeads (Thermo Scientific) and used for overnight chromatin immunoprecipitation in IP buffer (1% Triton X-100, 0.1% DOC, 1x TE buffer, 1x cOmplete Protease Inhibitor Cocktail). Beads with immunoprecipitated chromatin were washed with 150µL of following buffer: four times with RIPA Low Salt Buffer (0.1% SDS, 1% Triton X-100, 2mM EDTA, 20mM Tris-HCl pH 8.0, 300mM NaCl, 0.1% DOC), two times in RIPA High Salt Buffer (0.1% SDS, 1% Triton X-100, 1mM EDTA, 20mM Tris-HCl pH 8.0, 500mM NaCl, 0.1% DOC), twice in LiCl Buffer (250mM LiCl, 0.5% NP-40, 0.5% DOC, 1mM EDTA, 10mM Tris-HCl pH 8.0) and twice in 1X TE buffer (10mM Tris-HCl pH 8.0, 1mM EDTA). Next samples were eluted in 150 µL of Direct Elution Buffer (0.1% SDS, 10mM Tris-HCl pH 8.0, 5mM EDTA) and reverse crosslinked by incubation for 15 min at 65°C with rotation and subsequent incubation with 5 µL RNAse (Sigma) for 1h at 37°C and Proteinase K Solution (20 mg/mL, Thermo Fisher Scientific) for 1h at 55°C. After reverse crosslinking samples were purified with 2X Agencourt AMPure XP DNA beads (Beckman Coulter), eluted in 30 µL of H₂O and Qubit (Thermo Scientific) quantified. Libraries were generated using KAPA Hyper Prep Kit (KAPA Biosystems) and KAPA Real Time Library Amplification Kit

(KAPA Biosystems) following manufacturers manual. Libraries were barcoded using TruSeq RNA Indexes (Illumina), size selected for 300 bp to 500 bp, and sequenced with either 100 bp paired-end or 150 bp paired-end reads on an Illumina HiSeq 4000 (an average of 44 million read pairs/per sample).

## QUANTIFICATION AND STATISTICAL ANALYSIS

### Data Processing

#### Whole Genome Sequencing

We downloaded the VCF in hg19 for 273 iPSCORE individuals[13,14] (see **Molecular data generation and sequencing: Whole Genome Sequencing)** from dbGaP (phs001325.v5.p1), and performed liftOver to hg38 using CrossMap[74] and the hg38 reference genome from UCSC (https://hgdownload.soe.ucsc.edu/goldenPath/hg38/bigZips/).

#### RNA-seq

All RNA-seq samples were processed in a uniform manner (Table S2). Libraries that were sequenced more than once were merged by concatenating the FASTQ files. The reads were aligned onto the hg38 human reference genome downloaded from Gencode version 44[71,95] using STAR 2.7.10b (https://github.com/alexdobin/STAR) with the following parameters: –outSAMattributes All –outSAMun-mapped Within –outFilterMultimapNmax 20 –outFilterMismatchNmax 999 –alignIntronMin 20 –alignIntronMax 1000000 –alignMates-GapMax 1000000. PCR duplicates were marked with Picard (https://github.com/broadinstitute/picard) (v3.1.0) and counted using samtools flagstat[77] (v1.17). Number and percentage of mapped reads were calculated using samtools flagstat[77] (v1.17). Percentage of intergenic and mRNA bases were determined using Picard (v3.1.0) CollectRnaSeqMetrics. Gene TPM expression and read counts were calculated using RSEM[76] (v1.3.3) with gene annotations from Gencode version 44[71,95] (hg38) and the following parameters: –seed 3272015 –estimate-rspd –forward-prob 0 –paired-end. RNA-seq samples were examined for quality using GTEx standards.[1] Specifically, we required that samples met the following metrics: 1) the number of mapped reads >10 million; 2) percent of intergenic bases <30; 3) percent of mRNA bases >70; 4) percent of duplicate reads <30; 5) percent mapped reads >85%; 6) number of reads passing filters >25M, and 7) matched via a sample identity check to the correct subject with PI_HAT >0.90 (Figure S16).

For all downstream analyses, gene expression values were normalized and filtered using the same procedure as GTEx.[1] Specifically, 1) read counts were TMM normalized across all genes using edgeR[78] (v3.38.4) with functions *DGEList*, *calcNormFactors*, and *cpm*; 2) autosomal genes were selected and filtered based on expression thresholds of $\geq$ 0.1 TPM in $\geq$ 20% of samples and $\geq$ 6 reads (unnormalized) in $\geq$ 20% of samples; 3) TMM expression values for each gene were inverse normal transformed across samples using *rank* and *qnorm* in R v4.2.1 and used as input for eQTL analyses. This resulted in 18,720 iPSC, 18,314 CVPC, and 20,738 PPC genes used for eQTL mapping.

#### ATAC-seq

All ATAC-seq samples were processed in a uniform manner using the same procedure as the ENCODE (https://github.com/ENCODE-DCC/atac-seq-pipeline) (Table S2). Illumina adapters were removed from the reads using cutadapt.[96] Reads were aligned using BWA MEM[79] (https://bio-bwa.sourceforge.net/bwa.shtml) onto the hg38 human reference genome from UCSC (https://hgdownload.soe.ucsc.edu/goldenPath/hg38/bigZips/). Multi-mapped reads were randomly assigned using ENCODE's custom script (assign_multimappers.py) (https://github.com/ENCODE-DCC/atac-seq-pipeline). Using samtools,[77] reads that were either unmapped, not in primary alignment, failed Illumina QC metrics, or had an unmapped mate were removed (samtools view -F 1804). Properly paired reads with mapping quality $\geq$ 30 were retained (samtools view -f 2 -q 30). Duplicates were marked by Picard and then removed with samtools. Mitochondrial reads were also excluded from downstream analyses. Filtered BAM files were converted to bed files (bedtools bamtobed) and shifted for Tn5 bias, and then used to call narrow peaks using MACS2[88] with ENCODE default parameters (https://github.com/ENCODE-DCC/atac-seq-pipeline): –shift 75 –extsize 150 -q 0.01 –nomodel -B –SBMR –keep-dup all. Peaks overlapping blacklisted regions were removed. ATAC-seq samples were examined for quality and excluded if they did not pass one of the following metrics: 1) non-redundant fraction (NRF) > 0.9; 2) PCR-bottlenecking coefficient 1 (PBC1) > 0.9; 3) PCR-bottlenecking coefficient 2 (PBC2) > 3; 4) percent of mapped reads >0.95; 5) fraction of reads in peaks (FRIP) > 10; 6) TSS enrichment[97] (TSSE) > 4, and 7) matched via a sample identity check to the correct subject with PI_HAT >0.90 (Figure S17). Across all ATAC-seq samples, the number of read pairs passing filters ranged from 6.4 million to 46.2 million with an average of 31.4 million.

To identify consensus peaks for each tissue, we selected high quality reference samples using the following filters: 1) 25 < FRiP <45; 2) 5 < TSSE <25; and 3) 75,000 < number of peaks <200,000) from unrelated individuals in different families with two or more individuals in the iPSCORE collection (Figure S17). If multiple samples from the same family passed these filters, we selected the sample with the highest TSSE, which resulted in 24 iPSC, 24 PPC, and 23 CVPC reference samples. For each reference sample, we removed short peaks (<150 bp), then concatenated and merged peaks across all the reference samples for each tissue, resulting in 208,581 iPSC, 278,471 CVPC, and 289,980 PPC reference peaks. For each ATAC-seq sample, we used featureCounts[81] (v2.0.6) to count the number of reads in each reference peak.

For all downstream analyses, for each dataset, we first TMM-normalized the reference ATAC-seq peak counts across samples using the *calcNormFactors* and *cpm* functions in in the edgeR package v3.38.4. We then removed ATAC-seq peaks on sex chromosomes or with low accessibility (TMM <1 in at least 20% of the samples), resulting in 172,075 iPSC, 202,941 CVPC, and 193,428 PPC ATAC-seq peaks.

### H3K27 acetylation ChIP-seq

All H3K27ac ChIP-seq data was processed in a uniform manner using the same procedure as the ENCODE (https://github.com/ENCODE-DCC/chip-seq-pipeline2) (Table S2). Illumina adapters were removed from the reads using cutadapt.[96] Reads were aligned using BWA MEM onto the hg38 human reference genome downloaded from UCSC (https://hgdownload.soe.ucsc.edu/goldenPath/hg38/bigZips/). Properly paired reads were retained (samtools view -f 2). Using samtools unmapped and non-uniquely mapped reads were removed (samtools view -F 1804), and PCR duplicates were marked with Picard and removed (samtools -F 1804). BAM files for each sequencing run were merged by library and then used to call narrow peaks using MACS2 with ENCODE default parameters: –shift −75 –extsize 150 -p 0.01 –nomodel -B –SPMR –keep-dup all. H3K27ac ChIP-seq samples were examined for quality and excluded if they did not pass one of the following metrics: 1) > 20 million usable fragments; 2) NRF >0.9; 3) PBC1 > 0.9; 4) PBC2 > 10; and 5) FRIP >1; and 6) matched via a sample identity check to the correct subject with PI_HAT >0.90 (Figure S18). Across all H3K27ac ChIP-seq samples, the number of read pairs passing filters ranged from 10.0M to 64.6M with an average of 27.2M.

To identify CVPC H3K27ac ChIP-seq consensus peaks, we selected 29 high-quality reference samples using the following filters: 1) 5 < FRiP <10; 2) 40M < number of reads passing filters <100M; and 3) 25,000 < number of peaks <75,000) from unrelated individuals in 29 families (Figure S18). If multiple samples from the same family passed these filters, we selected the sample with the highest FRiP. Since there were only 43 iPSC H3K27ac ChIP-seq samples, we did not consider sample quality to establish consensus peaks and selected 28 iPSC reference samples from 28 families. For families that had multiple individuals with iPSC H3K27ac ChIP-seq samples, we selected the sample with the highest FRiP. For each reference sample, we filtered peaks (500 bp > peak length >5000 bp), then concatenated and merged peaks across all the reference samples in the corresponding tissue, resulting in 45,729 iPSC, and 63,811 CVPC reference peaks. For each H3K27ac ChIP-seq sample, we used featureCounts[81] v2.0.6 to count the number of reads in each reference peak.

For all downstream analyses, for each dataset we first TMM-normalized the reference ChIP-seq peak counts across samples using the *calcNormFactors* and *cpm* functions in the edgeR[78] package v3.38.4. We then removed ChIP-seq peaks on sex chromosomes or with low read counts (TMM <1 in at least 20% of the samples), resulting in 44,206 iPSC, and 60,556 CVPC ChIP-seq peaks.

### Sample identity

Sample identity was performed as previously described.[2,3,13,14,19,21] Briefly, genotypes were called from BAM files of each molecular dataset for common variants with minor allele frequency (MAF) > 45% and <55% using bcftools[77] mpileup and call, and then compared to WGS genotypes using plink –genome,[83] which calculates IBD between each pair of samples. Samples that matched the correct subject with PI_HAT >0.90 passed sample identity check.

### Characterization of epigenomic properties

### Chromatin state peak Annotation

We annotated the five ATAC-seq and H3K27ac ChIP-seq peak datasets using the ChIPseeker (version 1.26.2) R package.[82] We followed the tutorial (https://hbctraining.github.io/Intro-to-ChIPseq/lessons/12_functional_analysis.html) and used the UCSC hg38 default parameters for gene annotations. We used the *plotAnnotBar* function to generate Figure S3.

### Comparing Epigenomes Across Tissues

Independent consensus ATAC-seq peaks and H3K27ac ChIP-seq peaks were defined for each tissue and molecular dataset, therefore we could not use them to compare chromatin accessibility or histone acetylation across tissues. To create global consensus ATAC-seq peaks in order to compare all 391 ATAC-seq samples, we concatenated the 172,075 iPSC, 202,941 CVPC, and 193,428 PPC accessible (TMM <1 in at least 20% of the samples) ATAC-seq peaks, then sorted and merged using bedtools merge with parameters: -c 4 -o distinct, which reports the IDs of the peaks that were merged. After merging, there were 348,429 global consensus ATAC-seq peaks from all three tissues. Likewise, we created 83,680 global consensus ChIP-seq peaks by concatenating, sorting, and merging the 44,206 iPSC, and 60,556 CVPC ChIP-seq peaks. We applied featureCounts[81] (v2.0.6; as described in **Data Processing: ATAC-seq**) to count the number of reads in 391 ATAC-seq samples and 143 ChIP-seq samples, using the corresponding global consensus peaks. The counts were TMM-normalized, using edgeR[78] v3.38.4 (as described in **Data Processing: ATAC-seq** and **Data Processing: H3K27 acetylation ChIP-seq**). A PC analysis was performed on the top 2000 most variable global consensus peaks for ATAC-seq and ChIP samples, independently. A UMAP dimensionality reduction was performed on the top 9 PCs for ATAC and the top 10 PCs for ChIP samples (Figure S4A and S4B).

### Tissue-specific and shared ATAC-seq peaks

Using the 348,429 global consensus ATAC-seq peaks calculated above (see **Comparing Epigenomes Across Tissues),** we identified peaks that were only present in one tissue. An ATAC-seq peak from a given tissue that overlapped (≥ 1 bp) at least one ATAC-seq peak from a different tissue was considered "Shared" (Figure S4C).

### ATAC-seq Peak Transcription Factor Predictions

The TOBIAS[42] algorithm leverages the distribution of reads across the genome for a given sample, therefore to profile TF occupancy, we ran TOBIAS to predict binding at 1,147 motifs across ATAC-seq peaks for each tissue, independently. We first merged BAM files for the reference samples used to establish reference peaks for each tissue. We followed the standard workflow in the TOBIAS tutorial (https://github.com/loosolab/TOBIAS). Briefly, for each merged reference BAM file, we applied *ATACorrect* to correct for cut site biases introduced by the Tn5 transposase within the 172,075 iPSC, 202,941 CVPC, and 193,428 PPC ATAC-seq peaks, using the following parameters: –genome hg38.fa (https://hgdownload.soe.ucsc.edu/goldenPath/hg38/bigZips/)

and –blacklist hg38-blacklist.v2.be (https://github.com/Boyle-Lab/Blacklist/blob/master/lists/hg38-blacklist.v2.bed.gz). Next, we calculated footprints scores with *ScoreBigwig*, using the corresponding narrowPeak file for each tissue as input. To identify the predicted transcription factor binding sites, we ran *BINDetect* with 746 motifs from JASPAR[43] (2020 version) and 401 HOCOMOCO[44] v11 TF motifs independently, using the hg38 fasta as the reference genome file. The tables for predicted TFBSs at JASPAR and HOCOMOCO motifs are deposited on Figshare.

### ATAC-seq peak TFBS enrichment

We performed two-sided Fisher's Exact tests to calculate the enrichment of TFBSs in tissue-specific ATAC-seq peaks. For the three sets of tissue-specific ATAC-seq peaks, we calculated the differential TFBS enrichment relative to the corresponding shared ATAC-seq peaks, using peaks with at least one TFBS. For example, of the 66,333 tissue-specific and 105,742 shared ATAC-seq peaks in the iPSCs (Figure S4C), we used 5,109 tissue-specific and 40,633 shared peaks that were bound by at least one TFBS to calculate enrichment, using the shared peaks as background.

### Quantitative Trait loci (QTLs) mapping

#### Whole Genome sequencing variant selection

For all QTL analyses, we used single nucleotide polymorphisms (SNPs) that met the following criteria across the 273 individuals (see **Data Processing: Whole Genome Sequencing**): 1) passed Illumina QC; 2) in Hardy-Weinberg equilibrium ($p > 0.000001$); 3) genotyped in at least 99% of the individuals; and 4) had MAF >0.05. After filtering, 5,536,303 SNPs remained.

#### Kinship matrix

To account for genetic relatedness between samples, we performed LD pruning on the 5,536,303 variants using plink[83] 1.90b6.21 (-indep-pairwise 50 5 0.2). We then used the 323,697 LD-pruned variants to construct a kinship matrix for the 273 iPSCORE individuals using plink[83] 1.90b6.21 (–make-rel square).

#### Genotype principal component analysis

To calculate global ancestry, we performed genotype principal component analysis (PCA) across all 273 individuals in the iPSCORE Collection. First, we intersected the 323,697 LD-pruned variants above with 1000 Genomes[98–100] single nucleotide polymorphisms (SNPs). Then, using plink[83] 1.90b6.21 (–pca-cluster-names AFR EUR AMR EAS SAS –pca), we performed PCA excluding 1000 Genome subjects without super-population information. We determined that the first five genotype PCs for QTL analysis were sufficient and captured the majority of the variability that was due to global ancestry (Figure S1c). The ancestries reported for the 221 subjects in this study (Figure S1a-b), were assigned in a previous study describing the iPSCORE Collection.[13]

#### PEER Factor Calculation

To account for hidden technical and biological confounders that influence gene expression variability, we used Probabilistic Estimation of Expression Residuals[101] (PEER) to estimate a set of latent factors for each tissue (iPSC, CVPC, PPC) and molecular data type (RNA-seq, ATAC-seq, H3K27ac ChIP-seq). We used the top 2,000 most variable genes/peaks to calculate a maximum number of PEER factors that is equivalent to ~25% of the samples (for instance, for the CVPC RNA-seq dataset of 178 samples, we set the maximum number of PEER factors to 50), as recommended by the original developers.[101] As previously described,[2,3] to determine the number of PEER factors to use for QTL discovery, we piloted QTL mapping on a random set of 1,000 genes or 4,000 peaks using varying numbers of PEER factors as covariates (Table S2) and selected the least number of PEERs that resulted in maximum eGene, caPeak, and haPeak discovery (Figure S15). For eQTLs, we used 51, 35, and 22 PEER factors for iPSC, CVPC, and PPC, respectively as covariates. For caQTLs, we used 28, 28, and 20 PEER factors for iPSC, CVPC, and PPC, respectively, as covariates. For haQTLs, we used 8 and 19 PEER factors for iPSC and CVPC, respectively, as covariates. We found that the variance captured by PEER factors was correlated with known biological and technical factors recorded for each sample (Figure S14). In particular, we observed that the top PEER factors across all the molecular data types were highly correlated with sequencing quality, differentiation efficiency, and sex.

#### QTL covariates

For all QTL analyses, we included the following as general covariates: sex, iPSC passage number, the first five genotype PCs to control for global ancestry, and PEER factors to account for hidden confounders of molecular phenotype variability (see **Quantitative Trait Loci (QTL) Mapping: PEER Factor Calculation**).

#### QTL mapping

QTL mapping was performed in two steps: 1) QTL Discovery, and 2) QTL Filtering, for each of the 8 iPSCORE molecular datasets independently (see Supplemental Methods).

Step 1: QTL Discovery

The QTL Discovery step was performed using a linear mixed model (LMM) with the kinship matrix as a random effect to account for the genetic relatedness between samples. First, using rank and *qnorm* in R (v4.2.1), we inverse normal transformed the TMM gene expression/peak accessibility or acetylation values across the samples. Genes within 1 Mb and peaks within 100 kb of the MHC region[102] (chr6:28,510,120-33,480,577) were removed due to the complex LD structure in the interval. For the elements (i.e., genes and peaks) outside the MHC region, we used bcftools[77] query to obtain the genotypes for all the variants within 1 Mb for genes or 100 kb

for ATAC-seq peaks and H3K27ac ChIP-seq peaks. Then, we applied the scan function in limix (v3.0.4) (https://github.com/limix/limix) to run the following linear mixed model:

$$Y_i = \beta_j X_{ij} + \sum_{m=5}^{M} \gamma_m PC_{im} + \sum_{n=1}^{N} \gamma_n PEER_{in} + \sum_{p=1}^{P} \gamma_p C_{ip} + u_i + \epsilon_{ij}$$

Where $Y_i$ is the normalized expression value for sample $i$, $\beta_j$ is the effect size (fixed effect) of SNP $j$, $X_{ij}$ is the genotype of sample $i$ at SNP $j$, $M$ is the number of genotype principal components used ($M = 5$ for all QTL analyses), $\gamma_m$ is the effect size of the $m$th genotype principal component, $PC_{im}$ is the value of the $m$th genotype principal component for the individual associated with sample $i$, $N$ is the number of PEER factors (See **Quantitative Trait Loci (QTL) Mapping: PEER Factor Calculation**), $\gamma_n$ is the effect size of the $n$th PEER factor, $PEER_{in}$ is the value for the $n$th PEER factor for sample $i$, $P$ is the number of covariates used ($P = 1$ for all QTL analyses corresponding to the iPSC passage number), $\gamma_p$ is the effect size of the $p$th covariate, $C_{ip}$ is the value for the $p$th covariate for sample $i$, $u_i$ is a vector of random effects for the individual associated with sample $i$ defined from the kinship matrix, and $\epsilon_{ij}$ is the error term for individual $i$ at SNP $j$.

For FDR correction, we used a two-step procedure described in Huang et al.,[103] which first corrects at the gene or peak level and then at the genome-wide level. First, we performed FDR correction on the $p$-values of all independent variants tested for each gene or peak using eigenMT,[84] which considers the LD structure of the variants. Then, we extracted the lead variant for the QTL for each gene or peak based on the most significant FDR-corrected $p$-value. If more than one variant had the same FDR-corrected $p$-value, we selected the one with the largest absolute effect size as the lead variant for the QTL. For the second correction, we performed FDR-correction on all lead variants using Benjamini-Hochberg (q-value). We considered only QTLs with q-value <0.05 as significant (Table S4).

To identify additional independent QTL associations for a gene or peak (i.e., conditional QTLs), we performed stepwise regression analysis in which we re-performed QTL analysis with the genotype of the lead variant for the QTL as a covariate. We repeated the procedure to discover up to three conditional associations. For each iteration, we performed the two-step procedure described above and considered conditional eQTLs with q-values <0.05 as significant.

Step 2: QTL Filtering

The conditional QTL analysis corrects for variants in high LD with the lead variants from the primary QTL (i.e., the primary signal is corrected for in the conditional 1 signal). However, we observed that ~43% of the conditional QTL lead variants were still in high D′ with the primary QTL lead variants, suggesting that they are not independent genetic signals. Therefore, we incorporated a filtering step that identified and removed conditional QTLs in high LD (D' $\geq$ 0.8 and/or $r^2 \geq$ 0.8) with the primary QTL (Figure S6). To filter conditional QTLs with lead variants in LD with their corresponding primary QTL lead variants, we required that both variants be present in the 1000 Genomes Project EUR population. We removed conditional QTLs that did not meet this requirement, but we retained the primary QTLs for use in downstream analyses even if they were not present in the 1000 Genomes Project EUR population.

To filter conditional QTLs with lead variants in LD (D' $\geq$ 0.8 and/or $r^2 \geq$ 0.8) with their corresponding primary QTL lead variants, we first required that the conditional lead variants be present in the 1000 Genomes Project EUR population, hence 254 conditional QTLs were removed. After this initial filtering, 46,264 qElements only had a primary QTL signal, while 14,042 qElements had conditional signals. Of these 14,042 qElements, 255 had primary QTLs with lead variants that are not present in the 1000 Genomes EUR population. In these cases, we cannot determine the relationship between the primary lead variant and the conditional lead variant(s); therefore, we retained the 255 primary QTLs associated with these qElements and removed the 343 conditional QTLs. We retained the 255 primary QTLs even though their lead variants were not in the 1000 Genomes Project EUR population because downstream analyses, such as GWAS colocalization, do not require the QTL lead variant to be present to calculate the posterior probabilities for the remaining SNPs in the loci.

After the above filtering, we retained 13,787 qElements with both primary and conditional QTL lead variants in the 1000 Genomes Project EUR population. We next calculated the LD and D′ between the 17,949 conditional QTL lead variants and their corresponding 13,787 primary QTL lead variant in the 1000 Genomes EUR population, using *plink*.[83] We identified 7,792 non-independent conditional QTLs ($r^2 \geq$ 0.8 and/or D' $\geq$ 0.8), 4 primary QTLs with monomorphic lead variants (in the 1000 Genomes Project EUR population) with 5 associated conditional QTLs, and 12 conditional QTLs with monomorphic lead variants. We removed the 7,792 non-independent conditional QTLs, 12 monomorphic conditional QTLs, and the 5 conditional QTLs associated with monomorphic primary QTLs. This resulted in 4,963 qElements that lost all associated conditional QTLs and 8,820 qElements with 10,140 conditional QTLs. The 8,820 primary QTLs with conditional QTLs were regressed prior to GWAS colocalization (see **GWAS Associations with QTLs: Primary QTL Regression** and Figure S10).

In summary, QTL Filtering (Step 2), resulted in 70,446 QTLs including 60,306 primary (51,486 non-regressed and 8,820 regressed) and 10,140 conditional. Since conditional QTLs are identified in a stepwise manner, we sorted and re-numbered the remaining conditional QTLs sequentially after filtering non-independent conditional QTLs (Table S4). We characterize the 60,306 primary and 10,140 conditional QTLs after the filtering step in Figures 2B–2D; use the 60,306 primary lead variants in characterization analyses (Figures 2E, 3, and 4); and use the 60,306 primary QTLs for GWAS colocalization (Figures 5 and 6).

## QTL characterization

### eGene discovery rate

To examine the relative power of identifying eQTLs, we compared the three iPSCORE EDev-like tissues to the 49 tissues in the GTEx Consortium[1] and showed that they had similar eGene discovery rates (Figure S7). We note that sequencing depth differences across molecular data types may have resulted in different caPeak and haPeak discovery rates.

### caPeaks enriched near eGenes

We performed a two-sided Fisher's Exact test to evaluate the enrichment of caPeaks within a 100 kb window upstream any eGene compared to 100 kb window upstream of any expressed gene without an eQTL.

### caQTL discovery rate by peak width

To evaluate whether the caQTL discovery rate differed by ATAC-seq peak width, we first divided all the ATAC-seq peaks into seven bins based on their peak width. For each bin, we calculated the fraction of peaks that had at least one caQTL signal (caPeaks; Figure S8).

### Chromatin state enrichment

We obtained ChromHMM chromatin states in the hg38 build for iPSC line 18a (BSS00737) and embryonic stem cell-derived cardiac muscle (BSS00171) from the EpiMap Repository (https://compbio.mit.edu/epimap). We collapsed the 18 chromatin states into 5 categories: Promoter (which included TssA, TssBiv, TssFlnkU, TssFlnkD, TssFlnk), Enhancer (EnhA1, EnhA2, EnhG1, EnG2, EnhWk, EnhBiv), Repressed (ReprPC, ReprPCWk, Het, ZNF/Rpts), Transcribed (Tx, TxWk), and Quiescent (Quies). For iPSCs and CVPCs, using *bedtools intersect*, we annotated the lead variant, regardless of significance, from each QTL test across the three molecular phenotypes with the 5 collapsed chromatin states that they overlapped in their corresponding EpiMap tissue. For both tissues, we performed two-sided Fisher's Exact tests to test the enrichment of unique lead variants from significant QTLs of each molecular data type in each of the 5 collapsed chromatin states, using the unique lead variants from non-qElements (i.e., genes and peaks without a QTL, q-value >0.05) as background.

### TFBS enrichment in CVPC caPeaks and haPeaks

To characterize TF binding in regulatory elements affected by variation, we analyzed the CVPC ATAC-seq and H3K27ac ChIP-seq datasets. We first classified CVPC ATAC-seq peaks based on whether they were a caPeak and whether that caPeak overlapped ($\geq$ 1 bp) a CVPC haPeak. Using only ATAC-seq peaks with at least one predicted TFBS, we performed a two-sided Fisher's Exact test to calculate the enrichment of 444 JASPAR motifs of TFs that are expressed in CVPCs (see **ATAC-seq Peak Transcription Factor Predictions)** in caPeaks, caPeaks-haPeaks and CVPC ATAC-seq peaks without a caQTL (no QTL).

## Temporal eQTL annotations

### Identification of temporal eQTLs with mashr

We used *mashr*[55] to identify temporal (early developmental-specific and adult-specific) and shared eQTLs. First, we downloaded full eQTL summary statistics from https://console.cloud.google.com/storage/browser/gtex-resources for 47 adult tissues in the GTEx Consortium. We filtered all INDEL lead variants, resulting in 281,938 SNPs. Using the eQTL analysis vignette in https://stephenslab.github.io/mashr/articles/eQTL_outline.html, we computed posterior summaries (local false sign rate, LFSR) for all lead variants discovered in the 3 iPSCORE tissues and the 47 GTEx tissues using a model fitted on: 1) data-driven covariances calculated from the "top" eQTL for each eGene based on maximum absolute beta effect size, and 2) canonical covariances and correlation structure calculated on a random set of 200,000 SNP-gene pairs tested in more than 50% of all 50 tissues.

We used the 47 adult GTEx tissues to represent the adult-stage and the three iPSCORE tissues to represent the early developmental (EDev-like) stage. We removed all SNP-gene pairs that either were not tested in at least one adult and at least one EDev-like tissue or were not significant in any tissue (minimum LFSR across the 50 tissues >0.05). The mashr output is deposited on Figshare.

### Temporal annotations of SNP-eGene pairs

We calculated the minimum LFSR across adult and EDev-like tissues and considered a QTL to be EDev-specific if it was significant (LFSR <0.05) in any EDev-like iPSCORE tissue and not significant (LFSR >0.05) in any adult tissue. Alternatively, adult-specific QTLs were only significant (LFSR <0.05) in one or more adult GTEx tissues. Shared QTLs were significant in at least one EDev-like tissue and at least one adult tissue. This resulted in 2,299 EDev-specific, 27,881 adult-specific, and 72,195 shared eQTLs between EDev-like and adult tissues (i.e., CVPC vs. adult left ventricle). We note that the set of iPSCORE EDev-specific QTLs may vary from those that would be identified using early developmental tissues.

### Temporal eQTL validation

To validate the activity of temporal-specific and shared eQTLs in EDev-like and adult tissues, we independently calculated the Pearson correlation coefficient ($r^2$) of the effect sizes of the 2,299 EDev-specific, 27,881 adult-specific, and 72,195 shared eQTLs. We performed two-sided t-tests to show that the effect sizes of the temporal eQTLs were not correlated in EDev-like and adult tissues, while the effect sizes of the shared eQTLs were correlated.

### EDev or Shared Annotation of iPSCORE eQTLs

We used the *mashr* classifications to annotate the iPSCORE EDev-like eQTLs for downstream analyses not including the GTEx eQTLs. We annotated the 19,305 primary eQTL lead variants from the EDev-like tissues with *mashr* specificity assignment (EDev-specific, adult-specific, and shared). *mashr* can calculate a non-significant LFSR for an eQTL lead variant or assign specificity

that is not aligned with the original annotation (i.e., a CVPC eQTL is annotated as adult-specific), which we classified as "No Association". Initially, we annotated 2,269 EDev-specific, 10,998 Adult-specific, and 6,038 No Association eQTLs. Since complex QTLs are composed of qElements affected by the same QTL, they cannot be composed of qElements with lead variants with different mashr assignments. If a complex QTL was annotated by a "No Association" SNP-eGene pair, we considered all qElements within that complex QTL to be "No Assocation". Further, complex QTLs composed of qElements of both "EDev-specific" and "Shared" lead variants were considered shared. After the reassignments, we annotated 2,046 EDev-specific, 10,717 Shared, and 6,542 No Association eQTL lead variants. This set of eQTLs were used to examine the fraction of EDev-specific eQTLs found in the iPSCs, CVPCs, PPCs, and evaluate the effect size differences between EDev-specific and shared eQTLs in the 3 iPSCORE tissues. To account for differences in statistical power between the 3 EDev-like tissues, we performed two-sided Mann Whitney U tests to compare the absolute effect size of the lead variants of EDev-specific and shared eQTLs within each EDev-like tissue, independently.

### Identification of Complex QTL
#### Complex QTLs
We sought to identify qElements (eGenes, caPeaks, and haPeaks) across the three molecular data types that shared primary QTLs. We used *plink*[83] to extract the primary QTL lead variants from the 1000 Genomes EUR population for each chromosome independently and calculated the LD between all variants. Within each tissue, QTLs with lead variants in high LD ($r^2 > 0.8$) and within 100 kb were considered shared.

To identify complex QTLs affecting multiple qElements, we loaded each pair of shared QTLs ($r^2 > 0.8$) as edges into an *igraph* (v1.3.2)[85,86] network, for each tissue independently. We clustered the QTL networks using the *cluster_louvain* function to assign QTLs to independent modules. In total, there were 5,672 modules representing complex QTLs affecting two or more qElements, and 46,702 singleton QTLs that were not in LD with another QTL (Table S4).
#### Complex and Singleton QTL qElement distance
We calculated the minimum distance between the lead variant of the 25,013 primary CVPC QTLs and their corresponding qElement. For all CVPC QTLs, we calculated the minimum distance between the lead variants and the nearest TSS of an expressed gene. We performed two-sided Mann Whitney U tests to evaluate whether complex and singleton QTLs had different distributions.

### GWAS associations with QTLs
#### GWAS traits
From the UK Biobank (https://pan.ukbb.broadinstitute.org/downloads/index.html), we downloaded summary statistics for ten traits: angina pectoris, atrial fibrillation, body mass index, HDL cholesterol, ischemic heart disease, LDL direct, acute myocardial infarction, pulse rate, QRS duration, and ventricular rate. From the Early Growth Genetic Consortium (http://egg-consortium.org/), we downloaded summary statistics for two traits: childhood obesity[66] and birth weight.[67] From the Meta-Analyses of Glucose and Insulin-related Traits Consortium (http://magicinvestigators.org/downloads/), we downloaded summary statistics for fasting glucose.[64] From the DIAGRAM Consortium (https://diagram-consortium.org) we downloaded summary statistics for type 2 diabetes.[68] From the Edinburgh Data Share (https://datashare.ed.ac.uk/handle/10283/3209; https://datashare.ed.ac.uk/handle/10283/3599), we downloaded summary statistics for a multivariate GWAS that accounted for parental lifespan, healthspan, and longevity.[69] All traits are listed in Table S5, along with their study sources. All summary statistics were provided in hg19 coordinates. To convert coordinates from hg19 to hg38, we used the liftOver software downloaded from UCSC (https://genome-store.ucsc.edu/). Then, we sorted and indexed each GWAS summary statistics file using tabix.[77]
#### LD Score Regression
To estimate the enrichment of heritability for the 15 developmental and adult GWAS traits in ATAC-seq and ChIP-seq peaks in the three EDev-like tissues, we performed LD Score Regression (LDSC, v1.0.1)[87] using the HapMap3 variants that the developers found to be optimal for the analysis. First, we annotated each HapMap3 variant with a binary label (1/0) indicating whether the variant overlapped the ATAC-seq peak or H3K27ac ChIP-seq peak in iPSC, CVPC, and PPC. Then, we estimated LD scores for each annotation with ldsc.py –l2 using 1000 Genomes Phase 3 reference files in hg38 available at broad-alkesgroup-public-requester-pays/LDSCORE/GRCh38/plink_files.tgz. Finally, we tested for heritability enrichment with ldsc.py –h2 using regression weights downloaded from broad-alkesgroup-public-requester-pays/LDSCORE/GRCh38/weights.tgz and baseline annotations (v.1.2) from broad-alkesgroup-public-requester-pays/LDSCORE/GRCh38/baseline_v1.2.tgz. Annotations were enriched for trait heritability if *p*-values (Enrichment_p) < 0.01 (Figure S11).
#### Primary QTL regression
The presence of multiple signals for a given qElement (eGene, caPeak, and haPeak) can affect Bayesian colocalization, therefore, prior to GWAS-QTL colocalization. we regressed the effects of conditional QTLs from the primary QTL signals (Figure S10). After non-independent conditional QTLs were removed (Figure S6; see section **QTL Filtering**), for each Primary QTL ($n = 8,820$ with at least one conditional QTL), we modified the QTL equation (see section **QTL Discovery**) by including the conditional QTL lead variant genotype(s) as covariates. We next performed FDR correction on the *p*-values of all independent variants tested for each gene or peak using eigenMT[84] and assigned the variant with the most significant FDR-corrected *p*-value as the lead variant for the GWAS colocalization analysis. If two or more variants had equally significant FDR-corrected *p*-values, we selected the one with the largest

absolute effect size as the lead variant for the QTL. The summary statistics for the 8,820 regressed Primary QTLs, along with the 51,486 non-regressed Primary QTLs (Figure S6; Figure S10), were used for GWAS-QTL colocalization.

### GWAS-QTL colocalization

For all 60,306 primary QTLs (51,486 non-regressed and 8,820 regressed), we performed pairwise colocalization with GWAS variants for 15 traits (see **GWAS Traits** for list of GWAS summary statistics) using effect size and variance as input into the *coloc.abf* function in *coloc*[25] (v5.2.2). Given that complex QTLs influence multiple qElements, we chose one qElement per complex and assigned its corresponding QTL to represent the complex QTL for GWAS colocalization (Table S5). We randomly chose from all qElements whose lead variants were in the 1000 Genomes Project EUR population to represent the complex QTL; but if none of the lead variants were present in the 1000 Genomes Project EUR population, we randomly chosen the lead variant from all qElements in the complex QTL. To determine whether representative complex or singleton QTLs colocalized with a GWAS loci, we required that all of the following criteria were satisfied: 1) had at least 50 overlapping variants; 2) PP.H4 $\geq$ 80%; 3) the lead candidate causal variant is genome-wide significant for GWAS association ($p$-value $\leq$ 5x10$^{-8}$); 4) the lead candidate causal variant is genome-wide significant for QTL association ($p$-value $\leq$ 5x10$^{-5}$); and 5) the lead candidate causal variant had a PP $\geq$ 1%. For each GWAS-QTL colocalization, *coloc*[25] outputs a causal PP for each variant that was tested during colocalization. We assigned a lead candidate causal variant for each GWAS-QTL pair by taking the variant with the highest causal PP.

### Fraction of GWAS loci colocalized with QTLs

To determine the fraction of GWAS loci explained by QTLs, we calculated the number of independent genome-wide significant loci for each of the 15 GWAS studies. Specifically, we first filtered for variants that were above the genome-wide significant threshold of $p < 5x10^{-8}$. Then, we LD pruned the GWAS variants using plink[83] with the following parameters: *–indep-pairwise 500 5 0.1*, where 500 is the variant count window, 5 is the step count, and 0.1 is the LD threshold. We found that these parameters yielded the same number of distinct association signals in type 2 diabetes previously observed ($n$ = 403).[68] Considering each of the 15 traits independently, the above command outputs a list of 11 to 1,837 independent variants (i.e., not in LD) that each represent an independent genome-wide significant GWAS locus. Combining all 15 traits, 5,192 GWAS loci were found.

We then sought to identify the subset of the 5,192 GWAS loci that colocalized to a QTL (either a complex or singleton). Using 1000 Genomes (Europeans only) as reference, we calculated LD between the lead candidate causal variants from the GWAS-QTL colocalization (see **GWAS-QTL Colocalization)** and the LD-pruned GWAS variants using *plink –tag-kb 350 –tag-kb 0.7*.[83] If the lead candidate causal variant was in high LD ($r^2 \geq$ 0.7 within 350 kb) with an LD-pruned GWAS variant, then we assigned the complex or singleton QTL to that GWAS locus. For lead candidate causal variants absent from the reference panel, and therefore LD could not be calculated, we assigned the complex or singleton QTL to the nearest GWAS locus. We observed that 80 of the 5,192 (1.5%) GWAS loci were in LD with multiple complex/singleton QTLs from the same tissue (range 2–4 complex/singleton QTLs per GWAS signal) (Table S5), which could reflect independent signals within the same GWAS locus or a limitation of *coloc* in assuming a single causal variant.[25] In total, 863 (164 representative complex and 699 singleton) QTLs colocalized with 540 GWAS loci.

### Comparing GWAS Loci Distance to Nearest Gene

We calculated the distance between the LD-pruned GWAS variants for the 5,192 GWAS loci and the Gencode version 44[71,95] coordinates of the nearest protein-coding gene's TSS, using *bedtools closest*.[80] We performed a two-sided Mann Whitney U test to evaluate if distance distributions were different between colocalized and non-colocalized GWAS loci (Figure 5D).

### Comparing GWAS loci-gene distance by QTL Type

We annotated the 540 colocalized GWAS loci with the QTL types that they were associated with (caQTL, eQTL, haQTL, caQTL-eQTL, caQTL-haQTL, eQTL-haQTL, caQTL-eQTL-haQTL). We performed two-sided Mann Whitney U tests to evaluate if distance distributions (calculated in **Comparing GWAS Loci Distance to Nearest Gene**) were different between colocalized GWAS loci associated with different QTL types (Figure 5C).

### Enrichment of Complex QTLs with GWAS variants

We annotated each QTL based on their associated molecular phenotypes and whether they were a representative complex QTL or a singleton QTL (Table S4). For example, if the QTL represented a complex QTL affecting only caPeak(s) and eGene(s), we annotated the complex QTL as a "complex caQTL-eQTL". If an eQTL was a singleton, we annotated the QTL as "eQTL singleton". We considered a total of ten categories: 1) complex caQTL-haQTL-eQTL, 2) complex caQTL-haQTL, 3) complex haQTL-eQTL, 4) complex caQTL-eQTL, 5) complex caQTL, 6) complex eQTL, 7) complex haQTL, 8) singleton caQTL, 9) singleton eQTL, 10) singleton haQTL (Figure 5B). Enrichment of each of these categories for GWAS colocalization was calculated using a two-sided Fisher's Exact test, where the contingency table consisted of two classifications: 1) if the complex or singleton QTL corresponded to the category, and 2) if the complex or singleton QTL colocalized with at least one GWAS trait. A category was considered enriched for GWAS colocalization if the $p$-value <0.05. In addition to performing enrichment analysis across all three tissues (Figure 5E), we examined each tissue independently (Figure S12).

### GWAS loci stage-specificity

Since *mashr* compares lead variant effect sizes to calculate specificity (LFSR),[55] an eQTL assigned as "EDev-specific" can still exhibit diminished, but significant regulatory activity in adult tissues, thus affecting the annotation of temporal regulatory variation and the interpretation of GWAS colocalization. We annotated 239 GWAS loci that colocalized with an eQTL as EDev-specific, Shared, or "No association" based on the annotation of their colocalized iPSCORE eQTLs (section **EDev or Shared Annotation of iPSCORE eQTLs**). There were 7 GWAS loci that colocalized with multiple eQTLs with different temporal annotations which could be a result of

the *mashr* LFSR calculation or the assumption of a single causal variant in *coloc*. To resolve a single temporal annotation, we considered GWAS loci "Shared" ($n$ = 181) if they colocalized with any Shared eQTL, including those that also colocalized with an eQTL with an EDev-specific and/or "No association" annotation. We next considered GWAS loci as "EDev-specific" ($n$ = 13) if they colocalized with an EDev-specific eQTL, including those that also colocalized with an eQTL with "No association" annotation. Finally, we annotated the remaining GWAS loci as "No association" ($n$ = 45) because they did not colocalize with either an EDev-specific or Shared eQTL.

### Putative casual variant TF Overlap

To identify putative causal variants for GWAS loci, we calculated the 99% credible sets from each GWAS-QTL colocalization from the *coloc*[25] output. We aggregated the credible sets for the 699 singleton and 164 representative complex QTLs that colocalized with at least one of the 540 GWAS loci. To include additional putative causal SNPs in complex QTLs, we also aggregated 129 non-representative QTLs (i.e., QTLs associated with other qElements in the 164 complex QTL modules) that colocalize with the same GWAS loci as the corresponding representative complex QTL for a total of 992 GWAS-QTL colocalizations. We characterized the number of SNPs in these credible sets and defined 611 high-confidence credible sets as those with fewer than 25 SNPs. We next aggregated all predicted JASPAR[43] and HOCOMOCO[44] TF motifs identified using TOBIAS (**ATAC-seq Peak Transcription Factor Predictions**), regardless of whether they had a predicted binding site.

To identify motif-overlapping putative causal variants (MOPCVs), we intersected 6,164 SNPs in the 611 high-confidence credible sets with the predicted TF motifs, using *bedtools intersect*.[80] For the 164 complex QTLs modules, we collapsed credible sets from QTLs for different qElements. We assigned priority ranks to the MOPCVs based on the strength of their causal evidence. The High ranked MOPCVs were in caPeaks containing the associated caQTL lead variant, the Moderate ranked MOPCVs were in caPeaks containing a caQTL lead variant associated with a different caPeak, and the Low rank MOPCVs were in ATAC-seq peaks that were not associated by a caQTL. In Figure 6B, we report the number of GWAS loci that colocalized with a QTL (GWAS-QTL colocalization) with one or more SNPs in a credible set that overlapped a TF motif. In Table S6, for each GWAS locus (GWAS-QTL colocalization), we report the MOPCVs, their affected motifs, priority ranks, and other information that enable experimental validation.

