## [Document S2. Transparent peer review records for Arthur et al · Cell Genomics]

Summary

Initial submission: Received : 3/19/2024

Scientific editor: Laura Zahn

First round of review: Number of reviewers: 4
Revision invited : 5/1/2024
Revision received : 10/18/2024

Second round of review: Number of reviewers: 4
Accepted : 1/24/2025

Data freely available: Yes

Code freely available: Yes

This transparent peer review record is not systematically proofread, type-set, or edited. Special characters, formatting, and equations may fail to render properly. Standard procedural text within the editor's letters has been deleted for the sake of brevity, but all official correspondence specific to the manuscript has been preserved.

Referees' reports, first round of review

Reviewer #1: Since only 42% of GWAS loci can be colocalized with eQTLs, the authors tested whether QTLs with other modalities could increase the rate. They conducted multi-omic QTL mapping in iPSCs, iPSC-derived CVPCs, and PPCs and performed QTL module analysis and GWAS colocalization analysis. Their two main findings are: 1) Integration of QTLs from other molecular phenotypes, such as those related to chromatin states, could significantly improve the colocalization rate of GWAS loci. 2) Some of the GWAS loci colocalize with EDV-unique QTLs. However, QTLs unique to early development are generally significantly depleted in adult GWAS loci compared to adult-shared QTLs, suggesting that early developmental-specific regulatory variation plays a limited role in adult diseases and traits. Given the scale of this study, the multi-omic QTL datasets provide a valuable resource for the community. Overall, the analyses are rigorous, and the findings provide interesting insights into distinct regulatory mechanisms with different molecular signatures. My specific comments are the following:

1. One of the most interesting aspects of the study is that iPSCs from different ancestry groups are profiled. However, this information was not explored, representing a missed opportunity.
2. The efficiency and rate at which different iPSC lines differentiate into specific cell types, such as CVPC and PPC, vary significantly. How is this issue controlled in the study? Should it be controlled by comparing the RNA-seq data to single-cell RNA-seq data to capture such heterogeneity?
3. Line 110, what are the genomic distributions (e.g., intron, intergenic, exon, etc.) of ATAC-seq peaks and H3K27ac ChIP-seq peaks?
4. One would expect that the majority of H3K27ac peaks are ATAC-seq peaks. Line 110, in the corresponding cell types, what are the proportions of overlapped and distinct peaks between ATAC-seq and H3K27ac ChIP-seq peaks? What are the TF motifs in the overlapped and distinct peaks? It may be interesting to address these questions, as the overlapped ATAC-seq and H3K27ac ChIP-seq peaks that are not promoters are more likely to be active enhancers.
5. Line 146, the authors showed similar eGene discovery rates between their study and GTEx. Can the authors also show the overlapping rate of their mapped eQTL-eGene pairs with those from GTEx?
6. Line 193, can the authors briefly discuss the TFs identified in caPeaks that overlapped haPeaks and those that did not overlap haPeaks? Which developmental stages do these TFs regulate, e.g., during early development or after cell differentiation?
7. Line 231, 18% of QTL signals are associated with QTL modules, while 82% are singleton QTLs. Can the authors further characterize the 82% singleton QTLs? For example, are the singleton eQTLs enriched in promoters more than modular eQTLs? Since the chromatin state of promoters may be less likely affected by genetic variation, could it be possible that only singleton eQTLs but no caQTL/haQTL are enriched in promoters? Or are these singleton eQTLs enriched in exonic, UTR, etc.? What are the distance distributions between singleton eQTLs and TSS of eGenes compared to modular eQTLs? Additionally, are the singleton caQTLs and haQTLs enriched in the non-overlapped snATAC-peaks and H3K27ac ChIP-seq peaks, as the non-overlapped peaks may be primed/poised enhancers?
8. Some functional validation of the findings from the study would be useful to further strengthen the manuscript. For example, it would be informative to conduct a reporter assay on the top candidate to show that `cvpc_atac_peak_233871` is an enhancer in CVPC, or/and `ppc_atac_peak_197599` is an enhancer in PPC. Do the two SNPs disrupt any TF motifs?
9. Line 306, the authors found the effect sizes of EDev-unique eQTLs are lower than those of adult-shared eQTLs. Did they also examine whether the effect sizes of EDev-unique caQTLs and haQTL are lower compared to adult-shared caQTLs and haQTL?
10. Figure 4a and Line 237, since the QTLs were mapped in IPSCORE cell lines that are more likely to reveal the genetic variant effects on gene regulation during early development, the authors should also consider the GWAS for developmental diseases related to the derived iPSC cell lines. For example, as they mapped QTLs in iPSC-derived cardiovascular progenitor cells (CVPC), could they also fine-map GWAS of congenital heart disease, congenital anomaly of the arteries, etc.?
11. Line 328, the authors should modify the title and conclusion of this section to "Early developmental-unique QTLs are depleted for colocalizing with Adult GWAS loci".

Minor comments:

1. Gene names in both text and figures should be italicized.
2. Inconsistent WGS sequence depth mentioned in the method part. Line 612 mentions 50x WGS, but line 761 states 30x depth of coverage.
3. On line 669, there appears to be a missing closing parenthesis.
4. The font sizes within figures lack consistency; ensuring uniformity at least within individual figures would enhance their readability and aesthetic appeal.
5. In Figure 3, panel d is missing a color key.

Reviewer #2: Comments enter in this field will be shared with the author; your identity will remain anonymous. In their paper, Arthur and colleagues describe results of an important study evaluating potential explanations for the smaller than expected proportion of GWAS loci that colocalize with eQTLs. They specifically evaluated two complementary theories: that causal GWAS variants regulate the expression of genes only expressed during development (so will be missed in eQTL studies in adult tissues - i.e., GTEx) and/or that causal GWAS variants do not have direct effects on gene regulation (and visible as eQTLs) but rather affect the regulation of genes through their effect on chromatin accessibility and/or histone modifications. They perform these studies in the iPSC Omics Resource, using both previously published data and new data reported here for the first time, focusing on gene expression (eQTLs), chromatin accessibility (caQTLs), and H3K27ac (haQTLs) in undifferentiated iPSCs (to reflect early development cells) and those differentiated into cardiovascular progenitor cells and pancreatic progenitor cells (to reflect adult cells), as well as comparisons to eQTL data from GTEx (many adult tissues). They presented the concept of QTL modules and qElements that integrated across QTL types and revealed many important features about the gene regulatory landscapes associated with GWAS loci. Overall, this is an excellent study with an extremely rigorous design that revealed two important take-home messages, in addition to many other valuable insights. First, QTLs that are unique to developing tissues (iPSCs) are depleted in GWAS, thereby debunking the suggestion that the smaller than expected proportion of GWAS loci that colocalize with eQTLs are due to the paucity of development-specific eQTLs used in these studies. Second, QTL modules, especially those with caQTLs and haQTLs, colocalize with GWAS loci nearly twice as often as eQTLs alone, highlighting the importance of including these types of QTLs in post-GWAS colocalization studies.

This paper is very clearly written and easy to read; figures and legends are excellent. I have only minor comments that may help improve the clarity of a few points.

Minor Comments

1. The term "conditional" is introduced early in the paper but I could not find it defined outside of the methods. I had assumed these referred to 3 different "conditions" that the cells were subjected to, and it wasn't until I poured through the methods that I saw this was "statistical conditional analysis" to identify up to 3 secondary independent QTLs. It would be helpful to say this when the term is introduced on line 142.
2. It is not clear to me why conditional QTLs would "reflect greater uncertainty in the posterior probabilities" (lines 169-171). Is this because they have smaller effect sizes?
3. It might be worth mentioning that the 17 traits from GWAS were selected to reflect cardio-metabolic traits (or something like that) (line 238), which fits with the cells being studied, but is different from the GWAS traits usually selected for such studies.
4. I couldn't find anywhere in the paper why H3K27ac was not performed in the PPCs? I may have missed it in the methods but it would be helpful to mention this.

Reviewer #3: See attached file as it contains a table that couldn't be pasted in.

The paper asks if development QTLs, as approximated from iPSC and iPSC derived tissues, might colocalize with some proportion of GWAS signals that are not colocalized with GTEx (adult) tissue eQTL signals. It also examines if chromatin accessibility QTL (caQTLs) or H3K27ac QTL (haQTLs) might also colocalize with GWAS signals not detected by eQTLs. The paper finds evidence for enrichment of GWAS signal colocalization in ca and ha QTLs particularly in those signals that appear in at least one other QTL (ca, ha, e). It finds disenrichment in developmental QTL, as represented by iPSC and iPSC derived tissues. The paper is clearly written. There are multiple times where the tests performed assume that the input data are independent, results that are combined based on the same starting set of iPSC cells. The use of multiple omics increases interest in the results and the questions that can be asked. The uneven sample sizes across tissues and omic decreases interpretability somewhat.

*1. Data availability: The data availability is summarized in the table below, The newly generated data is marked with an "**". Two of the newly generated datasets do not appear to be present in the ascension number listed in Data availability. Red indicates*

missing data. Please add the missing ascension numbers.

Data availability

Tissue	Data type		
	RNA-seq	ATAC-seq	H3K27ac
iPSC	No ascension number	GSE203377	GSE261276*
CVPC	No ascension number	Not in GSE261276*	Not in GSE261276*
PPC	GSE182758	GSE261276*	No data generated

2. Code availability:

Main Figure code exists.

Please add code for Supplemental Figures or label files better to reflect where the code is.

Samples

3. The Supplemental Figure 3 Cardiac marker plots (b) shows samples with low levels of the cardiac cell markers in the cardiac cells. What criteria were used to include a cardiac cell line and why were these samples included in the analysis?

4. How much overlap is there in the samples used for the three omics type within a cell type. For each cell type please show the Venn diagram of overlap of samples used for the 3 omics types.

5. How much overlap is there in the samples used for three cell types within an omics type?. For each omics type please show the Venn diagram of overlap of samples used for the 3 cells types.

Abstract

6. Line 34. The word “functionally annotated” is not defined, does it mean colocalized?

7. Line 34. The 49% functionally annotated GWAS loci number needs context in terms of the actual % of GWAS loci that are colocalized.

Major analysis questions

8. Many of the tests in the paper use results across omics type and/or cell types. These assays are from the same set of samples, which can affect statistical tests that assume samples used in each test are independent. One example is Figure 2e. The Figure 2e is combining the results across iPSC and CVPC. Many of the signals are the same signals across the three iPSC and CVPC from the same people. This means the signals are not independent of each other. Likewise this question arises for 1) colocalization of omics signals within a cell/tissue type (which assumes two independent datasets), 2) Figure 4b enrichment of colocalization of different combinations of molecular signals relative to eQTLs, and 3) Figure 7a enrichment colocalization with GWAS of of adult shared eQTL compared to EDex-unique QTL, and potentially other tests. Please explain how you are accounting for the non-independence of the data in each test. How can you run the tests with (more) independent data?

9. Line 1058 What QTL data is used for colocalization of the primary signal? Does primary mean the ie marginal eQTL signal, before conditioning? Typically the primary signal is thenstrongest marginal signal adjusted for the other conditional variants (see below).

10. Line 1054 and 1058: The conditional analysis is described as a sequential identification

of QTL signals. When colocalizing with GWAS signals, is there a subsequent step to produce the distinct association signals for each distinct SNP by adjusting for all other distinct SNPs. For example if three conditionally distinct SNPs were identified are three separate regressions run for all SNPs in the 1MB region SNPs adjusting for SNP2 and 3 (for signal 1), SNP 1 and 3 (for signal 2), and SNP1 and 2 (for signal 3) This is the convention and is related to what Bayesian methods like SuSiE would do. The first signal, with adjustment for other signals, would be called the primary signal.

11. The statement that GWAS loci are highly enriched for colocalization with complex QTL molecules, may be because there is better power to detect colocalization with stronger signals (as is mentioned), not because there is inherently more colocalization. If you correct for the strength of signals do you see the same patterns of enrichments?

Results:

12. Line 265: The inclusion of the TF TBX20 example seems somewhat random. Why was there a focus on this locus in particular?

Discussion:

13. Line 371. There are reasons in addition to the two reasons listed that could explain why there is not complete overlap of eQTLs and GWAS loci: The relevant QTLs could be present in rarer cell types, and thus not show up on bulk data, and relatedly, sample sizes of bulk or single cell/nucleus data may be insufficient to detect the weaker eQTLs.

14. Address the limitations that there are not equivalent sample sizes for each tissue/cell type and thus it's not easy to compare across them and also that the developmental models may not fully capture the QTLs present in living developing tissue.

Figure 1

15. It is hard to see and compare the number of samples for the three tissue and three molecular data types. Including a 3x3 matrix with the numbers of samples in each cell could make this easier to read than the current format.

16. Please make a supplemental table that has the number of unrelated individuals for each tissue and omic type to help people judge the amount of data, or give an estimate of the effective number of people analyzed for each tissue and omics type.

Figure 2

17. Figure 2a: Indicate which of the tests are significant with a star or another demarcation in boxes. Please show the Shared signals for each cell type.

18. Figure 2f: Please explicitly state the comparison groups used for each of the three enrichment tests, in the Figure and in the Methods section.

19. Figure 2f Consider renaming the tests so they are distinctive without reading the legend.

20. Figure 2f. White color is used in the color scale to mean an OR of 1. The color is also set to white if the test is not significant. This is mixing two meanings. It could be clearer to use the $\log_2(\text{OR})$ color as the effect size (as expected from color scale) and indicate the significance with a star or other method in the appropriate boxes.

Figure 3.

21. Figure 3b. The pie chart would work (best) if there were an equal number of samples in each tissue and omics type. They are not equal and thus the proportions don't have an interpretable meaning. Is there another way to show this data?

Figure 4

22. Figure 4a. The scale of the bar plot for the total number of signals across all GWAS doesn't match the scale of the individual GWAS, and thus makes it hard to interpret the figure. The total sample bar could be shown on the original scale and then be expanded if needed for clarity.

Figure 5

23. *Figure 5a 7c-e LD pictures. The LD colors scale changes between the panels which makes it hard to track the LD patterns for the same reference variant. Please make the LD color scale the same for each plot, here and in any figure showing LD .*

Figure 7

24. *Figure 7e. The legend says this plot shows Adult eQTL, caQTL and haQTLs, however the points are not labeled as coming from any one of these specifically. Please clarify what is shown in the plot.*

Methods:

25. *Line 652: Clarify here that samples are from DO so are (presumably) iPSC cells.*

26. *Line 776 and others: There are substantial differences in the sequencing depths for the molecular phenotypes across the tissues. Consider a table containing this information and commenting on the effects of differential sequencing depth in the discussion on the discovery of QTLs.*

27. *Line 818: Why was the WGS data phased?*

28. *Line 972: There may be a typo as the text says the OR's were added to get the average: For example, the average of OR= .5 and OR =2, is OR=1, which is the result when adding on the log₂ (OR) scale not on the OR scale.*

29. *Line 1072: When testing for enrichment of caPeaks within a 100kb window around any gene, the larger the gene, the more caPeaks that will map to the gene by chance. If genes that take up smaller chromosomal regions are less likely to have an eQTL then there could be a bias. Please describe the test used for enrichment and verify that the length of the genomic region around a gene is not biasing the test (log(genomic region) could be included as a covariate (spline)).*

30 *Line 1076: The text in this section is a result. Please add to the results.*

31. *1Line 1148: There is no section 6.1.5 LD Score regression. Does this mean section 6.5.1 GWAS traits?*

32 *Line 1171: An r² criterion of >.1 is not a convincing threshold for independent GWAS signals. Is there a distance criteria (how far away does 500 SNPs get you). Change or talk about limitation of that criteria.*

Reviewer #4: Arthur, Nguyen et al. assayed three types of molecular QTLs, for expression (eQTLs), chromatin accessibility (caQTLs), and histone acetylation (haQTLs) in three different developmental tissues: iPSCs and their differentiation into cardiovascular progenitor cells (CVPCs) and pancreatic progenitor cells (PPCs). They first performed various analyses characterizing the properties of these eQTL types, e.g., their enrichment in different regulatory annotations and the patterns of QTL sharing. Then, they evaluated the colocalization of these molecular QTLs with GWAS hits.

I think the effort to catalog these QTL types in development-like tissues is useful, particularly given one prevailing hypothesis about the missing GWAS colocalizations, which suggests that trait variants may be context-specific and missed in conventional QTL assays. However, my assessment is that the authors could do more characterizations of these QTLs in relation to GWAS variants. Also, I have major concerns about many of the data interpretations and conclusions. That said, I think these can be addressed.

Major comments:

- As motivation, in the introduction, the authors mention a recent study showing systematic differences in the discovery of eQTLs and GWAS hits (Mostafavi et al.). For the search for missing GWAS colocalizations, it would be useful to characterize: (i) whether/how adult-tissue QTLs are different from development-like tissue QTLs, and (ii) how different QTL types compare to each other and control variants in terms of the properties of the tentative target genes (e.g., selective constraint), distance to genes, etc. My expectation is that the discovered QTLs are biased away from GWAS variants in important ways (though not exactly like

eQTLs), considering the small sample sizes used in the study. [I note that target genes are not known for caQTLs and haQTLs, for which perhaps the nearest gene can serve as a proxy.]

- I did not understand what "QTL modules" are and what the motivation for them is.

- How were these 17 traits chosen? While plausibly true for a subset of traits, it is not obvious whether the three studied tissues are relevant trait-contexts to search for colocalizations. The authors could nominate top cell types through heritability enrichment (using s-LDSC) in open chromatin regions of these tissues compared to other tissues and cell types (e.g., in the single-cell atlas of Zhang et al., Cell 2021). [I may have missed this if authors have already reported that.]

- The authors report that GWAS variants are more likely to colocalize with QTLs with large effects. This is most likely a power bias, meaning it is much easier to detect colocalization for stronger QTLs (see Hukku et al., AJHG 2021). As written, it is implied that this is an intrinsic feature of GWAS variants.

- For the example of TBX20, it is not obvious why a CVPC QTL may causally affect birth weight. I don't think the fact that birth weight affects the risk of heart defects means that CVPC is an important cell type for birth weight, as argued by the authors. Thus, I am not sure what biological insight we gain from the colocalization presented in Figure 5.

- Similar to my first comment, it would be illuminating to investigate the systematic differences between GWAS hits that colocalize with QTLs versus GWAS hits without any colocalization.

- I do not think I agree with the characterization of developmental-unique QTLs versus adult-shared QTLs, and I suspect the results in Figures 6 and 7a are a consequence of the analysis design rather than due to different intrinsic features of developmental and adult QTLs.

First, the text implies that developmental eGenes are high pLI genes. But my understanding is that this is relative to adult-shared eGenes, not random/control genes. Also, the number of eGenes used for comparison is not matched (532 versus 8,257). This is problematic because the properties of eGenes vary by their rank (or degree of significance). A proper test would be to compare constraint for the top 532 eGenes in both, as well as random/control genes.

Second, I think the trends shown in Figure 6 can be explained by power and discovery biases. Specifically, for unique and shared QTLs acting on the same gene, natural selection is stronger on the shared QTLs. Thus, discovery of shared QTLs will be more biased towards weakly selected genes. As such, I think the statement in the Discussion that "unique regulatory variants are strongly selected against" is incorrect; in fact, I think the opposite is true. The authors can test this by exploring the reverse, i.e., adult-unique versus shared QTLs, or by comparing QTLs with respect to measures such as CADD score.

Third, the interpretation of the differential colocalization of developmental-unique QTLs versus adult-shared QTLs with GWAS hits is complicated. As mentioned above, shared QTLs are likely skewed towards less GWAS-relevant genes. But this can be counterbalanced by the fact that shared QTLs have more paths to contribute to trait variations (through activity in multiple causal tissues). This is also compounded by different discovery biases mentioned above that have huge effects on colocalization analyses. For example, shared QTLs are more likely to be promoter variants, which have larger effects and are thus easier to detect in colocalization.

Authors' response to the first round of review

We extend our gratitude to all reviewers for their insightful feedback and 1 comments. In total, we addressed over 70 comments. While we attempted to incorporate all suggested analyses into the manuscript, we recognized that doing so might obscure the central message and prevent us from satisfying a request to simplify the manuscript. Thus, we integrated most analyses and provided justifications for omitting others in our responses. Additionally, in response to reviewer's comments we re-conducted several analyses using alternative methods, which not only yielded more reliable results but also enhanced the overall quality of the paper.

Reviewer #1

Since only 42% of GWAS loci can be colocalized with eQTLs, the authors tested whether QTLs with other modalities could increase the rate. They conducted multi-omic QTL mapping in iPSCs, iPSC-derived CVPCs, and PPCs and performed QTL module analysis and GWAS colocalization analysis. Their two main findings are: 1) Integration of QTLs from other molecular phenotypes, such as those related to chromatin states, could significantly improve the colocalization rate of GWAS loci. 2) Some of the GWAS loci colocalize with EDV-unique QTLs. However, QTLs unique to early development are generally significantly depleted in adult GWAS loci compared to adult-shared QTLs, suggesting that early developmental-specific regulatory variation plays a limited role in adult diseases and traits. Given the scale of this study, the multi-omic QTL datasets provide a valuable resource for the community. Overall, the analyses are rigorous, and the findings provide interesting insights into distinct regulatory mechanisms with different molecular signatures. My specific comments are the following:

We thank the reviewer very much for their detailed review and the valuable comments on the manuscript. Addressing their comments greatly improved the analyses, discussion, and overall impact of our paper.

Below, we detail the specific changes made in response to their comments.

1. One of the most interesting aspects of the study is that iPSCs from different ancestry groups are profiled. However, this information was not explored, representing a missed opportunity.

We agree that ancestry-specific genetics is an important and understudied area in genomics research. Despite having representation from all superpopulations (African, East Asian, European, and South Asian) and admixed individuals, iPSCORE is still underpowered to perform statistical genetics at the ancestry level. There are 170 Europeans, 34 East Asians, 6 South Asians, 4 Africans, and 7 admixed individuals, which is reflective of the demographics at the area of iPSCORE recruitment (La Jolla, CA). If we expand iPSCORE in the future, we will be sure to include a greater number of non-European individuals to address the reviewer's comment.

2. The efficiency and rate at which different iPSC lines differentiate into specific cell types, such as CVPC and PPC, vary significantly. How is this issue controlled in the study? Should it be controlled by comparing the RNA-seq data to single-cell RNA-seq data to capture such heterogeneity?

We thank the reviewer for the questions, which indeed are important to consider when working with iPSC derived tissues. We and others^{1–7} have observed phenotypic variability in iPSC-derived tissues that is attributable to different cellular compositions. Using cellular deconvolution, which utilizes single-cell RNA seq data, we have demonstrated that CVPC heterogeneity is attributed to different proportions of cardiac cell types (i.e. cardiomyocytes, smooth muscle, endothelial, etc)^{2,4}. In Figure S16, we demonstrate that PEER factor 1 is highly correlated with the percent of cells expressing cardiac troponins (%cTnT; Figure S16 d-f) in CVPCs, and with the percent of PDX1-NKX6.1 positive cells in PPCs (% PDX1-NKX6.1; Figure S16 g-h). Our inclusion of PEER factors as covariates in the QTL linear mixed model corrects for the variable differentiation outcomes in the iPSC-derived tissues. We have added text to Figure S16 legend (lines 925–927) to clarify this point.

3. Line 110, what are the genomic distributions (e.g., intron, intergenic, exon, etc.) of ATAC-seq peaks and H3K27ac ChIP-seq peaks?

We have performed this analysis and show the results below. We used the ChIPseeker R package⁸10/18/2024 3:02:00 PM to functionally annotate the five peak datasets. The results are also reported in the Results (Lines 123–128), Methods (Lines 1265–1269) and in Figure S3. To summarize our findings, we observed that the largest proportion of ATAC-seq and ChIP-seq peaks are intronic, likely because introns, of both expressed and not expressed transcripts, cover >39% of the genome, while exons only represent 2.8% of the genome⁹. We observed that ATAC-seq peaks had a lower proportion of peaks in promoters (mean = 16.3%) compared to ChIP-seq peaks (mean = 32.6%), which is consistent with H3K27ac being a marker for promoters and enhancers¹⁰, while ATAC-seq capture any regulatory element in open chromatin¹¹. Finally, we observed a higher proportion of ATAC-seq peaks (mean = 28%) in intergenic regions compared to H3K27ac ChIP-seq peaks (mean = 17%). In general, our results are consistent with previous characterizations of the epigenome^{9–11} and suggest that the peaks capture active regulatory elements.

4. One would expect that the majority of H3K27ac peaks are ATAC-seq peaks. Line 110, in the corresponding cell types, what are the proportions of overlapped and distinct peaks between ATAC-seq and H3K27ac ChIP-seq peaks? What are the TF motifs in the overlapped and distinct peaks? It may be interesting to address these questions, as the overlapped ATAC-seq and H3K27ac ChIP-seq peaks that are not promoters are more likely to be active enhancers.

The reviewer is correct - the majority of the H3K27ac ChIP-seq peaks overlap at least one ATAC-seq peak (74.8% in iPSCs and 89.1% in CVPCs). We have characterized the TFBSs in the overlapping CVPC caPeaks and haPeaks in downstream analyses (Figure 2f) and observed that they are enriched with enhancer-binding TFs, including TEAD12,13 and MEF (myocyte-specific enhancer-binding 77 factor)^{14,15}. Taken together, these findings support that many ATAC-seq and H3K27ac ChIP-seq peaks overlap at enhancers. We have modified the text on in Results (Lines 201-212 and 268-271), and Methods (Lines 1467-1474) to include this information.

5. Line 146, the authors showed similar eGene discovery rates between their study and GTEx. Can the authors also show the overlapping rate of their mapped eQTL-eGene pairs with those from GTEx?

To address the reviewer's comment, we have performed eQTL colocalization between the three iPSCORE tissues (iPSC, CVPC, and PPC) and the following five relevant GTEx tissues: Artery Aorta, Artery Coronary, Heart Left Ventricle, Heart Atrial Appendage, and Pancreas. While there are no equivalent adult tissues for iPSCs, we included iPSC eQTLs in this analysis to examine their overlap with the 5 GTEx tissues.

Across the comparisons, 60-83% of eGenes showed overlapping signals between developmental and adult tissues, while 17-40% had distinct signals (see figure below). This suggests that there is substantial overlap in the genetic effects on gene expression across the two stages but that a significant portion is distinct. We have included the colocalization results below.

To address Reviewer 4 Comment 7, we used a different method (mashr; Results Lines 216-250; Methods Lines 1475-1524) to identify EDev-specific eQTLs. Due the redundancy of these two analyses, we decided to omit the iPSCORE-GTEx colocalization in the manuscript and included the mashr approach.

Legend: Bar plot showing the percentage of eGenes (x-axis) that have distinct eQTLs (gold, PP.H3 > 80%, model for two signals having distinct associations or shared eQTLs (blue, PP.H4 > 80%, model for two signals having shared associations) between the iPSCORE tissue (separated by panel) and the adult GTEx tissue (y-axis). We report the total number of eGenes with either an H3 or H4 association > 80% to the right

of each bar. eGenes with H0, H1, or H2 were excluded from this analysis due to low power.

6. Line 193, can the authors briefly discuss the TFs identified in caPeaks that overlapped haPeaks and those that did not overlap haPeaks? Which developmental stages do these TFs regulate, e.g., during early development or after cell differentiation?

We thank the reviewer for their request. To examine the stage-specificity of TFs enriched in the overlapping caPeaks and haPeaks, caPeaks alone, or non-caPeaks, we first downloaded the TPM gene expression from 1,533 RNA-seq samples for four adult cardiac tissues (aorta, atrial appendage, coronary artery, and left ventricle) from the GTEx Consortium (v8)¹⁶. We compared the expression of the 16 TFs that exhibited a significant enrichment in one of the three categories from Figure 2f between the 178 CVPCs and the 1,533 adult cardiac tissues to evaluate whether they exhibited developmental stage specificity (see Figure below). Notably, GATA4, which is a known regulator of developmental processes for many tissues^{17,18} is expressed at much higher levels in CVPCs compared to the adult cardiac tissues (first row, fourth column). Additionally, TEAD4, is expressed at ~2.5-fold higher in CVPCs compared to adult cardiac tissues (third row, fourth column). TEAD4 is a key regulator of Hippo signaling and has recently garnered a lot of attention in the field of cellular regeneration¹⁹. The MEF TF family are known regulators of cardiac processes^{15,20} and we observed that MEF2B is more highly expressed in adult arterial tissues (aorta and coronary artery) and lowly expressed (mean TPM = 1.4) in CVPCs. While we observe differences in gene expression between CVPCs and adult cardiac tissues, there is no pattern in developmental specificity across overlapping caPeaks and haPeaks, caPeaks only, and non caPeaks. For this reason, we did not include this analysis in the revised manuscript.

Legend: Boxplots of gene expression (TPM) for the transcription factors with binding sites enriched in overlapping regions of caPeaks and haPeaks ("yellow"), caPeaks only ("light blue"), and ATAC-seq peaks without a caQTL ("red"). The x-axis contains the tissue (CVPCs from the iPSCORE Collection, and adult aorta, coronary artery, atrial appendage, and left ventricle from the GTEx Consortium16), the y-axis corresponds to the gene expression (TPM), and each panel corresponds to the transcription factor reported in the label.

7. Line 231, 18% of QTL signals are associated with QTL modules, while 82% are singleton QTLs. Can the authors further characterize the 82% singleton QTLs? For example, are the singleton eQTLs enriched in promoters more than modular eQTLs? Since the chromatin state of promoters may be less likely affected by genetic variation, could it be possible that only singleton eQTLs but no caQTL/haQTL are enriched in promoters? Or are these singleton eQTLs enriched in exonic, UTR, etc.? What are the distance distributions between singleton eQTLs and TSS of eGenes compared to modular eQTLs? Additionally, are the singleton caQTLs and haQTLs enriched in the non-overlapped snATAC-peaks and H3K27ac ChIP-seq peaks, as the non-overlapped peaks may be primed/poised enhancers?

We thank the reviewer for their questions. We have re-performed the QTL module (now referred to as complex QTLs) generation using a different approach in response Reviewer #3 Comment #8. To address this reviewer's comment, we evaluated whether complex and singleton QTLs had different distance distributions from their corresponding qElement. We calculated the minimum distance between the lead variants and the TSS of the nearest expressed gene. We observed that complex QTLs were closer to the TSS of the nearest expressed gene compared to singleton QTLs across all three phenotypes (Mann Whitney U test, eQTL P-value = 3.5×10^{-27} ; caQTL P-value = 2.4×10^{-44} ; haQTL P-value = 9.2×10^{-13} ; Figure 4e). These findings show that complex QTLs are closer to the promoters, and that singletons affect more distal regulatory elements (i.e. enhancers).

Legend: Overlaid histogram showing the different distributions of the distance between the lead variant and their corresponding qElements between complex and singleton QTLs for the three molecular phenotypes in CVPCs. The x-axis is the minimum distance between the lead variant and their corresponding qElements in kilobases, the y-axis is the log₁₀ of the number of QTLs, and the bars are filled by Membership (complex QTLs = "dark orange" and singletons QTLs = "light orange"). For plot legibility, the maximum distance for eQTLs was set to 1Mb and the maximum distance for chromatin QTLs was set to 100kb.

8. Some functional validation of the findings from the study would be useful to further strengthen the manuscript. For example, it would be informative to conduct a reporter assay on the top candidate to show that *cvpc_atac_peak_233871* is an enhancer in CVPC, or/and *ppc_atac_peak_197599* is an enhancer in PPC. Do the two SNPs disrupt any TF motifs?

Reviewer #3 Comment #12 and Reviewer #4 Comment #5 asked for us to replace the example in Figure 5 (*cvpc_atac_peak_233871*, *TBX20*, and birth weight) with another example that is more relevant to the tissue and disease. We have changed the message of this section to demonstrate that a utility of the iPScore multiomic QTLs is to prioritize GWAS-associated variants for experimental validation. To address the reviewer's question about TF motif disruption, we have performed an additional 172 analysis by intersecting the putative causal SNPs within the 99% credible sets with the TF motifs identified using TOBIAS (Figure 6b). We identified 296 GWAS loci with at least one putative causal SNPs that intersect a TF motif. We now show two GWAS loci that have likely causal SNPs with experimental validation or inferred disruption of TF binding^{21,22}. In Figure 6c-d, we show a Type 2 Diabetes locus in a *JAZF1* intron that colocalizes with PPC complex QTL 122 and in Figure S11, we show QRS duration locus in an *KLF12* intron that colocalizes with CVPC complex QTL 274. In both examples, the true causal variants disrupt TF binding and are tagged by the SNP with the highest posterior probability. By restructuring this section, we believe that we better highlight how these findings can be used guide experimental and functional characterization of disease associated loci. We have included these analyses into the main text as Figure 6 and Figure S11 (Lines 349-410). We appreciate the reviewers' request for experimental validation; however, we feel that we addressed their comment by identifying examples with experimental validation and inferred activity, and that experimental validation is beyond the scope of this manuscript.

9. Line 306, the authors found the effect sizes of EDev-unique eQTLs are lower than those of adult-shared

eQTLs. Did they also examine whether the effect sizes of EDev-unique caQTLs and haQTL are lower compared to adult-shared caQTLs and haQTL?

We thank the reviewer for the question. In response to Comment #7 from Reviewer #4, we have updated our approach to identify EDev-specific eQTLs. We now use the mashr R package²³ which applies multivariate adaptive shrinkage to determine if genetic signals are specific to different conditions (i.e. tissues, developmental stage). This approach is widely used but is limited by publicly available datasets that are used for comparisons in the model. We have extensively edited the Results to report the findings from our mashr analysis (Section: Identification and functional characterization of early developmental-specific QTLs) for eQTLs where we used 47 GTEx tissues to represent adult-specific tissues. Unfortunately, due to the dearth of public caQTL and haQTL datasets representing a variety of adult tissues, we had to limit our analysis to eQTLs, and we could not specifically address the question about EDev-unique caQTLs and haQTLs.

10. Figure 4a and Line 237, since the QTLs were mapped in IPSCORE cell lines that are more likely to reveal the genetic variant effects on gene regulation during early development, the authors should also consider the GWAS for developmental diseases related to the derived iPSC cell lines. For example, as they mapped QTLs in iPSC-derived cardiovascular progenitor cells (CVPC), could they also fine-map GWAS of congenital heart disease, congenital anomaly of the arteries, etc.?

Thank you for the suggestion. We attempted to include congenital heart abnormalities and septal defects in the analysis; however, there were very few significant GWAS loci (one and two loci, respectively), so we were unable to observe any colocalizations with IPSCORE QTLs. When we relaxed the GWAS significance threshold (from GWAS p-value < 5×10^{-8} to p-value < 5×10^{-5}), we identified one GWAS locus for septal defects that had strong evidence of colocalization with a CVPC caQTL (cvpc_atac_peak_45679, H4 = 91%) but is borderline significant (see figure below). Due to limited colocalization and significant GWAS loci, we have decided to not include these two traits in the updated manuscript.

Legend: Manhattan plot showing colocalization of a CVPC caQTL for cvpc_atac_peak_45679 with a GWAS locus associated with cardiac shunt / heart septal defects. Red horizontal line indicates p-value threshold of 5×10^{-8} . Blue horizontal line indicates p-value threshold of 5×10^{-5} .

11. Line 328, the authors should modify the title and conclusion of this section to "Early developmental unique QTLs are depleted for colocalizing with Adult GWAS loci".

We thank the reviewer for this suggestion. In response to Reviewer #4 Comment #7, we applied mashr to identify EDev-specific eQTLs which was more stringent than the previous approach using LD and yielded fewer EDev-specific eQTLs and was not applicable to chromatin QTLs. As a result, we only observe 13 GWAS loci that colocalized with EDev-specific eQTLs. Given these results we decided to rename the section to "Early developmental-specific eQTLs explain a small fraction of GWAS loci".

Minor comments:

1. Gene names in both text and figures should be italicized.

We have implemented these changes. See Lines 383, 385, 394, 397, 404 and Figure 6c.

2. Inconsistent WGS sequence depth mentioned in the method part. Line 612 mentions 50x WGS, but line 761 states 30x depth of coverage.

We thank the reviewer for identifying this discrepancy. WGS was sequenced at an average coverage of 50X, and we have updated the text on Line 1103.

3. On line 669, there appears to be a missing closing parenthesis.

We have implemented this change. See Line 1006.

4. The font sizes within figures lack consistency; ensuring uniformity at least within individual figures would enhance their readability and aesthetic appeal.

We have implemented these changes.

5. In Figure 3, panel d is missing a color key.

We have added a color key for Figure 3d.

Reviewer #2

In their paper, Arthur and colleagues describe results of an important study evaluating potential explanations for the smaller than expected proportion of GWAS loci that colocalize with eQTLs. They specifically evaluated two complementary theories: that causal GWAS variants regulate the expression of genes only expressed during development (so will be missed in eQTL studies in adult tissues - i.e., GTEx) and/or that causal GWAS variants do not have direct effects on gene regulation (and visible as eQTLs) but rather affect the regulation of genes through their effect on chromatin accessibility and/or histone modifications. They perform these studies in the iPSC Omics Resource, using both previously published data and new data reported here for the first time, focusing on gene expression (eQTLs), chromatin accessibility (caQTLs), and H3K27ac (haQTLs) in undifferentiated iPSCs (to reflect early development cells) and those differentiated into cardiovascular progenitor cells and pancreatic progenitor cells (to reflect adult cells), as well as comparisons to eQTL data from GTEx (many adult tissues). They presented the concept of QTL modules and qElements that integrated across QTL types and revealed many important features about the gene regulatory landscapes associated with GWAS loci. Overall, this is an excellent study with an extremely rigorous design that revealed two important take-home messages, in addition to many other valuable insights. First, QTLs that are unique to developing tissues (iPSCs) are depleted in GWAS, thereby debunking the suggestion that the smaller than expected proportion of GWAS loci that colocalize with eQTLs are due to the paucity of development-specific eQTLs used in these studies. Second, QTL modules, especially those with caQTLs and haQTLs, colocalize with GWAS loci nearly twice as often as eQTLs alone, highlighting the importance of including these types of QTLs in post-GWAS colocalization studies.

This paper is very clearly written and easy to read; figures and legends are excellent. I have only minor comments that may help improve the clarity of a few points.

We thank the reviewer for their feedback on our manuscript. Their suggestions and comments have greatly improved the clarity of our results. Below, we outline the adjustments made in light of their feedback.

Minor Comments

1. The term "conditional" is introduced early in the paper but I could not find it defined outside of the methods. I had assumed these referred to 3 different "conditions" that the cells were subjected to, and it wasn't until I poured through the methods that I saw this was "statistical conditional analysis" to identify up to 3 secondary independent QTLs. It would be helpful to say this when the term is introduced on line 142.

We thank the reviewer for their comment. We agree that a more detailed description of conditional QTLs is required. We have described conditional QTLs in the Results section (Lines 155-157).

2. It is not clear to me why conditional QTLs would "reflect greater uncertainty in the posterior probabilities" (lines 169-171). Is this because they have smaller effect sizes?

We thank the reviewer's request for clarification. In the original version of the manuscript, we observed significantly fewer fine-mapped variants that have a posterior probability greater than 1% in the conditional QTLs, suggesting that it is difficult to assign a lead causal variant. The conditional QTLs had smaller effect sizes and this likely contributed to the fewer fine-mapped variants. In response to Reviewer #3 Comment #10, we removed this analysis from the manuscript and implemented a QTL filtering step that found ~50% of the conditional QTLs were in high D' with the primary QTL, suggesting that they are not truly independent. Based on this observation, we no longer focus on the conditional QTLs in the main manuscript. We have extensively described our new approach for identifying conditional QTLs in the Results (Lines 157-158), Supplemental Notes (Lines 656-744), and Methods (Lines 1397-1438).

3. It might be worth mentioning that the 17 traits from GWAS were selected to reflect cardio-metabolic traits (or something like that) (line 238), which fits with the cells being studied, but is different from the GWAS traits usually selected for such studies.

Thank you for the recommendation. As the reviewer points out, we selected traits that were relevant to the iPSCs, CVPCs, and PPCs and have made this point clearer. To respond to a similar comment from Reviewer #4 Comment #3, we performed LD Score Regression on open chromatin regions in the three iPSCORE tissues and observed cell type-specific enrichments of GWAS variants. We have included these results in the revised manuscript in the Results (Line 287-293), Figure S9, and Methods (Lines 1561-1573). In the revised manuscript we analyze 15 GWAS traits.

Legend: Heatmap showing the enrichment of GWAS variants in ATAC-seq and H3K27ac ChIP-seq peaks from all three tissues. Enrichment (the ratio of the proportion of heritability explained by the annotation and the proportion of SNPs in the annotation) of GWAS variants in the peaks was calculated using LDscore regression²⁴. The y-axis corresponds to the 15 summary statistics and the x-axis corresponds to the five tested peak sets. Each cell is filled with the LDscore regression enrichment. For plot legibility, the maximum enrichment was set to 20 and non-significant (P-value > 0.01) tests were filled white.

4. I couldn't find anywhere in the paper why H3K27ac was not performed in the PPCs? I may have missed it in the methods, but it would be helpful to mention this.

The H3K27ac was not performed in the PPCs primarily due to funding and time constraints. We have the frozen PPCs, and this would be an interesting dataset to add in the future.

Reviewer #3

The paper asks if developmental QTLs, as approximated from iPSC and iPSC derived tissues, might colocalize with some proportion of GWAS signals that are not colocalized with GTEx (adult) tissue eQTL signals. It also examines if chromatin accessibility QTL (caQTLs) or H3K27 QTL (haQTLs) might also colocalize with GWAS signals not detected by eQTLs. The paper finds evidence for enrichment of GWAS signal colocalization in ca and ha QTLs particularly in those signals that appear in at least one other QTL (ca, ha, e). It finds disenrichment in developmental QTL, as represented by iPSC and iPSC derived tissues. The paper is clearly written. There are multiple times where the tests performed assume that the input data are independent, results that are combined based on the same starting set of iPSC cells. The use of multiple omics increases interest in the results and the questions that can be asked. The uneven sample sizes across tissues and omic decreases interpretability somewhat.

We thank the reviewer for their thorough and constructive feedback on our manuscript. The reviewer's comments have provided us valuable insights that are crucial for improving both the analysis and interpretation of our findings.

In response to the reviewer's concerns about data independence and uneven sample sizes across tissues and omics types, as described below, we have undertaken a comprehensive review and revision of our methodologies and discussion sections. We appreciate the reviewer for pointing out these critical aspects, as they are essential for ensuring the robustness and reliability of our conclusions.

Below, we detail the specific changes made to address the reviewer's comments and concerns. We believe these revisions enhance the impact and scientific rigor of our paper.

1. Data availability: The data availability is summarized in the table below. The newly generated data is marked with an "*". Two of the newly generated datasets do not appear to be present in the ascension number listed in Data availability. Red indicates missing data. Please add the missing ascension numbers.

Tissue	Data type		
	RNA-seq	ATAC-seq	H3K27ac
iPSC	No ascension number	GSE203377	GSE261276*
CVPC	No ascension number	Not in GSE261276*	Not in GSE261276*
PPC	GSE182758	GSE261276*	No data generated

We agree with the reviewer on the importance of releasing our data into a public repository. After consulting with dbGaP as to how best release our data, we have decided to upload everything onto dbGaP under the accession phs000924 (see links below). dbGaP personnel strongly advised us to remove the data from GEO (GSE182758, GSE261276, GSE203377) in order to avoid redundancy in SRA (existing SRA data from GEO cannot be linked to dbGaP), therefore we have removed the raw data from these GEO accessions. We have also updated the Frazer Lab website with details about the published iPSCORE studies and their corresponding datasets (<http://frazer.ucsd.edu/ipSCORE-ipSC-collection-omics-research>).

dbGaP Study Pages

WGS (VCF file) and summary statistics – updated version is in queue for release:

https://www.ncbi.nlm.nih.gov/projects/gap/cgi-bin/study.cgi?study_id=phs001325.v5.p1

After version 6 for phs001325 is released, this will be the link: https://www.ncbi.nlm.nih.gov/projects/gap/cgi-bin/study.cgi?study_id=phs001325.v6.p1

Molecular data (FASTQ files):

https://www.ncbi.nlm.nih.gov/projects/gap/cgi-bin/study.cgi?study_id=phs000924.v5.p2

2. Code availability:

Main Figure code exists. Please add code for Supplemental Figures or label files better to reflect where the code is.

We agree that publishing the code to generate figures improves the reproducibility of the analyses and greatly appreciate the reviewer for checking on this. We have included code for the plotting of the supplemental figures (https://github.com/frazer-lab/iPSCORE_Multi-QTL_Resource).

Samples

3. The Supplemental Figure 3 Cardiac marker plots (b) shows samples with low levels of the cardiac cell markers in the cardiac cells. What criteria were used to include a cardiac cell line and why were these samples included in the analysis?

We thank the reviewer for the question. During differentiation, we utilized 388 a lactate selection which depletes non-cardiac cells. We and others have shown that CVPC lines are mostly composed of cardiomyocytes and epicardial cell types, including smooth muscle, cardiac fibroblasts, and vascular endothelial cells². The proportion of these cell types vary across CVPC lines, therefore filtering samples based on cardiomyocyte markers like NKX2-5 and TNNT would remove samples that were more epicardial-like. We note that PEER factor 1 is highly correlated with experimental measurements of cardiac troponin (%cTnT; Figure S16d-f) which shows that this cell type heterogeneity is accounted for in the QTL analyses. For these reasons, we only filtered based on technical factors, as described in the Methods (Lines 1176-1180, 1204-1209, 1235-1239).

4. How much overlap is there in the samples used for the three omics type within a cell type. For each cell type please show the Venn diagram of overlap of samples used for the 3 omics types.

See response to Comment #5 below.

5. How much overlap is there in the samples used for three cell types within an omics type? For each omics type please show the Venn diagram of overlap of samples used for the 3 cells types.

We have generated Venn diagrams showing overlap of samples for each omic data type and cell type (see figure below). We have included them as Figure S2 and described them in further detail in the corresponding Figure legend (Line 762-775). We thank the reviewer for this request, as the information provides clarity on the samples used in our analysis and improves the utility of the data.

Figure S2. Overlap of samples across molecular data types.

a) Venn diagram showing the overlap of iPSC lines (“iPSC_Line_ID” in Tables S2) across all data types. We note that only one iPSC line was used for each subject.

b) Venn diagram showing the overlap of CVPC differentiations (“UDID” in Tables S2) across all data types.

c) Venn diagram showing the overlap of PPC differentiations (“UDID” in Tables S2) across the data types.

d) Venn diagram showing the overlap of RNA-seq samples across the three tissue types based on iPSC line. For example, 79 iPSC lines from 79 subjects have RNA-seq for iPSC, CVPC, and PPC; 11 have RNA-seq for CVPC only; and 6 have RNA-seq for CVPC and PPC only.

e) Venn diagram showing the overlap of ATAC-seq samples across the three tissue types based on iPSC line. For example, 70 iPSC lines from 70 subjects have ATAC-seq for iPSC, CVPC, and PPC; 5 have ATAC-seq for CVPC only; and 6 have ATAC-seq for CVPC and PPC only.

f) Venn diagram showing the overlap of H3K27ac ChIP-seq samples between iPSC and CVPC. For example, 26 iPSC lines from 26 subjects have ChIP-seq for both iPSC and CVPC, and 71 have ChIP-seq for CVPC only.

Abstract

6. Line 34. The word “functionally annotated” is not defined, 422 does it mean colocalized?

We meant for the term “functionally annotated” to indicate GWAS loci that colocalize with at least one QTL. To improve clarity, we have defined “functionally annotated” (Lines 65 and 79).

7. Line 34. The 49% functionally annotated GWAS loci number needs context in terms of the actual % of GWAS loci that are colocalized.

Thank you for the recommendation. We agree that the sentence should be in the context of percent of all GWAS loci, not percent of those that are colocalized. Of the 5,291 GWAS loci, we found that 10.4% colocalized with at least one iPSCORE QTL. We have updated the manuscript to reflect this change in numerous locations (Abstract Line 32, Introduction Lines 86-87 and 101, Result Lines 294-305).

Major Analysis Questions

8a. Many of the tests in the paper use results across omics type and/or cell types. These assays are from

the same set of samples, which can affect statistical tests that assume samples used in each test are independent.

8b. One example is Figure 2e. The Figure 2e is combining the results across iPSC and CVPC. Many of these signals are the same signals across the three iPSC and CVPC from the same people. This means the signals are not independent of each other.

8c. Likewise this question arises for 1) colocalization of omics signals within a cell/tissue type (which assumes two independent datasets), 2) Figure 4b enrichment of colocalization of different combinations of molecular signals relative to eQTLs, and 3) Figure 7a enrichment colocalization with GWAS of adult shared eQTL compared to EDex-unique QTL, and potentially other tests. Please explain how you are accounting for the non-independence of the data in each test. How can you run the tests with (more) independent data?

We acknowledge that sample overlap and phenotype correlation may lead to false positives; however, to address the reviewer's comment we contacted several colocalization experts and received conflicting interpretations on the importance of this issue. Several published studies, including from the GTEx Consortium¹⁶, have performed colocalization using overlapping samples. A separate study co-authored by Dr. Chris Wallace, the developer of coloc, has demonstrated that the coloc package is suitable for analyzing pairs of eQTL signals from the same individuals²⁵ (Supplemental Figure 1 in manuscript). In Pickrell et al²⁶, the authors developed gwas-pw that enables colocalization of GWAS loci for correlated traits identified using overlapping samples, however this method has not previously been applied for QTL colocalization in a peer⁴⁵⁶ reviewed study and therefore not suitable. Another tool, mvSuSiE²⁷, is being developed to directly address sample overlap in GWAS and QTL studies, however, it is currently undergoing peer review. Previous multiomic QTL studies^{28–30} have used LD to determine whether two qElements share a QTL. This approach is considered less stringent than Bayesian colocalization, however, to our knowledge, it is the only approach that does not assume that the pairs are independent, therefore we implemented the LD approach in our study. Briefly, using plink³¹, we extracted the primary QTL lead variant genotypes from the EUR population in the 1000 Genomes VCF, and, for each lead variant, calculated the LD between all variants within 100 kb. We loaded all pairs of primary QTLs with lead variants in high LD ($r^2 \geq 0.8$) and all pairs of primary QTLs that share a lead variant into a network and performed Louvain clustering to identify QTL modules, using the igraph R package³². By using the LD structure from the EUR population in 1000G, the QTL modules are generated in a more independent manner, as suggested by the reviewer.

Below, we address the reviewer's specific comments on several points ⁴⁶⁷ throughout the manuscript: "8b. One example is Figure 2e. The Figure 2e is combining the results across iPSC and CVPC. Many of the signals are the same signals across the three iPSC and CVPC from the same people. This means the signals are not independent of each other."

To address the reviewer's concern about the effects of the overlap of individuals between tissues, we redid the enrichment analysis for the iPSC and CVPC QTLs independently. These analyses exhibited concordant observations with the original analysis. Since CVPCs had at least 100 samples across the molecular datasets, we included the enrichment analysis for the CVPCs as a panel in the main figure (Figure 2e, right panel below), and the iPSC enrichments in a supplemental figure (Figure S8, left panel below). We note that the QTLs from the different datasets from each tissue are not being directly compared, therefore the enrichment is not affected by sample overlap.

In Comment #8c, the reviewer highlights three examples that may be confounded by sample overlap. Here, we address each example:

#1 "colocalization of omics signals within a cell/tissue type (which assumes two independent datasets)".

In response to this comment, we have implemented a different method for identifying QTL modules by using the LD structure from an independent dataset (1000 Genomes; as described in Results Lines 251-271 and Methods Lines 1526-1537). We believe replacing pairwise Bayesian colocalization with the LD method addresses this comment.

#2 "Figure 4b enrichment of colocalization of different combinations of molecular signals relative to eQTLs".

In response to reviewer's Comment #11, we have modified our method for colocalizing QTLs with GWAS. Specifically, we now only use a single randomly selected representative QTL in each module (referred to as complex QTLs in the current manuscript) to test for colocalization with GWAS (using coloc). If the representative QTL does not colocalize with a GWAS locus, we designate that all QTLs in the module do not colocalize in the GWAS analysis. This circumvents testing the GWAS colocalization with multiple QTLs in each module, which were identified in datasets containing overlapping samples. Since the GWAS datasets are independent of the QTLs, there is no sample overlap and therefore we continue to use coloc to determine colocalization between QTLs and GWAS.

#3 "Figure 7a enrichment colocalization with GWAS of adult shared eQTL compared to EDex-unique QTL, and potentially other tests".

As described above, we implemented the LD method for identifying QTL modules (now referred to as complex QTLs) and then randomly selected a representative QTL for each QTL module. We tested the colocalization between a GWAS locus and a single QTL, either the representative QTL for each module or a singleton QTL, therefore the current enrichment analysis is not confounded by sample overlap. Reviewer

#4 Comment #7 suggested that there was an ascertainment bias for the EDev-unique QTLs, which led us to modify how we identify EDev-unique (now EDev-specific) eQTLs.

We hope that the Reviewer agrees with us that the modifications and new analyses that we have conducted are not confounded by sample overlap.

9. Line 1058 What QTL data is used for colocalization of the primary signal? Does primary mean the ie marginal eQTL signal, before conditioning? Typically the primary signal is the strongest marginal signal adjusted for the other conditional variants (see below).

We address this comment in response to the following comment.

10. Line 1054 and 1058: The conditional analysis is described as a sequential identification of QTL signals. When colocalizing with GWAS signals, is there a subsequent step to produce the distinct association signals for each distinct SNP by adjusting for all other distinct SNPs. For example, if three conditionally distinct SNPs were identified are three separate regressions run for all SNPs in the 1MB region SNPs adjusting for SNP2 and 3 (for signal 1), SNP 1 and 3 (for signal 2), and SNP1 and 2 (for signal 3) This is the convention and is related to what Bayesian methods like SuSiE would do. The first signal, with adjustment for other signals, would be called the primary signal.

Based on the reviewer's comments and concerns, we have extensively updated our QTL pipeline and described the updates in Supplemental Notes 1 and 2 (Lines 656-744) and the Methods (Lines 1359-1438). Briefly, the QTL pipeline now consists of two steps; 1) QTL Discovery, and 2) QTL Filtering. The QTL Discovery step is unchanged and calculates up to four association signals (primary and three conditional) for a given qElement and reports the lead variant for each signal. The QTL Filtering step identifies and removes conditional QTLs that are not truly independent (see below). Finally, prior to performing QTL GWAS colocalization, we perform Primary Signal Regression step, which regresses out the effects of conditional signals from the primary signal. The QTL Filtering step and Primary Signal Regression were added to address the reviewer's comment.

11. The statement that GWAS loci are highly enriched for colocalization with complex QTL molecules, may be because there is better power to detect colocalization with stronger signals (as is mentioned), not because there is inherently more colocalization. If you correct for the strength of signals do you see the same patterns of enrichments?

We thank the reviewer for this question, which is similar to Reviewer #4's Comment 530 #4. We have modified the GWAS enrichment analysis by using LD between lead variants to construct the QTL modules (now referred to as complex QTLs). Compared to Bayesian colocalization, LD is less dependent on effect size, therefore decreasing the differences between complex and singleton QTLs. We randomly selected a single representative QTL for each complex QTL, therefore only one QTL was used to determine whether a module colocalized with a GWAS locus. We found that complex QTLs affecting all three phenotypes (caQTLs, haQTLs, and eQTLs) were the most enriched, and complex QTLs affecting caQTLs and haQTLs and affecting caQTLs and eQTLs, as well as singleton eQTLs (Figure 5e) were also enriched. While we believe this adjustment improved our analysis, we also acknowledge that power biases may still influence our

observation and have included this interpretation in the Discussion (Lines 446-449). We thank the reviewer for the suggestion.

Legend: Plot showing the relative enrichment of GWAS colocalization by complex QTL and singleton composition. We categorized each complex and singleton QTL based on their associated molecular phenotype(s). Two-sided Fisher’s Exact Tests were performed to test the relative enrichment (odds ratio) of each QTL category for GWAS colocalization compared to all other categories. In the first three rows of the x-axis, the black circles indicate the QTL composition categories (i.e. QTLs affecting different combinations of qElements). In the last two rows of the x-axis, red circles indicate the QTL membership (i.e. Complex or Singletons). The y-axis is the log2(odds ratio) enrichment. Tests that had P-value < 0.05 were considered significant (colored in black).

Results:

12. Line 265: The inclusion of the TF TBX20 example seems somewhat random. Why was there a focus on this locus in particular?

We appreciate the reviewer’s input. We have addressed a similar question asked by Reviewer #4 Comment #5. Although we initially found this locus to be interesting because there is an epidemiological observation that low birth weight is associated with an increased risk of cardiovascular disease, we have replaced it with two examples of published causal GWAS variants that highlight the strongest utility of multiomic QTLs which, we believe is informing experimental validation. To better highlight this utility, we have performed an additional analysis by intersecting the putative causal SNPs within the 99% credible sets with the TF motifs identified using TOBIAS (Figure 6b). We identified 296 GWAS loci with at least one putative causal SNPs that intersect a TF motif. In Figure 6c, we show a experimentally validated motif-overlapping putative causal variant (MOPCV) in a Type 2 Diabetes locus with experimental validation²¹ in 562 a JAZF1 intron that colocalizes with PPC complex QTL 122 and in Figure S11, we show a MOPCVs with inferred TF disrupting activity in a QRS duration locus in an KLF12 intron that colocalizes with CVPC complex QTL 274. In both examples, the true causal variants disrupt TF binding^{21,22} and are tagged by the SNP with the highest posterior probability. By restructuring this section, we believe that we better highlight how these findings can be used guide experimental and functional characterization of disease-associated loci. We believe that these revisions highlight how multiomic QTLs and complex QTLs (modules) can be used as a resource for experimental validation and functional characterization of disease-associated loci.

Discussion:

13. Line 371. There are reasons in addition to the two reasons listed that could explain why there is not complete overlap of eQTLs and GWAS loci: The relevant QTLs could be present in rarer cell types, and thus not show up on bulk data, and relatedly, sample sizes of bulk or single cell/nucleus data may be insufficient to detect the weaker eQTLs.

We agree with the reviewer's interpretation of the incomplete overlap of QTLs. We have edited the Introduction (Line 48) to include "rare cell types" as a context-specific QTL type and the Discussion (Lines 416-417) to suggest "that there are several leading hypotheses", and list "rare cell types".

14. Address the limitations that there are not equivalent sample sizes for each tissue/cell type and thus it's not easy to compare across them and also that the developmental models may not fully capture the QTLs present in living developing tissue.

We agree and now discuss the limitations of comparing datasets with different samples sizes in the Discussion (Lines 438-440) and Methods (Lines 1498-1499).

Figure 1

15. It is hard to see and compare the number of samples for the three tissue and three molecular data types. Including a 3x3 matrix with the numbers of samples in each cell could make this easier to read than the current format.

We thank the reviewer for the recommendation. We have included a table in Figure 1 that reports the number of samples in each dataset.

16. Please make a supplemental table that has the number of unrelated individuals for each tissue and omic type to help people judge the amount of data, or give an estimate of the effective number of people analyzed for each tissue and omics type.

We have included a "Unrelated_Set" column in each of the Table S2 sheets (RNA-seq, ATAC-seq, and ChIP-seq) that indicate the optimal set of samples, including unrelated individuals within families, for each of the eight datasets.

Figure 2

17. Figure 2a: Indicate which of the tests are significant with a star or another demarcation in boxes. Please show the Shared signals for each cell type.

Thank you for this suggestion. We calculate the enrichment of TFBSs in tissue-specific peaks using the shared peaks as background, therefore the enrichments are complementary (see below). For this reason, we have decided to exclude the TFBS enrichments in the "Shared" peaks in Figure 2a. We now indicate in the figure which tests are significant with an asterisk. We have also edited the Figure legend text (Lines 540-541) to indicate that TFBSs that are depleted in Tissue-Specific ATAC-seq peaks (i.e., E2F TFs, CTCF, EGR1), are enriched in "Shared" peaks.

Legend: Heatmap showing TFBS enrichments specific and shared ATAC-seq peaks for iPSCs, CVPCs, and PPCs. Two-sided Fisher's Exact tests were performed to test the enrichment (odds ratio) of the predicted TFBSs in each of the six ATAC-seq peak sets. Each cell is filled with the log₂(Odds Ratio) of the association between predicted TFBSs (y-axis) and tissue-specific and shared ATAC-seq peaks (x-axis).

18. Figure 2f: Please explicitly state the comparison groups used for each of the 611 three enrichment tests, in the Figure and in the Methods section.

We have added in the Figure 2f Legend and in the Methods (Lines 1467-1474) a description of the comparison groups for each enrichment.

19. Figure 2f Consider renaming the tests so they are distinctive without reading the legend.

Thank you for this suggestion. We have renamed the axis to further clarify the tests.

20. Figure 2f. White color is used in the color scale to mean an OR of 1. The color is also set to white if the test is not significant. This is mixing two meanings. It could be clearer to use the log₂(OR) color as the effect size (as expected from color scale) and indicate the significance with a star or other method in the appropriate boxes.

We agree with the reviewer that the white color has two meanings and could confuse the readers. We have modified Figure 2f to use log₂(OR) to color the boxes and to use an asterisk to indicate significance.

Figure 3.

21. Figure 3b. The pie chart would work (best) if there were an equal number of samples in each tissue and omics type. They are not equal and thus the proportions don't have an interpretable meaning. Is there another way to show this data?

We thank the reviewer for this comment. We have re-created the figure (now Figure 4a) to be a bar plot to better represent the data.

Figure 4

22. Figure 4a. The scale of the bar plot for the total number of signals across all GWAS doesn't match the scale of the individual GWAS, and thus makes it hard to interpret the figure. The total sample bar could be shown on the original scale and then be expanded if needed for clarity.

Thank you very much for this comment. In Figure 4a (now Figure 5a-b), we have modified the scale for the total number of signals across all GWAS to match the scale for the individual GWAS.

Figure 5

23. Figure 5a 7c-e LD pictures. The LD colors scale changes between the panels which makes it hard to track the LD patterns for the same reference variant. Please make the LD color scale the same for each plot, here and in any figure showing LD.

Thank you for this suggestion. We have removed Figure 7 and used the same color scheme for all panels in Figure 6.

Figure 7

24. Figure 7e. The legend says this plot shows Adult eQTL, caQTL and 642 haQTLs, however the points are not labeled as coming from any one of these specifically. Please clarify what is shown in the plot.

We have removed Figure 7e from the manuscript.

Methods:

25. Line 652: Clarify here that samples are from D0 so are (presumably) iPSC cells.

Thank you for this comment. The cell state of naive and D0 iPSCs differs slightly. For the iPSC RNA-seq, data was generated on conventionally cultured naive iPSCs. ATAC-seq and H3K27ac ChIP-seq was generated from D0 iPSC, which corresponds to the start of CVPC differentiation, where iPSCs were treated with Rho kinase (ROCK) inhibitor for 24 hours, but not treated with the compounds to initiate the cardiac differentiation. We have described this in more detail in the Methods (Lines 1046-1048).

26. Line 776 and others: There are substantial differences in the sequencing depths for the molecular phenotypes across the tissues. Consider a table containing this information and commenting on the effects of differential sequencing depth in the discussion on the discovery of QTLs.

We thank the reviewer for this comment. We have included sequencing information in Supplemental Table 2 for the RNA-seq, ATAC-seq, and H3K27ac ChIP-seq samples. We have also included text in the Methods to highlight the effects of differential sequencing depth (Lines 1444-1445).

27. Line 818: Why was the WGS data phased?

Thank you for catching this comment. The WGS was phased for a different project. The phasing was not used for this study; therefore we removed this detail from the Methods (Line 1160).

28. Line 972: There may be a typo as the text says the OR's were added to get the average: For example, the average of $OR = .5$ and $OR = 2$, is $OR = 1$, which is the result when adding on the \log_2 (OR) scale not on the OR scale.

In response to the reviewer's comment #17, we have removed the "Shared" column from the plot, as TFBSs that are depleted in tissue-specific are enriched in "Shared" peaks. We have also removed the sentence from Line 972. We thank the reviewer for pointing out this issue.

29. Line 1072: When testing for enrichment of caPeaks within a 100kb window around any gene, the larger the gene, the more caPeaks that will map to the gene by chance. If genes that take up smaller chromosomal regions are less likely to have an eQTL then there could be a bias. Please describe the test used for enrichment and verify that the length of the genomic region around a gene is not biasing the test ($\log(\text{genomic region})$ could be included as a covariate (spline)).

We thank the reviewer for their comment. We have refined this analysis to account for gene length by only considering regions 100kb upstream of any expressed gene. For each tissue, we annotated the 100kb upstream window based on whether it overlapped a caPeak. We then performed Fisher's Exact tests to evaluate the enrichment of caPeaks in windows upstream of eGenes, using genes without an eQTL as

background. In all tissues, caPeaks were enriched upstream of eGenes (iPSC Odds Ratio = 1.3, P-value = 1.1×10^{-16} ; CVPC Odds Ratio = 1.3, P-value = 2.6×10^{-17} ; PPC Odds Ratio = 677 1.4, P-value = 4.4×10^{-26}), compared to genes without eQTLs. By fixing the window to 100kb for all genes, regardless of the length of the gene body, we have addressed the reviewer's comment about potential biases arising from the original approach. We have edited the Results (Lines 169-176) to better reflect the findings from the updated approach.

30 Line 1076: The text in this section is a result. Please add to the results.

We thank the reviewer for the suggestion. We have included the observation that wider ATAC-seq peaks are more likely to have caQTLs than shorter peaks in the Results (Lines 167-168).

31. Line 1148: There is no section 6.1.5 LD Score regression. Does this mean section 6.5.1 GWAS traits?

Thank you for identifying this discrepancy. We had performed LDscore regression in an early version of the manuscript and removed it to shorten the manuscript. We have now added it back to address Reviewer 2 Comment 3 and Reviewer 4 Comment 3. LDscore regression results can be found in the Result (Lines 289-689 290), Figure S9, and Methods (Lines 1561-1573).

32 Line 1171: An r^2 criterion of $>.1$ is not a convincing threshold for independent GWAS signals. Is there a distance criteria (how far away does 500 SNPs get you). Change or talk about limitation of that criteria. In our original manuscript we used 500 variants, a step size of 50 and $r^2 > 0.1$. To address the reviewer's comment, we reviewed multiple studies to understand what is standard in the field^{31,33-36}, and determined that many use variant count as the window size criteria. We calculated the distance spanned by random sets of 500 variants and found that the median window was 330 kb. We also tested various values for variant count, step size, and r^2 , and determined that using a window size of 500 variants, step size of 5, and $r^2 > 0.1$ yielded similar numbers of independent signals as reported in the original GWAS studies. For example, the GWAS study for type 2 diabetes³⁷ (Mahajan et al., 2018) reported 403 distinct association signals, and in our study, we also identified 403 signals. Similarly, for the GWAS birth weight study³⁸, the authors reported 60 association signals, and we identified a similar number (65) signals. We appreciate the reviewer's input and have modified the step size value to 5 and added additional text in Methods to explain the rationale for choosing these parameters (Lines 1606-1629).

Reviewer #4

Arthur, Nguyen et al. assayed three types of molecular QTLs, for expression (eQTLs), chromatin accessibility (caQTLs), and histone acetylation (haQTLs) in three different developmental tissues: iPSCs and their differentiation into cardiovascular progenitor cells (CVPCs) and pancreatic progenitor cells (PPCs). They first performed various analyses characterizing the properties of these eQTL types, e.g., their enrichment in different regulatory annotations and the patterns of QTL sharing. Then, they evaluated the colocalization of these molecular QTLs with GWAS hits.

I think the effort to catalog these QTL types in development-like tissues is useful, particularly given one prevailing hypothesis about the missing GWAS colocalizations, which suggests that trait variants may be context-specific and missed in conventional QTL assays. However, my assessment is that the authors could do more characterizations of these QTLs in relation to GWAS variants. Also, I have major concerns about many of the data interpretations and conclusions. That said, I think these can be addressed.

We thank the reviewer for their very insightful and constructive feedback. Their comments have been instrumental in refining our analysis designs and improving the interpretation of our results.

We agree with their concerns regarding the need for more extensive characterization of the QTLs in relation to GWAS variants, and the issues raised about our interpretations and conclusions. In response, we have carefully reviewed all our analyses, extensively revised and re-performed a major analysis, and changed nomenclature according to the reviewer's recommendations.

Below, we outline the specific amendments made to the manuscript.

Major comments:

1. As motivation, in the introduction, the authors mention a recent study showing systematic differences in the discovery of eQTLs and GWAS hits (Mostafavi et al.). For the search for missing GWAS colocalizations, it would be useful to characterize: (i) whether/how adult-tissue QTLs are different from development-like tissue QTLs and (ii) how different QTL types compare to each other and control variants in terms of the properties of the tentative target genes (e.g., selective constraint), distance to genes, etc. My expectation is that the discovered QTLs are biased away from GWAS variants in important ways

(though not exactly like eQTLs), considering the small sample sizes used in the study. [I note that target genes are not known for caQTLs and haQTLs, for which perhaps the nearest gene can serve as a proxy.]

We thank the reviewer for their request to further characterize the differences between (1) adult-shared and developmental QTLs and (2) different QTL molecular types.

Several of the reviewer's comments are related to the discovery and characterization of EDev-unique QTLs. To address these comments, we have modified the analysis to use mashr²³. mashr is a widely used approach to identify context-specific genetic effects, and because the method uses effect size and standard error as input, it has more power to detect eQTLs compared to other approaches²³. mashr also requires independent QTL datasets from different tissues or developmental stages. As chromatin QTL datasets are limited, we have focused on developmental stage-specific eQTLs. Mashr is more stringent than the LD approach we used in the original manuscript, and yielded fewer EDev-specific eQTLs. The inability to calculate EDev-specific chromatin QTLs and the fewer EDev-specific eQTLs discovered by mashr, affected our ability to test the evolutionary constraint hypothesis posited by Mostafavi³⁹. First, we will describe the approach at large and then address the reviewer's Comment #1.

Mashr Approach

Using the eQTL analysis vignette (https://stephenslab.github.io/mashr/articles/eQTL_outline.html), we applied mashr to all lead eQTLs in the 3 iPSCORE tissues and 47 GTEx tissues. First, we estimated the correlation structure in the null tests by testing a random set of 200,000 variant-gene pairs in 50% of the 50 tissues. Then, we calculated 1) data-driven covariances using the "top" eQTL for each eGene based on maximum absolute beta effect size across all 50 tissues, and 2) canonical covariances using the 200,000 random tests. Finally, we fitted the model using the covariances above and computed the local false sign rates (LFSR) for each lead eQTL in each tissue. LFSR is used to measure eQTL activity. An eQTL in a tissue with LFSR < 0.05 was considered to be significantly active. We have detailed the Methods in Lines 1475-1524 in the updated manuscript.

Characterizing differences between EDev-Specific, Adult-Specific, and Shared eQTLs

To identify temporal and shared eQTLs, we first filtered variants that were not significant (LFSR > 0.05) and not present and tested in both iPSCORE and GTEx tissues, resulting in 102,375 eQTL SNP-eGene pairs. We classified 27,881 adult-specific eQTLs (LFSR < 0.05 in at least one GTEx tissue and LFSR > 0.05 in all iPSCORE tissues), 2,299 EDev-specific eQTLs (LFSR < 0.05 in at least one iPSCORE tissue and lfsr > 0.05 in all GTEx tissues), and 72,195 Shared eQTLs (LFSR < 0.05 in at least one iPSCORE tissue and one GTEx tissues).

Mashr was more stringent than using the LD approach in the original submission and yielded 13 GWAS loci associated with an EDev-specific eQTL, therefore we were underpowered to assess differences between colocalized and not colocalized EDev-specific eQTLs.

Characterizing differences between eQTLs, caQTLs, and haQTLs

As recommended by the reviewer, we further characterized the differences between eQTLs, caQTLs, and haQTLs, by assigning each of the 60,306 QTLs a target gene based on the nearest TSS to the lead variant.

We evaluated the differences in distance to the target gene between eQTLs and chromatin QTLs (caQTL and haQTLs) that colocalize with a GWAS locus. Since eQTLs have known target genes but chromatin QTLs do not, we annotated all colocalized QTLs with the closest protein coding gene, using bedtools, regardless of their original eQTL-eGene association. We performed Mann Whitney U tests to test if GWAS colocalized eQTLs, caQTLs, and haQTLs had different distributions. We found that GWAS-colocalized caQTLs were more distal than GWAS-colocalized eQTLs across all 3 tissues (iPSC P-value = 1.1×10^{-5} ; CVPC P-value = 0.008; PPC P-value = 6.5×10^{-5}). In CVPCs, haQTLs did not have a significantly different distribution than eQTLs (P-value = 0.08) or caQTLs (P-value = 0.48). These findings suggest that eQTLs that colocalize with GWAS tend to affect promoters, which supports previous observations^{16,39}, while GWAS colocalized caQTLs capture more distal elements.

In response to the Reviewer's Comment 6, we performed a similar analysis that has been incorporated in the Results (Figure 5c-d) and described in detail in Methods (Lines 1630-1640). We thank the reviewer for the suggestion.

797

Legend: Box plot showing the distance between GWAS-colocalized QTL lead variants and their nearest TSS. The x-axis is the tissue, the y-axis is the distance to the nearest TSS (in kilobases), and the boxes are filled by the three molecular QTL types (eQTLs = orange; caQTLs = brown; haQTLs = light blue).

2. I did not understand what "QTL modules" are and what the motivation for them is.

We thank the reviewer for their comment and agree that we need to better define and explain the rationale for identifying QTL modules. To clarify the utility of integrating multiomic QTLs, we change the nomenclature from "QTL modules" to "complex QTLs". We believe that "complex QTLs" is a more accurate description and clarifies that they represent a single QTL that affects multiple elements (multiple genes and/or peaks). Here are a couple of potential mechanisms that can explain the phenomenon; 1) a locus affecting a gene and an associated cis regulatory element, 2) the disruption of a TF binding site that is a cis-regulatory element for multiple genes⁴⁰, 3) the disruption of a CTCF binding sites at topological associated domain (TAD) boundaries⁴¹, and 4) the overlap of a caPeak and haPeak that capture the same TF binding site.

Identifying complex QTLs that affect multiple molecular elements is an intuitive way to integrate QTLs from intermediate phenotypes, like chromatin accessibility and H3K27 histone acetylation, that can help functionally characterize mechanisms underlying eQTLs. Additionally, in our manuscript, we sought to determine whether QTL complexity, as defined by QTLs that affect multiple and/or different combinations of qElements (eGene, caPeaks, and haPeaks), was associated with higher colocalization with GWAS. We note that the method for identifying complex QTLs has changed in response to Reviewer #3's Comment #8. We explain these changes in more detail in response your Comment #4 below.

3. How were these 17 traits chosen? While plausibly true for a subset of traits, it is not obvious whether the three studied tissues are relevant trait-contexts to search for colocalizations. The authors could nominate top cell types through heritability enrichment (using s-LDSC) in open chromatin regions of these tissues compared to other tissues and cell types (e.g., in the single-cell atlas of Zhang et al., Cell 2021). [I may have missed this if authors have already reported that.]

We understand the reviewer's concern that it is important to ensure that tissues are relevant trait-contexts to conduct colocalizations with GWAS loci. While we did not include the analysis in the original manuscript, we had performed LD score regression on open chromatin regions of iPSC, PPC, and CVPC, and as expected, observed enrichment of open chromatin regions of CVPC and PPC for their respective diseases. However, one trait, type 1 diabetes, did not show enrichment. Additionally, we found that the longevity and health span GWAS were redundant because they were included in the multivariate longevity GWAS study⁴². Based on the reviewer's comment we have decided to put the analysis in the Results (Lines 289-290, Figure S9) and Methods (Lines 1561-1573) in the revised manuscript and to remove type 1 diabetes and health span as target traits.

27

Legend: Heatmap showing the enrichment of GWAS variants in ATAC-seq and H3K27ac ChIP-seq peaks from all three tissues. Enrichment (the ratio of the proportion of heritability explained by the annotation and the proportion of SNPs in the annotation) of GWAS variants in the peaks was calculated using LDscore regression²⁴. The y-axis corresponds to the 15 summary statistics and the x-axis corresponds to the five tested peak sets. Each cell is filled with the LDscore regression enrichment. For plot legibility, the maximum enrichment was set to 10 and non-significant (P-value > 0.01) tests were filled white.

4. The authors report that GWAS variants are more likely to colocalize with QTLs with large effects. This is most likely a power bias, meaning it is much easier to detect colocalization for stronger QTLs (see Hukku et al., AJHG 2021). As written, it is implied that this is an intrinsic feature of GWAS variants.

In the original manuscript, we showed that QTLs in modules (i.e., QTLs associated with multiple qElements) had larger effect sizes than singleton QTLs (i.e., QTLs associated with a single qElement). We also showed that QTL modules (now referred to as complex QTLs) were enriched for colocalizing with GWAS variants compared with singleton QTLs. We agree that this analysis could be refined to correct for technical biases and that our claims about the properties of GWAS loci could be better stated to reflect our findings. In response to this comment and Reviewer #3's Comment #11, we have modified the GWAS enrichment analysis by using LD between lead variants to construct the complex QTLs. Compared to Bayesian colocalization, LD is less dependent on effect size, therefore decreasing the differences between complex QTLs and singleton QTLs. We still observe that complex QTLs tend to be closer to promoters (which tend

to have larger effects), but the differences between complex and singleton QTLs identified with the colocalization approach are less pronounced. We randomly selected a single representative QTL in each complex QTL, therefore only one QTL was used to determine whether a complex QTL colocalized with a GWAS locus. We still observed that complex QTLs associated with chromatin accessibility and histone acetylation or gene expression were enriched for GWAS colocalization while singleton eQTLs were the least enriched, and singleton caQTLs and singleton haQTLs were depleted. These 865 data are shown below and in Figure 5e. Despite the recapitulation of our previous results, we agree that our observations are aligned with higher rates of GWAS loci colocalizing with promoter-acting QTLs, thus may still be explained by power biases. We have acknowledged these limitations in the Discussion (Lines 446-449).

Legend: Plot showing the relative enrichment of GWAS colocalization by complex QTL and singleton composition. We categorized each complex and singleton QTL based on their associated molecular phenotype(s). Two-sided Fisher’s Exact Tests were performed to test the relative enrichment (odds ratio) of each QTL category for GWAS colocalization compared to all other categories. In the first three rows of the x-axis, the black circles indicate the QTL composition categories (i.e. QTLs affecting different combinations of qElements). In the last two rows of the x-axis, red circles indicate the QTL membership (i.e. Complex or Singletons). The y-axis is the log2(odds ratio) enrichment. Tests that had P-value < 0.05 were considered significant (colored in black).

5. For the example of TBX20, it is not obvious why a CVPC QTL may causally affect birth weight. I don't think the fact that birth weight affects the risk of heart defects means that CVPC is an important cell type for birth weight, as argued by the authors. Thus, I am not sure what biological insight we gain from the colocalization presented in Figure 5.

We thank the reviewer for their input. Reviewer #3 Comment #12 asked a similar question. Although we initially found this locus to be interesting because there is an epidemiological observation that low birth weight is associated with an increased risk of cardiovascular disease, we have replaced it with two examples of published causal GWAS variants that highlight the strongest utility of multiomic QTLs which, we believe is informing experimental validation. To better highlight this utility, we have performed an additional analysis by intersecting the putative causal SNPs within the 99% credible sets with the TF motifs identified using TOBIAS (Figure 6b). We identified 296 GWAS loci with at least one putative causal SNPs that intersect a TF motif. In Figure 6c, we show a experimentally validated motif-overlapping putative causal variant (MOPCV) in a Type 2 Diabetes locus with experimental validation21 in a JAZF1 intron that colocalizes with PPC complex QTL 122 and in Figure S11, we show a MOPCVs with inferred TF disrupting activity in a QRS duration locus in an KLF12 intron that colocalizes with CVPC complex QTL 274. In both examples, the true causal variants disrupt TF binding21,22 and are tagged by the SNP with the highest posterior probability. By

restructuring this section, we believe that we better highlight how these findings can be used guide experimental and functional characterization of disease-associated loci. We believe that these revisions highlight how multiomic QTLs and complex QTLs (modules) can be used as a resource for experimental validation and functional characterization of disease-associated loci.

6. Similar to my first comment, it would be illuminating to investigate the systematic differences between GWAS hits that colocalize with QTLs versus GWAS hits without any colocalization. As the reviewer suggested, we have examined differences between GWAS hits that colocalize with QTLs versus GWAS hits without any colocalization with regard to distance to genes. Here, we used the same approach described in the response to the reviewer's first comment and annotated the indexed variant for each GWAS loci with the closest protein-coding gene, compared the distance to the nearest TSS and observed that GWAS loci that colocalize with at least one QTL are closer to a TSS compared to the GWAS loci that do not colocalize with a QTL (Figure 5d; Mann Whitney U test P-value = 1.2×10^{-11}). Further, we took an alternative approach to address the Reviewer's Comment 2, by looking at the distribution of colocalized GWAS loci to their nearest TSS by the combination of associated QTL types (Figure 5c). We observed that colocalized GWAS loci associated with an eQTL were closer to the nearest TSS, while GWAS that just colocalized with only chromatin QTLs were more distal (Figure 5c). We agree that this is an important point, and we thank the reviewer for their comment, and we have included this in the Results (Lines 306-324; Figure 5c-d), and Methods (Lines 1630-1640).

Legend: c) Box plot showing the distance to the nearest TSS for colocalized GWAS loci (n=540) by QTL types. The x-axis is the distance between the colocalized GWAS loci index and the TSS of the nearest protein-coding gene in kilobases, and the y-axis is the combination of QTLs that colocalize with a GWAS locus. For plot legibility, the maximum distance was set to 350kb.

d) Box plot showing the distance to the nearest TSS for GWAS loci by colocalization status. The x-axis is the distance between the GWAS loci index and the TSS of the nearest protein-coding gene in kilobases, and the y-axis is the GWAS loci colocalization status. The asterisks (**) indicate that there is significantly different distribution (Mann Whitney U test P-value = 1.2×10^{-11}) between GWAS loci with and without colocalization. For plot legibility, the maximum distance was set to 250kb.

7. I do not think I agree with the characterization of developmental-unique QTLs versus adult-shared QTLs, and I suspect the results in Figures 6 and 7a are a consequence of the analysis design rather than due to different intrinsic features of developmental and adult QTLs.

First, the text implies that developmental eGenes are high pLI genes. But my understanding is that this is relative to adult-shared eGenes, not random/control genes. Also, the number of eGenes used for comparison is not matched (532 versus 8,257). This is problematic because the properties of eGenes vary by their rank (or degree of significance). A proper test would be to compare constraint for the top 532 eGenes in both, as well as random/control genes.

Second, I think the trends shown in Figure 6 can be explained by power and discovery biases. Specifically, for unique and shared QTLs acting on the same gene, natural selection is stronger on the shared QTLs. Thus, discovery of shared QTLs will be more biased towards weakly selected genes. As such, I think the statement in the Discussion that "unique regulatory variants are strongly selected against" is incorrect; in fact, I think the opposite is true. The authors can test this by exploring the reverse, i.e., adult-unique versus shared QTLs, or by comparing QTLs with respect to measures such as CADD score.

Third, the interpretation of the differential colocalization of developmental-unique QTLs versus adult-shared QTLs with GWAS hits is complicated. As mentioned above, shared QTLs are likely skewed towards less GWAS-relevant genes. But this can be counterbalanced by the fact that shared QTLs have more paths to contribute to trait variations (through activity in multiple causal tissues). This is also compounded by different discovery biases mentioned above that have huge effects on colocalization analyses. For example, shared QTLs are more likely to be promoter variants, which have larger effects and are thus easier to detect in colocalization.

We thank the reviewer for their comments. As mentioned in the response above, we used a more standardized and stringent method, mashr, to identify EDev-specific eQTLs which was incompatible with chromatin QTLs. We believe that mashr addresses the reviewer's concern about discovery bias, because we require variants to be tested in iPSCORE and GTEx tissues (Methods Lines 1487-1490). We conducted the effect size comparison analysis on those variants identified by a completely different approach and observed findings that were consistent with those in the original version of our manuscript (Figure 3c); while we mention these findings in the revised manuscript they are no longer emphasized.

Due to the nature of the mashr analysis, we were underpowered to evaluate evolutionary constraint and simplified our message in the Results by stating that "Of the 239 colocalized GWAS loci associated with at least one eQTL, 5.4% (n=13) were associated with EDev-specific eQTLs, 75.7% (n=181) were associated with Shared eQTLs ..." (Lines 341-344). We note that a recent paper has observed that temporal regulatory variation active in the developing brain has smaller effects and tends to affect genes that are under higher evolutionary constraint compared to regulatory variation in the adult brain⁴³. Thus, additional studies are required to fully resolve this question.

References

1. Panopoulos, A.D., D'Antonio, M., Benaglio, P., Williams, R., Hashem, S.I., Schuldt, B.M., DeBoever, C., Arias, A.D., Garcia, M., Nelson, B.C., et al. (2017). iPSCORE: A Resource of 222 iPSC Lines Enabling Functional Characterization of Genetic Variation across a Variety of Cell Types. *Stem Cell Reports* 8, 1086–1100. <https://doi.org/10.1016/j.stemcr.2017.03.012>.
2. D'Antonio-Chronowska, A., Donovan, M.K.R., Young Greenwald, W.W., Nguyen, J.P., Fujita, K., Hashem, S., Matsui, H., Soncin, F., Parast, M., Ward, M.C., et al. (2019). Association of Human iPSC Gene Signatures and X Chromosome Dosage with Two Distinct Cardiac Differentiation Trajectories. *Stem Cell Reports* 13, 924–938. <https://doi.org/10.1016/j.stemcr.2019.09.011>.
3. Nguyen, J.P., Arthur, T.D., Fujita, K., Salgado, B.M., Donovan, M.K.R., iPSCORE Consortium, Matsui, H., Kim, J.H., D'Antonio-Chronowska, A., D'Antonio, M., et al. (2023). eQTL mapping in fetal-like pancreatic progenitor cells reveals early developmental insights into diabetes risk. *Nat Commun* 14, 6928. <https://doi.org/10.1038/s41467-023-42560-4>.
4. D'Antonio, M., Nguyen, J.P., Arthur, T.D., iPSCORE Consortium, Matsui, H., D'Antonio-Chronowska, A., and Frazer, K.A. (2023). Fine mapping spatiotemporal mechanisms of genetic variants underlying cardiac traits and disease. *Nat Commun* 14, 1132. <https://doi.org/10.1038/s41467-023-36638-2>.
5. Wang, J., Morgan, W., Saini, A., Liu, T., Lough, J., and Han, L. (2022). Single-cell transcriptomic profiling reveals specific maturation signatures in human cardiomyocytes derived from LMNB2-inactivated induced pluripotent stem cells. *Front Cell Dev Biol* 10, 895162. <https://doi.org/10.3389/fcell.2022.895162>.

6. Veres, A., Faust, A.L., Bushnell, H.L., Engquist, E.N., Kenty, J.H.-R., Harb, G., Poh, Y.-C., Sintov, E., Gürtler, M., Pagliuca, F.W., et al. (2019). Charting cellular identity during human in vitro β -cell differentiation. *Nature* 569, 368–373. <https://doi.org/10.1038/s41586-019-1168-5>.
7. Elorbany, R., Popp, J.M., Rhodes, K., Strober, B.J., Barr, K., Qi, G., Gilad, Y., and Battle, A. (2022). Single-cell sequencing reveals lineage-specific dynamic genetic regulation of gene expression during human cardiomyocyte differentiation. *PLoS Genet* 18, e1009666. <https://doi.org/10.1371/journal.pgen.1009666>.
8. Yu, G., Wang, L.-G., and He, Q.-Y. (2015). ChIPseeker: an R/Bioconductor package for ChIP peak annotation, comparison and visualization. *Bioinformatics* 31, 2382–2383. <https://doi.org/10.1093/bioinformatics/btv145>.
9. Rigau, M., Juan, D., Valencia, A., and Rico, D. (2019). Intronic CNVs and gene expression variation in human populations. *PLoS Genet* 15, e1007902. <https://doi.org/10.1371/journal.pgen.1007902>.
10. Ernst, J., and Kellis, M. (2017). Chromatin-state discovery and genome annotation with ChromHMM. *Nat Protoc* 12, 2478–2492. <https://doi.org/10.1038/nprot.2017.124>.
11. Buenrostro, J.D., Wu, B., Chang, H.Y., and Greenleaf, W.J. (2015). ATAC-seq: A Method for Assaying Chromatin Accessibility Genome-Wide. *Curr Protoc Mol Biol* 109, 21.29.1–21.29.9. <https://doi.org/10.1002/0471142727.mb2129s109>.
12. Lin, K.C., Park, H.W., and Guan, K.-L. (2017). Regulation of the Hippo Pathway Transcription Factor TEAD. *Trends Biochem Sci* 42, 862–872. <https://doi.org/10.1016/j.tibs.2017.09.003>.
13. Zanconato, F., Forcato, M., Battilana, G., Azzolin, L., Quaranta, E., Bodega, B., Rosato, A., Bicciato, S., Cordenonsi, M., and Piccolo, S. (2015). Genome-wide association between YAP/TAZ/TEAD and AP-1 at enhancers drives oncogenic growth. *Nat Cell Biol* 17, 1218–1227. <https://doi.org/10.1038/ncb3216>.
14. Cserjesi, P., and Olson, E.N. (1991). Myogenin induces the myocyte-1016 specific enhancer binding factor MEF-2 independently of other muscle-specific gene products. *Mol Cell Biol* 11, 4854–4862. <https://doi.org/10.1128/mcb.11.10.4854-4862.1991>.
15. Moustafa, A., Hashemi, S., Brar, G., Grigull, J., Ng, S.H.S., Williams, D., Schmitt-Ulms, G., and McDermott, J.C. (2023). The MEF2A transcription factor interactome in cardiomyocytes. *Cell Death Dis* 14, 240. <https://doi.org/10.1038/s41419-023-05665-8>.
16. GTEx Consortium (2020). The GTEx Consortium atlas of genetic regulatory effects across human tissues. *Science* 369, 1318–1330. <https://doi.org/10.1126/science.aaz1776>.
17. Rojas, A., Schachterle, W., Xu, S.-M., Martín, F., and Black, B.L. (2010). Direct transcriptional regulation of Gata4 during early endoderm specification is controlled by FoxA2 binding to an intronic enhancer. *Dev Biol* 346, 346–355. <https://doi.org/10.1016/j.ydbio.2010.07.032>.
18. Yamak, A., Latinkic, B.V., Dali, R., Temsah, R., and Nemer, M. (2014). Cyclin D2 is a GATA4 cofactor in cardiogenesis. *Proc Natl Acad Sci U S A* 111, 1415–1420. <https://doi.org/10.1073/pnas.1312993111>.
19. Moya, I.M., and Halder, G. (2019). Hippo–YAP/TAZ signalling in organ regeneration and regenerative medicine. *Nat Rev Mol Cell Biol* 20, 211–226. <https://doi.org/10.1038/s41580-018-0086-y>.
20. Desjardins, C.A., and Naya, F.J. (2016). The Function of the MEF2 Family of Transcription Factors in Cardiac Development, Cardiogenomics, and Direct Reprogramming. *J Cardiovasc Dev Dis* 3, 26. <https://doi.org/10.3390/jcdd3030026>.
21. Fogarty, M.P., Panhuis, T.M., Vadlamudi, S., Buchkovich, M.L., and Mohlke, K.L. (2013). Allele specific transcriptional activity at type 2 diabetes-associated single nucleotide polymorphisms in regions of pancreatic islet open chromatin at the JAZF1 locus. *Diabetes* 62, 1756–1762. <https://doi.org/10.2337/db12-0972>.
22. Tan, W.L.W., Anene-Nzelu, C.G., Wong, E., Lee, C.J.M., Tan, H.S., Tang, S.J., Perrin, A., Wu, K.X., Zheng, W., Ashburn, R.J., et al. (2020). Epigenomes of Human Hearts Reveal New Genetic Variants Relevant for Cardiac Disease and Phenotype. *Circulation Research* 127, 761–777. <https://doi.org/10.1161/CIRCRESAHA.120.317254>.

23. Urbut, S.M., Wang, G., Carbonetto, P., and Stephens, M. (2019). Flexible statistical methods for estimating and testing effects in genomic studies with multiple conditions. *Nat Genet* 51, 187–195. <https://doi.org/10.1038/s41588-018-0268-8>.
24. Bulik-Sullivan, B.K., Loh, P.-R., Finucane, H.K., Ripke, S., Yang, J., Schizophrenia Working Group of the Psychiatric Genomics Consortium, Patterson, N., Daly, M.J., Price, A.L., and Neale, B.M. (2015). LD Score regression distinguishes confounding from polygenicity in genome-wide association studies. *Nat Genet* 47, 291–295. <https://doi.org/10.1038/ng.3211>.
25. Mitchelmore, J., Grinberg, N.F., Wallace, C., and Spivakov, M. (2020). Functional effects of variation in transcription factor binding highlight long-range gene regulation by epromoters. *Nucleic Acids Res* 48, 2866–2879. <https://doi.org/10.1093/nar/gkaa123>.
26. Pickrell, J.K., Berisa, T., Liu, J.Z., Séguirel, L., Tung, J.Y., and Hinds, D.A. (2016). Detection and interpretation of shared genetic influences on 42 human traits. *Nat Genet* 48, 709–717. <https://doi.org/10.1038/ng.3570>.
27. Zou, Y., Carbonetto, P., Xie, D., Wang, G., and Stephens, M. (2023). Fast and flexible joint fine1057 mapping of multiple traits via the Sum of Single Effects model. Preprint, <https://doi.org/10.1101/2023.04.14.536893> <https://doi.org/10.1101/2023.04.14.536893>.
28. Alasoo, K., Rodrigues, J., Mukhopadhyay, S., Knights, A.J., Mann, A.1059 L., Kundu, K., HIPSCI Consortium, Hale, C., Dougan, G., and Gaffney, D.J. (2018). Shared genetic effects on chromatin and gene expression indicate a role for enhancer priming in immune response. *Nat Genet* 50, 424–431. <https://doi.org/10.1038/s41588-018-0046-7>.
29. Currin, K.W., Erdos, M.R., Narisu, N., Rai, V., Vadlamudi, S., Perrin, H.J., Idol, J.R., Yan, T., Albanus, R.D., Broadaway, K.A., et al. (2021). Genetic effects on liver chromatin accessibility identify disease regulatory variants. *Am J Hum Genet* 108, 1169–1189. <https://doi.org/10.1016/j.ajhg.2021.05.001>.
30. Ayguñ, N., Liang, D., Crouse, W.L., Keele, G.R., Love, M.I., and Stein, J.L. (2023). Inferring cell-type specific causal gene regulatory networks during human neurogenesis. *Genome Biol* 24, 130. <https://doi.org/10.1186/s13059-023-02959-0>.
31. Chang, C.C., Chow, C.C., Tellier, L.C., Vattikuti, S., Purcell, S.M., and Lee, J.J. (2015). Second generation PLINK: rising to the challenge of larger and richer datasets. *Gigascience* 4, 7. <https://doi.org/10.1186/s13742-015-0047-8>.
32. Csárdi, G., Nepusz, T., Müller, K., Horvát, S., Traag, V., Zanini, F., and Noom, D. (2024). igraph for R: R interface of the igraph library for graph theory and network analysis. Version v2.0.2 (Zenodo). <https://doi.org/10.5281/ZENODO.7682609> <https://doi.org/10.5281/ZENODO.7682609>.
33. Schubert, M., Pérez Lanuza, L., Wöste, M., Dugas, M., Carmona, F.D., Palomino-Morales, R.J., Rassam, Y., Heilmann-Heimbach, S., Tüttelmann, F., Kliesch, S., et al. (2022). A GWAS in Idiopathic/Unexplained Infertile Men Detects a Genomic Region Determining Follicle-Stimulating Hormone Levels. *J Clin Endocrinol Metab* 107, 2350–2361. <https://doi.org/10.1210/clinem/dgac165>.
34. He, Y., Koido, M., Sutoh, Y., Shi, M., Otsuka-Yamasaki, Y., Munter, H.M., BioBank Japan, Morisaki, T., Nagai, A., Murakami, Y., et al. (2023). East Asian-specific and cross-ancestry genome-wide meta analyses provide mechanistic insights into peptic ulcer disease. *Nat Genet* 55, 2129–2138. <https://doi.org/10.1038/s41588-023-01569-7>.
35. Tabassum, R., Chauhan, G., Dwivedi, O.P., Mahajan, A., Jaiswal, A., Kaur, I., Bandesh, K., Singh, T., Mathai, B.J., Pandey, Y., et al. (2013). Genome-wide association study for type 2 diabetes in Indians identifies a new susceptibility locus at 2q21. *Diabetes* 62, 977–986. <https://doi.org/10.2337/db12-0406>.
36. Momozawa, Y., Merveille, A.-C., Battaille, G., Wiberg, M., Koch, J., Willesen, J.L., Proschowsky, H.F., Gouni, V., Chetboul, V., Tiret, L., et al. (2020). Genome wide association study of 40 clinical measurements in eight dog breeds. *Sci Rep* 10, 6520. <https://doi.org/10.1038/s41598-020-63457-y>.
37. Mahajan, A., Taliun, D., Thurner, M., Robertson, N.R., Torres, J.M., Rayner, N.W., Payne, A.J., Steinthorsdottir, V., Scott, R.A., Grarup, N., et al. (2018). Fine-mapping type 2 diabetes loci to single variant resolution using high-density imputation and islet-specific epigenome maps. *Nat Genet* 50, 1505–1513. <https://doi.org/10.1038/s41588-018-0241-6>.

38. Horikoshi, M., Beaumont, R.N., Day, F.R., Warrington, N.M., Kooijman, M.N., Fernandez-Tajes, J., Feenstra, B., van Zuydam, N.R., Gaulton, K.J., Grarup, N., et al. (2016). Genome-wide associations for birth weight and correlations with adult disease. *Nature* 538, 248–252. <https://doi.org/10.1038/nature19806>.
39. Mostafavi, H., Spence, J.P., Naqvi, S., and Pritchard, J.K. (2023). Systematic differences in discovery of genetic effects on gene expression and complex traits. *Nat Genet* 55, 1866–1875. <https://doi.org/10.1038/s41588-023-01529-1>.
40. Kim, S., and Wysocka, J. (2023). Deciphering the multi-scale, quantitative cis-regulatory code. *Mol Cell* 83, 373–392. <https://doi.org/10.1016/j.molcel.2022.12.032>.
41. Ding, Z., Ni, Y., Timmer, S.W., Lee, B.-K., Battenhouse, A., Louzada, S., 1102 Yang, F., Dunham, I., Crawford, G.E., Lieb, J.D., et al. (2014). Quantitative genetics of CTCF binding reveal local sequence effects and different modes of X-chromosome association. *PLoS Genet* 10, e1004798. <https://doi.org/10.1371/journal.pgen.1004798>.
42. Timmers, P.R.H.J., Wilson, J.F., Joshi, P.K., and Deelen, J. (2020). Multivariate genomic scan implicates novel loci and haem metabolism in human ageing. *Nat Commun* 11, 3570. <https://doi.org/10.1038/s41467-020-17312-3>.
43. Wen, C., Margolis, M., Dai, R., Zhang, P., Przytycki, P.F., Vo, D.D., Bhattacharya, A., Matoba, N., Tang, M., Jiao, C., et al. (2024). Cross-ancestry atlas of gene, isoform, and splicing regulation in the developing human brain. *Science* 384, eadh0829. <https://doi.org/10.1126/science.adh0829>.

Referees' reports, second round of review

Reviewer #1: The authors have done an excellent job addressing concerns raised by the reviewers by applying new analysis and modifying the manuscript accordingly. This is a well written manuscript describing significant amount of interesting data and trying to answer an important puzzle of the low overlapping between eQTL and GWAS hit. The main issue right now is the underpower, which will be addressed in the future scaled up studies.

Reviewer #2: The authors have done an outstanding job responding to the comments. The paper has improved as a result.

Reviewer #3: Comments enter in this field will be shared with the author; your identity will remain anonymous.

The authors have put in substantial work and have responded satisfactorily to almost all of the comments. Their responses, and re-reading of text in light of the responses, has raised a few additional questions.

New comments

1. The sentence -

Line 99 "In summary, our study shows that integrative QTL analyses explain a large proportion of GWAS loci that do not colocalize with eQTLs alone. "

- can be read to imply that a large proportion of the 95% of GWAS loci not localizing with eQTL are explained by ca/HA QTLs. This is not what the paper found: non-eQTL HA/ca QTLs colocalizations explain only 5% of the total GWAS, ie a very small % of the remaining loci.

Please modify this sentence to reflect what is found in this paper or use the formulation in the discussion that HA/ca QTL analysis has the potential to explain larger proportions of GWAS loci as HA/ca QTL sample sizes increase.

2. The Figure S4 referred to below now appears to be Figure S6. Please check all of the Figure numbers in the text.

"6.3.1 eGene discovery rate

1442 To examine the relative power of identifying eQTLs, we compared the three iPSCORE EDev-like tissues to the 49 tissues in the GTEx Consortium1 and showed that they had similar eGene discovery rates (Figure S4). We note that sequencing depth differences across molecular data types may have resulted in different caPeak and haPeak discovery rates."

Comments on responses.

Original Comment 12

Line 265: The inclusion of the TF TBX20 example seems somewhat random. Why

was there a focus on this locus in particular?

Summary of Author comment:: Two new examples were added.

New reviewer comment 12: The presence of TF binding site disruption does not prove that a given variant is causal. Please revise the sentence below to indicate this. Also the word "is" looks like it should be removed.

Line 404: "These findings recapitulate a previous observation that MOPCV (rs9573330) creates a MEF2A binding site in the KLF12 intron41 and demonstrate that the true causal variant is often does not have the highest posterior probability."

Original comment 3

3. The Supplemental Figure 3 Cardiac marker plots (b) shows samples with low levels of the cardiac cell markers in the cardiac cells. What criteria were used to include a cardiac cell line and why were these samples included in the analysis?

We thank the reviewer for the question. During differentiation, we utilized 388 a lactate selection which depletes non-cardiac cells. We and others have shown that CVPC lines are mostly composed of cardiomyocytes and epicardial cell types, including smooth muscle, cardiac fibroblasts, and vascular endothelial cells 2. The proportion of these cell types vary across CVPC lines, therefore filtering samples based on cardiomyocyte markers like NKX2-5 and TNNT would remove samples that were more epicardial-like. We note that PEER factor 1 is highly correlated with experimental measurements of cardiac troponin (%cTnT; Figure S16d-f) which shows that this cell type heterogeneity is accounted for in the QTL analyses. For these reasons, we only filtered based on technical factors, as described in the Methods (Lines 1176-1180, 1204-1209, 1235- 1239).

New reviewer comments 3:

a. All of the CVCP selection criteria described in the methods appear to be for cardiomyocytes. If there are no cardiomyocytes for some samples (no or almost no cardiomyocyte TFs expression, as shown in Figure S5) how would the sample beat as described in the methods?

b. For people who may not realize the composition of the CVPC lines can vary so widely, please show an epicardial cell marker in Figure S5 and insert a scatter plot of the cardiomyocyte vs epicardial marker to show the inverse relationship, and describe in the Figure legend.

c. Correction for the composition with a PEER factor doesn't remove the problem of loss of power due to cell type heterogeneity. Please add a sentence to the discussion about the loss of power in detecting QTLs within CVPC given the heterogeneity in differentiation.

Original Comment 16.

16. Please make a supplemental table that has the number of unrelated individuals for each tissue and omic type to help people judge the amount of data, or give an estimate of the effective number of people analyzed for each tissue and omics type.

Author response. We have included a "Unrelated_Set" column in each of the Table S2 sheets (RNA-seq, ATAC-seq, and ChIP-seq) that indicate the optimal set of samples, including unrelated individuals within families, for each of the eight datasets.

New reviewer comment 16. Please make a table that gives the number of unrelated individuals for the 8 data types. Giving the raw data in Table S2 is not sufficient, as a reader would need to calculate the numbers of unrelated individuals for each data type from this information. This unrelated table could be part of Figure 1, or of a new supplementary Figure that has three tables: 1) a copy of the total numbers matrix table in Figure 1, 2) an unrelated individuals matrix table and 3) potentially (See comment 26) a matrix table of the mean +/-SD of reads well mapped (whichever is most standard) for each of the 8 cell and data type.

Original Comment 18.

18. Figure 2f: Please explicitly state the comparison groups used for each of the three enrichment tests, in the Figure and in the Methods section.

Authors reply: We have added in the Figure 2f Legend and in the Methods (Lines 1467-1474) a description of the comparison groups for each enrichment.

New reviewer comment 18. For the sentence below please explicitly state what the background is (I think it is the first part of the sentence) and please also state the background group in the legend.

Line 1467 "Using only ATAC-seq peaks with at least one predicted TFBS, we performed a Fisher's Exact test to calculate the enrichment of 444 JASPAR motifs of TFs that are expressed in CVPCs (see 6.1.3 ATAC-seq Peak Transcription Factor Predictions) in caPeaks, caPeaks-haPeaks and CVPC ATAC-seq peaks without a caQTL (no QTL). "

Original comment 25.

25. Line 652: Clarify here that samples are from D0 so are (presumably) iPSC cells. Thank you for this comment. The cell state of naive and D0 iPSCs differs slightly. For the iPSC RNA-seq, data was generated on conventionally cultured naive iPSCs. ATAC-seq and H3K27ac

ChIP-seq was generated from D0 iPSC, which corresponds to the start of CVPC differentiation, where iPSCs were treated with Rho kinase (ROCK) inhibitor for 24 hours, but not treated with the compounds to initiate the cardiac differentiation. We have described this in more detail in the Methods (Lines 1046-1048).

New reviewer comment 25 For clarity of the methods sentence below, add "at D0" during the CVPC differentiation. In the CVPC differentiation section, define the " ** " in steps a and b. I assume the **'s are to indicate these are the samples used for the IPSC ATAC-seq and H3K27ac.

"iPSC nuclear pellets for the ATAC-seq and H3K27ac ChIP-seq assays were collected during the CVPC differentiation protocol (see below: 3.2 CVPdifferentiation)."

Original comment 26

26. Line 776 and others: There are substantial differences in the sequencing depths for the molecular phenotypes across the tissues. Consider a table containing this information and commenting on the effects of differential sequencing depth in the discussion on the discovery of QTLs.

We thank the reviewer for this comment. We have included sequencing information in Supplemental Table 2 for the RNA-seq, ATAC-seq, and H3K27ac ChIP-seq samples. We have also included text in the Methods to highlight the effects of differential sequencing depth (Lines 1444-1445).

New reviewer comment 26. As currently provided the reader would need to calculate the cell type sequencing depth from the individual level data in Table S2 for ATAC-seq and H3K27ac chip seq. Please add the average and SD of the relevant well mapped read counts for each data type to a table (Could be combined with tables from Comment 16 above, or add these numbers to the relevant section of the Methods (missing for each cell type for ATAC-seq and H3K27ac).

Reviewer #4: I am generally satisfied with the authors' response and appreciate the substantial effort they have made to address the reviewers' comments. I have a few minor comments in case the authors may find them helpful as they finalize their revisions.

Specifically, I believe my comments 1 and 7 may have been slightly misunderstood. Regarding comment 1: I appreciated the authors' response, which addressed most of my points, e.g., regarding the distance to the TSS. However, an additional aspect of my comment was to compare the different QTL types with respect to the properties of their target genes. For instance, the authors identified 27,881 adult-specific eQTLs and 2,299 EDev-specific eQTLs. Are the target genes (eGenes) for these different QTL types different in interesting ways, such as the fraction being high pLI? There is a need to control for power in this analysis, for example, by focusing on the top 1,000 or 2,000 eGenes in each category. How do these eGenes compare to random genes? A similar analysis could be informative for comparisons between eQTLs, caQTLs, and haQTLs.

Regarding comment 7: By "discovery bias", I was referring to the inherent biases in eQTL discovery in general, rather than biases specific to any particular method. For example, eQTLs are more likely to be discovered near promoters, which is unrelated to the choice of mashr or any other method.

Authors' response to the second round of review

We thank the reviewer for the clarifications on their original comments. Below we address the reviewer's comments 1 and 7, which we misinterpreted in the first round of revisions. To orient our responses, we have copied the original comments and pasted in italics.

Reviewer #2

The authors have put in substantial work and have responded satisfactorily to almost all of the comments. Their responses, and re-reading of text in light of the responses, has raised a few additional questions.

We thank the reviewer for carefully reading the manuscript and providing these additional suggestions.

New comments

1. The sentence -

Line 99 "In summary, our study shows that integrative QTL analyses explain a large proportion of GWAS loci that do not colocalize with eQTLs alone."

- can be read to imply that a large proportion of the 95% of GWAS loci not localizing with eQTL are explained by ca/HA QTLs. This is not what the paper found: non-eQTL HA/ca QTLs colocalizations explain only 5% of the total GWAS, ie a very small % of the remaining loci. Please modify this sentence to reflect what is found in this paper or use the formulation in the discussion that HA/ca QTL analysis has the potential to explain larger proportions of GWAS loci as HA/ca QTL sample sizes increase.

We agree that, as written, the sentence does not accurately reflect our study's findings. The sentence was intended to be suggestive for future studies, rather than declarative for our study, therefore we have edited the sentence: "In summary, our study shows that integrative multiomic QTL analyses **could** explain a large proportion of GWAS loci that do not colocalize with eQTLs alone."

2. The Figure S4 referred to below now appears to be Figure S6. Please check all of the Figure numbers in the text.

"6.3.1 eGene discovery rate

1442 To examine the relative power of identifying eQTLs, we compared the three iPSCORE EDev-like tissues to the 49 tissues in the GTEx Consortium1 and showed that they had similar eGene discovery rates (Figure S4). We note that sequencing depth differences across molecular data types may have resulted in different caPeak and haPeak discovery rates."

We thank the reviewer for identifying this discrepancy. The order of the supplemental figures has changed, but we have corrected this error and confirmed the remaining references to supplemental figures are accurate.

Comments on responses.

Original Comment 12

Line 265: *The inclusion of the TF TBX20 example seems somewhat random. Why was there a focus on this locus in particular?*

Summary of Author comment:: Two new examples were added.

New reviewer comment 12: The presence of TF binding site disruption does not prove that a given variant is causal. Please revise the sentence below to indicate this. Also the word "is" looks like it should be removed.

Line 404: *"These findings recapitulate a previous observation that MOPCV (rs9573330) creates a MEF2A binding site in the KLF12 intron41 and demonstrate that the true causal variant is often does not have the highest posterior probability."*

We have updated this sentence: *"These findings recapitulate a previous observation that MOPCV (rs9573330) creates a MEF2A binding site in the KLF12 intron⁴¹ and demonstrate that the **most likely** causal variant often does not have the highest posterior probability."*

Original comment 3

3. *The Supplemental Figure 3 Cardiac marker plots (b) shows samples with low levels of the cardiac cell markers in the cardiac cells. What criteria were used to include a cardiac cell line and why were these samples included in the analysis?*

We thank the reviewer for the question. During differentiation, we utilized a lactate selection which depletes non-cardiac cells. We and others have shown that CVPC lines are mostly composed of cardiomyocytes and epicardial cell types, including smooth muscle, cardiac fibroblasts, and vascular endothelial cells². The proportion of these cell types vary across CVPC lines, therefore filtering samples based on cardiomyocyte markers like NKX2-5 and TNNT would remove samples that were more epicardial-like. We note that PEER factor 1 is highly correlated with experimental measurements of cardiac troponin (%cTnT; Figure S16d-f) which shows that this cell type heterogeneity is accounted for in the QTL analyses. For these reasons, we only filtered based on technical factors, as described in the Methods (Lines 1176-1180, 1204-1209, 1235- 1239).

New reviewer comments 3:

a. All of the CVCP selection criteria described in the methods appear to be for cardiomyocytes. If there are no cardiomyocytes for some samples (no or almost no cardiomyocyte TFs expression, as shown in Figure S5) how would the sample beat as described in the methods?

We have edited the sentence in the Methods to clarify that only “CVPCs with a **sufficient** proportion of cardiomyocytes” typically exhibited beating cells between days 7 to 9 of differentiation.

b. For people who may not realize the composition of the CVPC lines can vary so widely, please show an epicardial cell marker in Figure S5 and insert a scatter plot of the cardiomyocyte vs epicardial marker to show the inverse relationship, and describe in the Figure legend.

Due to the comments being provided to us late in the revision process by the editor and in consultation with the editor we have chosen not to add this material. However, this information is present in a manuscript that we published previously. *Association of Human iPSC Gene Signatures and X Chromosome Dosage with Two Distinct Cardiac Differentiation Trajectories*, D'Antonio-Chronowska, Agnieszka et al. *Stem Cell Reports*, Volume 13, Issue 5, 924 – 938.

C. Correction for the composition with a PEER factor doesn't remove the problem of loss of power due to cell type heterogeneity. Please add a sentence to the discussion about the loss of power in detecting QTLs within CVPC given the heterogeneity in differentiation.

We disagree with the reviewer and believe that the inclusion of these PEER factors likely increases power by accounting for heterogeneity. In Figure S14, we observe that experimental measurements for the cell type composition of CVPCs (%cTnT) and PPCs (%PDX1+NKX6-1+) and both are strongly correlated with their corresponding first PEER factor, suggesting that inclusion of PEER factors accounts for cellular heterogeneity. Evidence for the increase of power would be that you detect more QTLs after including those PEER factors as covariates (Figure S15).

Of course, mapping QTLs in pure cell types (i.e. only cardiomyocytes) would likely have more power than heterogenous tissues. However, nearly all bulk QTL studies, including GTEx, are performed using samples generated from heterogenous tissues.

To address the reviewer's comment, we have added the following text to the Limitations:

Our analysis revealed that the PEER factors were highly correlated with sequencing depth-related sample attributes (i.e. number of reads passing filters) and biological variables, such as cellular heterogeneity (i.e. %cTnT in CVPCs and %PDX1-NKX6.1 in PPCs; Figure S14). During optimization (Figure S15), we observed that the inclusion of PEER factors increased the number of QTLs mapped, suggesting that they correct for both technical and biological variability and improve statistical power.

Original Comment 16.

16. Please make a supplemental table that has the number of unrelated individuals for each tissue and omic type to help people judge the amount of data, or give an estimate of the effective number of people analyzed for each tissue and omics type.

Author response. We have included a “Unrelated_Set” column in each of the Table S2 sheets (RNA-seq, ATAC-seq, and ChIP-seq) that indicate the optimal set of samples, including unrelated individuals within families, for each of the eight datasets.

New reviewer comment 16. Please make a table that gives the number of unrelated individuals for the 8 data types. Giving the raw data in Table S2 is not sufficient, as a reader would need to calculate the numbers of unrelated individuals for each data type from this information. This unrelateds table could be part of Figure 1, or of a new supplementary Figure that has three tables: 1) a copy of the total numbers matrix table in Figure 1, 2) an unrelated individuals matrix table and 3) potentially (See comment 26) a matrix table of the mean +/-SD of reads well mapped (whichever is most standard) for each of the 8 cell and data type.

To address the reviewer's comment, we have created another tab on Table S2 which includes the total number of samples, number of unique individuals, number of unrelated individuals, the mean number and standard deviation of mapped reads for each dataset. We feel that this is sufficient information for a reader to understand how many unrelated individuals there are and the distribution of sequencing depths. The editor has requested that we limit the amount of additional text, figures and supplemental figures, as the publication already exceeds several limits, therefore we have not generated a new supplemental figure.

Original Comment 18.

18. *Figure 2f: Please explicitly state the comparison groups used for each of the three enrichment tests, in the Figure and in the Methods section.*

Authors reply: We have added in the Figure 2f Legend and in the Methods (Lines 1467-1474) a description of the comparison groups for each enrichment.

New reviewer comment 18. For the sentence below please explicitly state what the background is (I think it is the first part of the sentence) and please also state the background group in the legend.

Line 1467 "Using only ATAC-seq peaks with at least one predicted TFBS, we performed a Fisher's Exact test to calculate the enrichment of 444 JASPAR motifs of TFs that are expressed in CVPCs (see 6.1.3 ATAC-seq Peak Transcription Factor Predictions) in caPeaks, caPeaks-haPeaks and CVPC ATAC-seq peaks without a caQTL (no QTL). "

We have edited the sentence and the legend.

Results: We next binned the peaks into three categories: ATAC-seq peaks without a caQTL (non-caPeak), caPeaks that overlap haPeaks (caPeaks-haPeak), and caPeaks that do not overlap an haPeak (caPeaks). For each category, we performed Fisher's Exact tests to evaluate the enrichment of TFBSs, using the other two categories as background (Figure 2f).

Figure 2f Legend: Heatmap showing the enrichment of TFBSs in CVPC ATAC-seq peaks without caQTLs (non-caPeaks), with caQTLs overlapping haQTLs (caPeak-haPeak), and with caQTLs not overlapping haQTLs (caPeaks). For each category, a two-sided Fisher's Exact test was performed to test the enrichment of TFBSs, using the other two categories as background.

Original comment 25.

25. *Line 652: Clarify here that samples are from D0 so are (presumably) iPSC cells.*

Thank you for this comment. The cell state of naive and D0 iPSCs differs slightly. For the iPSC RNA-seq, data was generated on conventionally cultured naive iPSCs. ATAC-seq and H3K27ac ChIP-seq was generated from D0 iPSC, which corresponds to the start of CVPC differentiation, where iPSCs were treated with Rho kinase (ROCK) inhibitor for 24 hours, but not treated with the compounds to initiate the cardiac differentiation. We have described this in more detail in the Methods (Lines 1046-1048).

New reviewer comment 25 For clarity of the methods sentence below, add "at D0" during the CVPC differentiation. In the CVPC differentiation section, define the " ** " in steps a and b. I assume the **'s are to indicate these are the samples used for the IPSC ATAC-seq and H3K27ac.

We have added this clarifying text in the methods (see below).

- a. *At D0 of the CVPC differentiation, 142 iPSC lines from 129 individuals were collected and frozen as nuclear pellets for the ATAC-seq assay**.*
- b. *At D0 of the CVPC differentiation, 43 iPSC lines from 41 individuals were collected and frozen as nuclear pellets for the H3K27ac ChIP-seq assay**.*

Asterisks (**) indicate that the iPSC ATAC-seq and H3K27ac ChIP-seq data were generated for iPSC lines treated with ROCK inhibitor before initiating CVPC differentiation, whereas the RNA-seq data were generated on ROCK inhibitor-naïve iPSC lines.

We also edited the sentence below:

"iPSC nuclear pellets for the ATAC-seq and H3K27ac ChIP-seq assays were collected at D0 of the CVPC differentiation protocol (see below: 3.2 CVPC differentiation)."

Original comment 26

26. Line 776 and others: There are substantial differences in the sequencing depths for the molecular phenotypes across the tissues. Consider a table containing this information and commenting on the effects of differential sequencing depth in the discussion on the discovery of QTLs.

We thank the reviewer for this comment. We have included sequencing information in Supplemental Table 2 for the RNA-seq, ATAC-seq, and H3K27ac ChIP-seq samples. We have also included text in the Methods to highlight the effects of differential sequencing depth (Lines 1444-1445).

New reviewer comment 26. As currently provided the reader would need to calculate the cell type sequencing depth from the individual level data in Table S2 for ATAC-seq and H3K27ac chip seq. Please add the average and SD of the relevant well mapped read counts for each data type to a table (Could be combined with tables from Comment 16 above, or add these numbers to the relevant section of the Methods (missing for each cell type for ATAC-seq and H3K27ac).

See response to Comment 16.

Reviewer #4: I am generally satisfied with the authors' response and appreciate the substantial effort they have made to address the reviewers' comments. I have a few minor comments in case the authors may find them helpful as they finalize their revisions. Specifically, I believe my comments 1 and 7 may have been slightly misunderstood. Regarding comment 1: I appreciated the authors' response, which addressed most of my points, e.g., regarding the distance to the TSS. However, an additional aspect of my comment was to compare the different QTL types with respect to the properties of their target genes. For instance, the authors identified 27,881 adult-specific eQTLs and 2,299 EDev-specific eQTLs. Are the target genes (eGenes) for these different QTL types different in interesting ways, such as the fraction being high pLI? There is a need to control for power in this analysis, for example, by focusing on the top 1,000 or 2,000 eGenes in each category. How do these eGenes compare to random genes? A similar analysis could be informative for comparisons between eQTLs, caQTLs, and haQTLs.

Original Comment 1. As motivation, in the introduction, the authors mention a recent study showing systematic differences in the discovery of eQTLs and GWAS hits (Mostafavi et al.). For the search for missing GWAS colocalizations, it would be useful to characterize: (i) whether/how adult-tissue QTLs are different from development-like tissue QTLs and (ii) how different QTL types compare to each other and control variants in terms of the properties of the tentative target genes (e.g., selective constraint), distance to genes, etc. My expectation is that the discovered QTLs are biased away from GWAS variants in important ways (though not exactly like eQTLs), considering the small sample sizes used in the study. [I note that target genes are not known for caQTLs and haQTLs, for which perhaps the nearest gene can serve as a proxy.]

We thank the reviewer for their clarifications. In the original round of revisions, the reviewer requested two analyses to further characterize the target gene properties (lead variant distance to TSS and evolutionary constraint [pLI score]) between: 1) adult-shared and EDev-specific QTLs (stage-specificity), and 2) eQTLs, caQTLs, and haQTLs (QTL type). In the reviewer's original comment 7 (see below), we interpreted the source of the discovery biases to arise from our original approach where we calculated the LD of the iPSCORE QTL lead variants with lead variants from adult QTLs in QTLbase1. Based on this interpretation of

discovery bias, we suspected that the smaller effect sizes that we originally reported could be an artifact of weaker QTLs being absent from existing QTL databases. Therefore, we employed mashr2, which is a more established method that is typically used to identify cell type-specific eQTLs, but can be employed to identify specificity between any two or more conditions. The application of mashr limits our ability to compare the target gene properties between adultshared and EDev-specific QTLs (requested analysis #1). Briefly, our original approach used LD between lead variants to determine whether a stage-specific QTL exists at the eGene resolution, while our modified mashr approach investigates patterns of effect sizes across tissues to calculate the specificity of pairs of SNPs and genes, which is a higher resolution than the original method. Therefore, we used the mashr SNP-eGene pair classifications to annotate iPSC, CVPC, and PPC eQTLs as either EDev-specific, Shared, or No Association (not significant). Multiple iPSCORE tissues can have an eQTL for the same gene and, in certain cases, the eQTL signals are distinct and have different temporal annotations (i.e. one tissue has an Adult-shared eQTL and another tissue has a EDev-specific). Therefore, transitions from the higher SNP-eGene pair resolution to the lower eGene resolution makes this analysis challenging.

To address the reviewer's requested analysis #1, we first annotated the iPSCORE eGenes as Shared or EDev-specific using their mashr annotations. We then annotated them with their gnomad3 pLI scores. eGenes with at least one EDev-specific and Shared eQTL were filtered, resulting in 1,123 EDev-specific and 4,949 Shared eGenes. We note that the distribution of pLI scores exhibits an extreme bimodal distribution (Figure A), therefore selecting the top hits for each annotation is not possible. When we attempted this, the top 500 Shared genes all had a pLI ~ 1 , while the top 500 EDev-specific range was 3.1×10^{-347} to 1 (Figure B). We randomly selected 500 EDev-specific and 500 Shared eGenes and observed that EDev-specific eGenes were under higher evolutionary constraint than Shared eGenes (Mann Whitney U-test p-value = 4.9×10^{-15} ; Figure C), which is consistent with our original observations and those described in Wen et al⁴. Due to the complexity of these analyses, we excluded them from the revised manuscript.

The reviewer's requested analysis #2 was to evaluate the target gene property (pLI and TSS distance) differences between QTL types. We indirectly show that eQTLs tend to be closer to the TSS compared to caQTLs and haQTLs in Figures 2e and 4e. Figure 2e shows that chromatin QTLs are more enriched in enhancers, while eQTLs are more enriched in promoters. In Figure 4e, although this analysis shows that Complex QTLs are closer to TSSs compared to Singletons across QTL types, the distance distributions are more even across the 100kb window for chromatin QTLs, while the eQTL distance distributions skew closer to the TSS (below).

Figure 4e from manuscript:

We previously analyzed difference of pLI scores between QTL types and didn't observe notable differences. While we share the reviewer's interest in this question, we felt that our dataset was underpowered for these analyses and their inclusion would detract from the more convincing messages within our manuscript.

Regarding comment 7: By "discovery bias", I was referring to the inherent biases in eQTL discovery in general, rather than biases specific to any particular method. For example, eQTLs are more likely to be discovered near promoters, which is unrelated to the choice of mashr or any other method.

Original Comment 7. I do not think I agree with the characterization of developmental-unique QTLs versus adult-shared QTLs, and I suspect the results in Figures 6 and 7a are a consequence of the analysis design rather than due to different intrinsic features of developmental and adult QTLs.

First, the text implies that developmental eGenes are high pLI genes. But my understanding is that this is relative to adult-shared eGenes, not random/control genes. Also, the number of eGenes used for comparison is not matched (532 versus 8,257). This is problematic because the properties of eGenes vary by their rank (or degree of significance). A proper test would be to compare constraint for the top 532 eGenes in both, as well as random/control genes.

Second, I think the trends shown in Figure 6 can be explained by power and discovery biases. Specifically, for unique and shared QTLs acting on the same gene, natural selection is stronger on the shared QTLs. Thus, discovery of shared QTLs will be more biased towards weakly selected genes. As such, I think the statement in the Discussion that "unique regulatory variants are strongly selected against" is incorrect; in fact, I think the opposite is true. The authors can test this by exploring the reverse, i.e., adult-unique versus shared QTLs, or by comparing QTLs with respect to measures such as CADD score.

Third, the interpretation of the differential colocalization of developmental-unique QTLs versus adult-shared QTLs with GWAS hits is complicated. As mentioned above, shared QTLs are likely skewed towards less GWAS-relevant genes. But this can be counterbalanced by the fact that shared QTLs have more paths to contribute to trait variations (through activity in multiple causal tissues). This is also compounded by different discovery biases mentioned above that have huge effects on colocalization analyses. For example, shared QTLs are more likely to be promoter variants, which have larger effects and are thus easier to detect in colocalization.

Our interpretation of the reviewer's "discovery bias" led us to utilize a more widely accepted, robust approach, mashr. While addressing the original comment, we hypothesized that the discovery bias was driven by an enrichment of QTLs with stronger effects in existing databases, thus spuriously skewing QTLs classified as EDev-specific towards weaker signals. This could have indirectly affected our interpretation of the pLI and colocalization results. We agree with the reviewer that, "the differential colocalization of developmental-unique QTLs versus adultshared

QTLs with GWAS hits is complicated" for a variety reasons, therefore we decided to deemphasize these points in the revised manuscript.

Although we misinterpreted the original comment, we thank the reviewer for making it because it led to a more robust approach and improved the quality of the manuscript.

References

1. Huang, D., Feng, X., Yang, H., Wang, J., Zhang, W., Fan, X., Dong, X., Chen, K., Yu, Y., Ma, X., et al. (2023). QTLbase2: an enhanced catalog of human quantitative trait loci on extensive molecular phenotypes. *Nucleic Acids Res* 51, D1122–D1128. <https://doi.org/10.1093/nar/gkac1020>.
 2. Uebachs, S.M., Wang, G., Carbonetto, P., and Stephens, M. (2019). Flexible statistical methods for estimating and testing effects in genomic studies with multiple conditions. *Nat Genet* 51, 187–195. <https://doi.org/10.1038/s41588-018-0268-8>.
 3. Chen, S., Francioli, L.C., Goodrich, J.K., Collins, R.L., Kanai, M., Wang, Q., Alföldi, J., Watts, N.A., Vittal, C., Gauthier, L.D., et al. (2024). A genomic mutational constraint map using variation in 76,156 human genomes. *Nature* 625, 92–100. <https://doi.org/10.1038/s41586-023-06045-0>.
 4. Wen, C., Margolis, M., Dai, R., Zhang, P., Przytycki, P.F., Vo, D.D., Bhattacharya, A., Matoba, N., Tang, M., Jiao, C., et al. (2024). Cross-ancestry atlas of gene, isoform, and splicing regulation in the developing human brain. *Science* 384, eadh0829. <https://doi.org/10.1126/science.adh0829>.
-